# EEG-FM-Bench: A Comprehensive Benchmark for the Systematic Evaluation and Diagnostic Analyses of EEG Foundation Models

Wei Xiong [1]   Jiangtong Li [1]   Jie Li [2 1]   Kun Zhu [1]   Changjun Jiang [1]

## Abstract

Electroencephalography foundation models (EEG-FMs) have advanced brain signal analysis, but the lack of standardized evaluation benchmarks impedes model comparison and scientific progress. Current evaluations rely on inconsistent protocols that render cross-model comparisons unreliable, while a lack of diagnostic analyses obscures the internal mechanisms driving transfer efficiency and scaling behaviors. To address this, we introduce **EEG-FM-Bench**, a unified system for the standardized evaluation of EEG-FMs. The benchmark integrates 14 datasets across 10 paradigms and incorporates diverse experimental settings, including multiple fine-tuning strategies, task organizations, and classifier configurations, supported by tools for gradient and representation analysis. Our experiments and analysis reveal several critical insights: (1) multi-task learning often acts as a useful regularizer that mitigates overfitting in data-scarce EEG contexts, although negative transfer can arise under specific task paradigms; (2) pre-training efficiency is currently limited by gradient conflicts between reconstruction objectives and downstream tasks; (3) under released checkpoints and a matched downstream protocol, model or data scale alone does not fully explain transfer performance, while objective alignment, adaptation compatibility, and EEG-specific design appear to be important factors. This benchmark enables fair comparison and reproducible analysis, providing a step toward fairer comparison and more interpretable analysis of EEG-FMs. Code is available at https://github.com/xw1216/EEG-FM-Bench.

[1]School of Computer Science and Technology, Tongji University, Shanghai, China [2]Translational Research Center, Shanghai Yangzhi Rehabilitation Hospital (Shanghai Sunshine Rehabilitation Center), China. Correspondence to: Jiangtong Li <jiangtongli@tongji.edu.cn>, Jie Li <jieli@tongji.edu.cn>.

*Proceedings of the 43rd International Conference on Machine Learning*, Seoul, South Korea. PMLR 306, 2026. Copyright 2026 by the author(s).

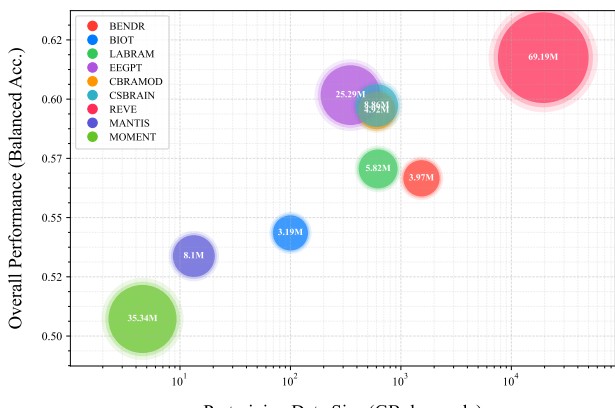

*Figure 1.* **Scaling analysis: Performance vs. Data and Model Size.** We plot the overall balanced accuracy against pre-training data size. Bubble size represents the model parameter count. The pretraining data storage volume is estimated based on the total duration of the raw data signal.

## 1. Introduction

Due to the high temporal resolution, non-invasiveness, and cost-effectiveness, EEG is widely used in neuroscience and clinical scenarios, ranging from cognition to pathology (Ramantani et al., 2016). Following the success of large-scale pre-training in CV (Radford et al., 2021) and NLP (Devlin et al., 2019), researchers have developed EEG foundation models (EEG-FM) (Jiang et al., 2024b) to learn general representations from large-scale unlabeled EEG datasets. These models aim to overcome major obstacles in EEG analysis, including high inter-subject and intra-subject variability, and the scarcity of expertly annotated datasets (Rashid et al., 2020). While early works adapted architectures from other domains like BENDR (Kostas et al., 2021), recent studies focus on designing structures and pre-training strategies tailored to EEG characteristics, exemplified by CBraMod (Wang et al., 2025a), CSBrain (Zhou et al., 2026), and REVE (El Ouahidi et al., 2026).

Despite the rapid progress of EEG-FMs, their **evaluation practices remain fragmented**. While foundation models aim for broad generalization among tasks and subjects, current evaluations often rely on task- or dataset-specific settings that contradict this goal. Existing studies benchmark models with inconsistent downstream datasets,

preprocessing pipelines, and evaluation protocols, rendering cross-architecture comparisons unreliable and masking true pre-training advances due to varying experimental setups (Lai et al., 2025). In practice, many methods compensate by introducing dataset-specific architectural and strategy tweaks, rather than learning representations that generalize across settings. For example, EEGPT (Wang et al., 2024) freezes the backbone with linear probe, REVE (El Ouahidi et al., 2026) employs two-stage fine-tuning, while CBraMod (Wang et al., 2025a) and CS-Brain (Zhou et al., 2026) use fixed-size classifiers tailored to specific datasets. Moreover, existing **evaluations rarely go beyond end-task metrics**. Detailed technical analyses remain scarce, such as *gradient relations across datasets*, *evolution of intermediate representations during fine-tuning*, and *identifying components that drive knowledge transfer*. As a result, the mechanisms underlying pre-training efficacy remain unclear, particularly why scaling data often fails to yield proportional downstream gains (Jiang et al., 2024b).

Recent studies have benchmarked subject transfer (Wu et al., 2025) and adaptation techniques like LoRA (Hu et al., 2022) for EEG-FMs (Lee et al., 2025). However, several important aspects remain underexplored, including multi-task capability, classifier configuration, pretraining efficiency, and internal model dynamics such as gradient flow and feature representations. Therefore, a unified benchmark is required to ensure fair comparison and provide diagnostic analyses, replacing fragmented results with interpretable insights.

To address these issues, we introduce **EEG-FM-Bench**, a benchmark designed for the systematic and standardized evaluation of pretrained EEG-FMs. **The benchmark integrates various downstream tasks, unified data processing pipelines, consistent evaluation protocols, and a rich set of analytical and visualization tools**. All components are implemented within a unified open-source codebase to ensure reproducibility and fair comparison. Rather than optimizing for specific settings, this benchmark focuses on the general capabilities of EEG-FMs. Specifically, EEG-FM-Bench covers 14 datasets across 10 common EEG paradigms, including motor imagery, sleep staging, emotion recognition, disease diagnosis, and *etc*. It incorporates diverse configurations to assess pre-training quality and generalization: (1) **three fine-tuning strategies** (frozen-backbone, full-parameter, and LoRA (Hu et al., 2022)); (2) **two task setups** (single-task and multi-task); and (3) **three classifier configuration** (average pooling, attention pooling, and temporal-spatial-embedding dimension aggregation).

We evaluate seven pre-trained EEG-FMs and two general time-series models: BIOT (Yang et al., 2023), BENDR (Kostas et al., 2021), LaBraM (Jiang et al., 2024b), EEGPT (Wang et al., 2024), CBraMod (Wang et al., 2025a), CSBrain (Zhou et al., 2026), REVE (El Ouahidi et al., 2026),

Mantis (Feofanov et al., 2025), and Moment (Goswami et al., 2024). To probe internal mechanisms, our benchmark employs visualization and quantitative tools, measuring model behavior and optimization dynamics through gradient cosine similarity, subspace affinity, Centered Kernel Alignment (CKA) (Kornblith et al., 2019), and Representational Similarity Analysis (RSA) (Kriegeskorte et al., 2008). This integrated design enables both standardized benchmarking and diagnostic analysis, helping diagnose where models succeed or fail across datasets and suggesting possible mechanisms behind these observed transfer behaviors.

Our evaluation on EEG-FM-Bench yields several key observations: 1) Frozen-backbone transfer exhibits a substantial generalization gap, indicating that current pre-trained EEG representations are not yet sufficiently task-ready for direct downstream use. 2) Multi-task fine-tuning often improves robustness, consistent with a regularization effect and possible cross-task knowledge sharing, but it can also introduce negative transfer for certain paradigms, highlighting the need for better task grouping and conflict-aware optimization. 3) Richer classifier heads mainly benefit motor imagery tasks, whereas simpler pooling heads remain a more reliable default for most other paradigms alleviating overfitting issue. 4) Layer-wise analysis suggests that pre-training stabilizes the Transformer backbone and shifts a large part of the adaptation burden to the input embedding and normalization-related components. 5) Gradient analyses show weak alignment between reconstruction and classification objectives, suggesting that **objective alignment** and adaptation compatibility are important bottlenecks beyond nominal model or data scale. These findings motivate future EEG-FM development toward better pre-training objectives, more stable adaptation interfaces, and carefully designed multi-task learning rather than relying on scale alone. Our contributions can be summarized as:

- We introduce **EEG-FM-Bench**, an open-source evaluation suite for EEG-FMs, integrating standardized protocols, diverse tasks, and diagnostic tools for end-to-end assessment.

- We conduct an empirical study on SOTA EEG-FMs, establishing baselines to compare their performance and generalization across various fine-tuning strategies.

- We analyze gradients and intermediate representations to identify potential pretraining bottlenecks and empirical architectural patterns that can inform future model and objective design.

## 2. Related Works

### 2.1. EEG Foundation Models

Early efforts adapt techniques from other fields; for instance, BENDR (Kostas et al., 2021) applies contrastive

learning from speech processing, while others focus on masked signal modeling. These include BrainBERT (Wang et al., 2023), which operates on EEG spectrograms, and EEG2Rep (Mohammadi Foumani et al., 2024) and Brant (Zhang et al., 2023), which perform masking in the latent space. Regarding input tokenization, BIOT (Yang et al., 2023) introduces channel-independent methods for variable inputs, while LaBraM (Jiang et al., 2024b) and EEGFormer (Wan et al., 2023) utilize vector-quantized approaches for raw and frequency-domain signals. In terms of architecture, EEGPT (Wang et al., 2024) integrates spatio-temporal alignment, CBraMod (Wang et al., 2025a) uses criss-cross attention, and CSBrain (Zhou et al., 2026) proposes a cross-scale structure. REVE (El Ouahidi et al., 2026) scales up pre-training, refining objectives using layer-wise features and employing 4D positional encodings. Furthermore, models like Brant-2 (Yuan et al., 2024) combine scalp with intracranial EEG, while LEAD (Wang et al., 2025b) targets clinical challenges such as Alzheimer's disease. This rapid evolution complicates fair comparisons between approaches, highlighting the need for standardized benchmarks to evaluate and analyze progress.

## 2.2. EEG Benchmarks

The BCI and broader EEG analysis communities contend with a reproducibility crisis, partly due to a lack of unifying standards. Early efforts, such as BCI competitions, provide public datasets and standardized metrics. Recent competitions continue this trend; for instance, BEETL (Wei et al., 2022) emphasizes transfer learning, while the EEG Foundation Challenge (Aristimunha et al., 2025) focuses on cross-subject cognitive tasks, though limited to the HBN dataset. MOABB (Jayaram & Barachant, 2018) offers an open-source evaluation platform, yet its scope restricts to specific paradigms like motor imagery, SSVEP, and P300. Other benchmarks target specific tasks or architectures, including denoising (EEGdenoiseNet (Zhang et al., 2021)), emotion recognition (LibEER (Liu et al., 2025)), Parkinson's diagnosis (Avola et al., 2025), and GNNs (GNN4EEG (Zhang et al., 2024)). Specifically for EEG-FMs, AdaBrain-Bench (Wu et al., 2025) and Lee et al. (2025) benchmark subject transfer and adaptation techniques. However, these studies overlook critical aspects, including multi-task capability, classifier configuration, pre-training efficiency, and internal dynamics such as gradient flow. Consequently, a unified benchmark for current EEG-FMs is lacking. Such a resource is necessary to ensure reproducibility and provide a fair basis for comparing methodological innovations.

## 3. Benchmark Pipeline

Fig. 2 provides an overview of the EEG-FM-Bench. Designed for fair, reproducible, and diagnostic evaluation, the

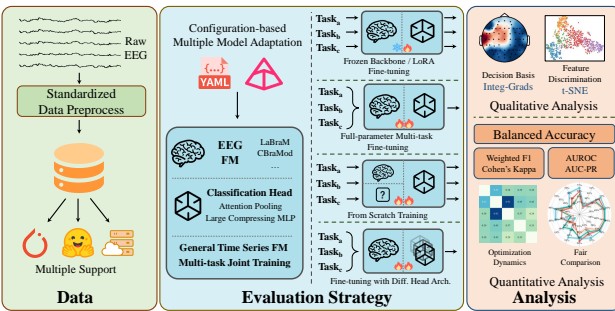

*Figure 2.* Overview of the **EEG-FM-Bench** framework. **Data Management:** A unified pipeline standardizes preprocessing across 14 datasets covering 10 canonical EEG paradigms. **Configurable Evaluation:** A configuration-based system facilitates flexible assessments by decoupling model backbones from decoder heads, supporting diverse fine-tuning strategies (*e.g.*, frozen-backbone, LoRA, multi-task). **Diagnostic Analyses:** The platform provides both qualitative visualizations (*e.g.*, decision basis, feature discrimination) and quantitative metrics to examine model performance and optimization dynamics.

system comprises three core modules: **Data Management**, **Configurable Evaluation**, and **Diagnostic Analyses**. It employs a modular open-source structure with unified abstractions for datasets, backbones, classifiers, and training setups, ensuring that different EEG-FMs are evaluated under *matched* preprocessing and optimization conditions. Extending beyond downstream accuracy, the pipeline supports **fine-tuning dynamics** by coupling gradient-space geometry with representation-space similarity, enabling controlled comparisons across `model initialization` (scratch *v.s.* pre-trained), `training phrase` (pre-train *v.s.* fine-tune), `training regime` (single-task *v.s.* multi-task), and `model component` (patching *v.s.* transformer).

## 3.1. Data Management

**Task and Dataset Curation.** Our benchmark incorporates 14 public datasets covering 10 canonical EEG paradigms, including motor imagery (BCIC-2a, PhysioMI, Mimul-11), emotion recognition (SEED, SEED-V, SEED-VII), sleep stage classification (HMC), seizure detection (Siena), mental stress assessment (Workload), abnormal detection (TUAB), event type classification (TUEV), visual target detection (Things-EEG-2), Alzheimer's Disease recognition (ADFTD), and slowing event classification (TUSL) (see Appx. A for more details). Through these diverse datasets, EEG-FM-Bench serves both as a ranking tool and a diagnostic instrument to identify architectural and representational weaknesses. The framework features an extensible API, enabling the integration of custom datasets and user-defined data assembly configurations.

**Standardized Data Processing.** Since inconsistent processing leads to incomparable results, we implement a standardized pipeline applied across datasets. This pipeline comprises the following steps (see Appx. B.1 for details): **1.**

**Selection**: The system selects data based on specified event markers and channel sets. **2. Filtering**: A band-pass filter (0.1–100 Hz) and a notch filter (50 or 60 Hz) remove noise while preserving signal information. **3. Resampling**: Signals are downsampled to model-specific target rates based on their pretraining setting. **4. Segmentation**: Continuous data is segmented into fixed-length windows (*e.g.*, 4-second trials) and assigned to splits based on task-specific rules. **5. Splitting**: Datasets are partitioned into three splits using a subject-independent strategy while preserving label distribution. For emotion recognition, a subject-dependent strategy aligns with common protocols. **6. Formatting:** Processed samples are saved in efficient formats ("`Parquet`" for storage, "`Arrow`" for accelerated loading).

**Overlap-sensitive datasets.** Some model-dataset pairs are overlap-sensitive because widely used downstream benchmarks may also appear in the pre-training corpora of released EEG-FMs. Specifically, LaBraM overlaps with SEED, while EEGPT overlaps with SEED and PhysioMI. We retain these datasets because they are standard reference points in the EEG literature and are important for continuity with prior work. However, we explicitly mark these overlap-sensitive cases in the results table and treat them as overlap-sensitive evidence rather than standalone evidence generalization. To avoid inflating aggregate comparisons, these overlapped model–dataset pairs are excluded from the overall performance calculation used in Fig. 1, and we avoid drawing conclusions from these subsets in isolation.

### 3.2. Configurable Evaluation

We evaluate pre-trained EEG-FMs by structuring the design into three dimensions: **fine-tuning strategy**, **task setup**, and **classifier head**, which allows controlled ablations on adaptability, transfer, and decoding inductive bias, while keeping data processing and optimization across models.
**Fine-tuning strategies.** We consider three strategies that probe different levels of parameter adaptation:

- **Frozen-backbone.** Only the classifier head is trained while the backbone remains frozen. This evaluates the intrinsic quality of pre-trained representations.
- **Full-parameter.** All model parameters are fine-tuned on downstream data, assessing how effectively pre-trained features adapt to the target task distribution.
- **Parameter-efficient.** The backbone is fine-tuned using LoRA, enabling efficient adaptation with fewer trainable parameters and reduced memory footprint.

**Task setups.** To evaluate both within-task adaptation and cross-task knowledge sharing, each fine-tuning strategy can be instantiated under two task setups:

- **Single-task setting.** The model is fine-tuned and evaluated on one downstream dataset at a time, reflecting the standard pre-training transfer protocol.

- **Multi-task setting.** The model is fine-tuned on a mixture constructed from all downstream tasks, evaluating whether joint training improves generalization and benefits each paradigm. We employ resampling to mitigate dataset imbalance and stabilize optimization.

**Classifier heads.** To decouple backbone capability from decoding inductive bias, we implement three classifier heads:

- **MLP with patch average pooling.** We average pool patch-level features and apply an MLP classifier.
- **MLP with dimension compression.** We apply a large MLP that reduces temporal, spatial, and embedding, testing whether a high-capacity improves performance.
- **MLP with attention pooling.** We use attention pooling to aggregate patches into a global representation, enabling adaptive weighting and improving performance when discriminative features are sparse.

### 3.3. Diagnostic Analyses

We provide both **standard performance evaluation** and **diagnostic analysis** of fine-tuning dynamics.
**Performance Metrics.** To address class imbalance, we select metrics including Balanced Accuracy, Weighted F1, AUROC, AUC-PR, and Cohen's Kappa (see Appx. B.4 for detailed definition). Within our benchmark, Balanced Accuracy is used for all classification task, AUROC and AUC-PR are used for binary classification tasks, and Cohen's Kappa and Weighted F1 are used to multi-class classification tasks.

**Fine-tuning Dynamics Analysis.** We analyze fine-tuning dynamics by linking *gradient-space geometry* with *representation-space similarity* across four settings: (1) `initialization` (scratch *vs.* pre-trained); (2) `phase` (pre-training *vs.* fine-tuning); (3) `regime` (single-task *vs.* multi-task); and (4) `component` (patching *vs.* transformer). At designated intervals, we collect parameter gradients and extract intermediate layer representations via forward hooks on fixed probe batches. To reduce memory overhead and ensure consistent comparisons, we compress gradients and features using CountSketch (Charikar et al., 2004) hashing projector with projection dimension 1024, followed by $\ell_2$-normalization.

**Gradient-space analysis.** For each parameter group and each condition (initialization, dataset or training phrase), we compute alignment measures by cosine similarity between projected gradient directions: $\cos(\mathbf{g}_a, \mathbf{g}_b) = \frac{\langle \mathbf{g}_a, \mathbf{g}_b \rangle}{\|\mathbf{g}_a\|_2 \|\mathbf{g}_b\|_2}$.
To capture the dominant subspace structure of optimization directions, we compute subspace affinity. For each condition, we construct a gradient matrix, apply PCA, and average the top singular values of the projection matrix $U^\top V$: $A = U^\top V$, Affinity $= \frac{1}{k} \sum_{i=1}^{k} \sigma_i(A)$, where $U$ and $V$ denote the $k$-dimensional orthonormal bases for two conditions, and $\sigma_i(\cdot)$ is the $i$-th singular value. In detail,

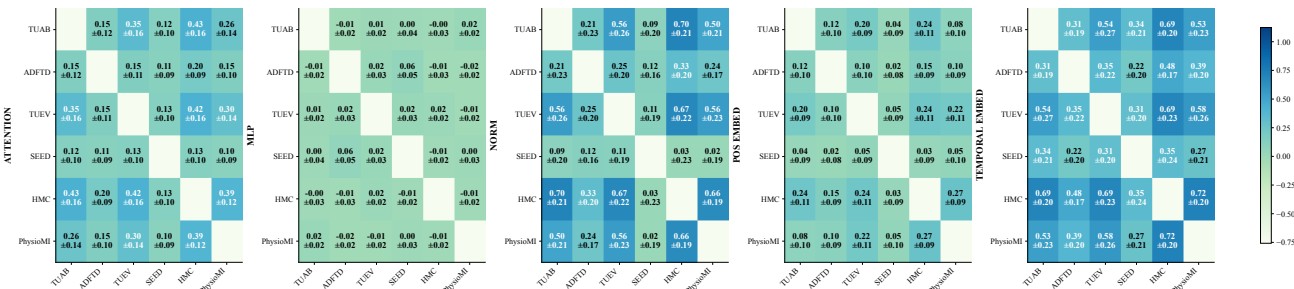

Figure 3. **Cross-task gradient correlation analysis.** The heatmaps visualize the pairwise cosine similarity of gradients between different tasks across specific model components, revealing how tasks interact during optimization. Best viewed via zoom-in.

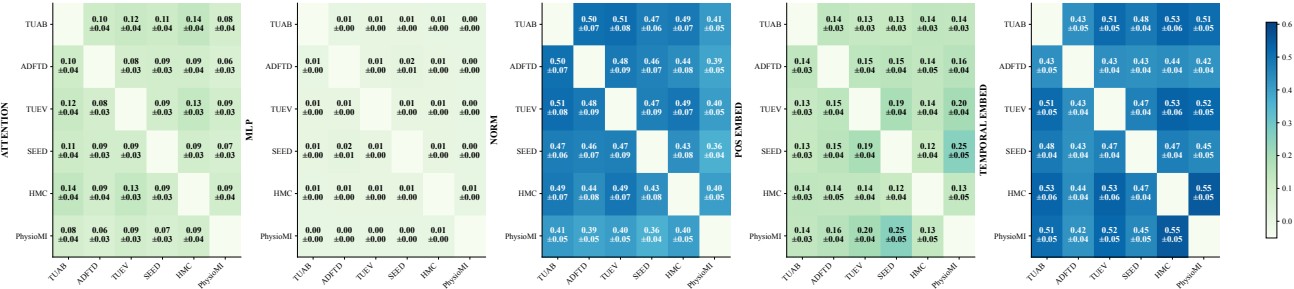

Figure 4. **Cross-task subspace affinity analysis.** This metric quantifies the geometric alignment of optimization subspaces among tasks, indicating the extent to which different tasks share a common optimization direction within each component. Best viewed via zoom-in.

subspace affinity is computed from a rank-6 PCA basis, as rank 6 captures most variation in model-space affinity and higher ranks yield saturated estimates.

We further quantify gradient conflicts using a metric inspired by PCGrad (Yu et al., 2020). For paired conditions, we compare gradients at aligned steps, identifying conflicts where the cosine similarity is negative: Conflict = $\mathbb{I}[\cos(\mathbf{g}_a, \mathbf{g}_b) < 0]$. Conflict frequency is the empirical mean over samples, while the mean conflict angle is derived from these negative cosine values.

We also analyze the distribution of gradient energy across parameter groups. The energy proportion $E_{\text{axis},G}$ for a specific condition and parameter group $G$ is defined as:

$$E_{\text{axis},G} = \frac{\sum_t \|\mathbf{g}^{(t)}_{\text{axis},G}\|_2^2}{\sum_{G' \in \mathcal{G}} \sum_t \|\mathbf{g}^{(t)}_{\text{axis},G'}\|_2^2}, \quad (1)$$

where $\mathcal{G}$ is the set of all parameter groups, and $t$ denotes the training step. This distribution highlights which parameter groups dominate the gradient updates.

**Representation-space analysis.** Representations are extracted from model layers using fixed probe batches, with their similarity quantified via Centered Kernel Alignment (CKA) and Representational Similarity Analysis (RSA). For CKA, we compute linear kernels $K_X = XX^\top$ and $K_Y = YY^\top$ for representations $X, Y \in \mathbb{R}^{n \times d}$, where $n$ is the sample size and $d$ is the feature dimension. We then calculate the Hilbert-Schmidt Independence Criterion (HSIC) (Freiwald & Tsao, 2010) using the centering matrix

$H = I_n - \frac{1}{n}\mathbf{1}_n\mathbf{1}_n^\top$ ($\mathbf{1}_n$ is an $n$-dimensional vector of ones):

$$\text{HSIC}(K_X, K_Y) = \frac{1}{(n-1)^2}\text{tr}(K_X H K_Y H), \quad (2)$$

where $\text{tr}(\cdot)$ denotes the matrix trace operator. CKA normalizes HSIC to achieve scale invariance:

$$\text{CKA}(X,Y) = \frac{\text{HSIC}(K_X,K_Y)}{\sqrt{\text{HSIC}(K_X,K_X)\cdot\text{HSIC}(K_Y,K_Y)}}. \quad (3)$$

RSA quantifies the similarity of representations from two models via representational dissimilarity matrices (RDMs). For a representation matrix $X \in \mathbb{R}^{n \times d}$, we compute the RDM using sample-wise Pearson correlation:

$$\text{RDM}_{ij} = 1 - \text{corr}_{\text{Pear}}(x_i, x_j). \quad (4)$$

We extract the upper triangular elements of the RDMs and compute the Spearman correlation between them:

$$\text{RSA}(X,Y) = \text{Corr}_{\text{Spear}}(\text{vec}(\text{RDM}_X), \text{vec}(\text{RDM}_Y)). \quad (5)$$

This provides a similarity metric for layer-wise or condition-aware feature evolvement comparison.

**Probing protocol.** All analyses are conducted during *fine-tuning with fixed budget*. Every $n$ steps, we probe gradients and representations using pre-sampled batches fixed throughout the run. This ensures consistent inputs for comparisons across optimization process and conditions (*e.g.*, scratch *vs.* pre-trained). We repeat the procedure across 10 random seeds and report aggregated statistics.

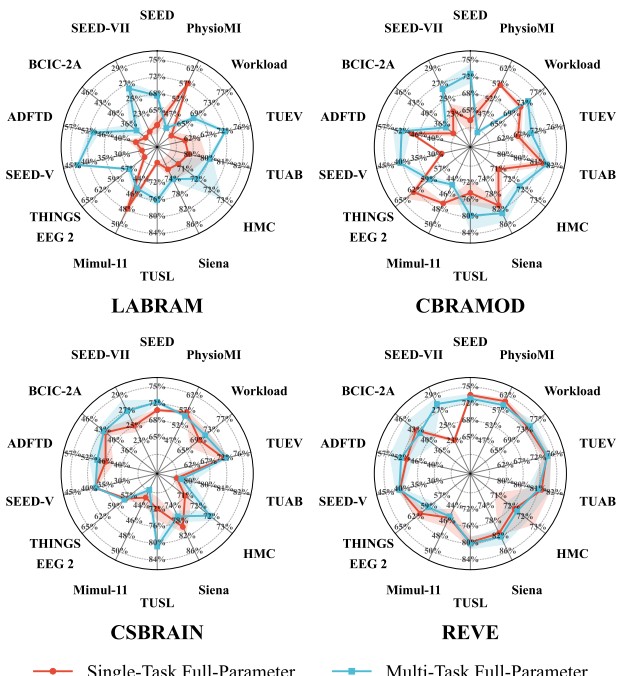

*Figure 5.* Performance comparison between single-task and multi-task fine-tuning on 14 downstream tasks. Best viewed via zoom-in.

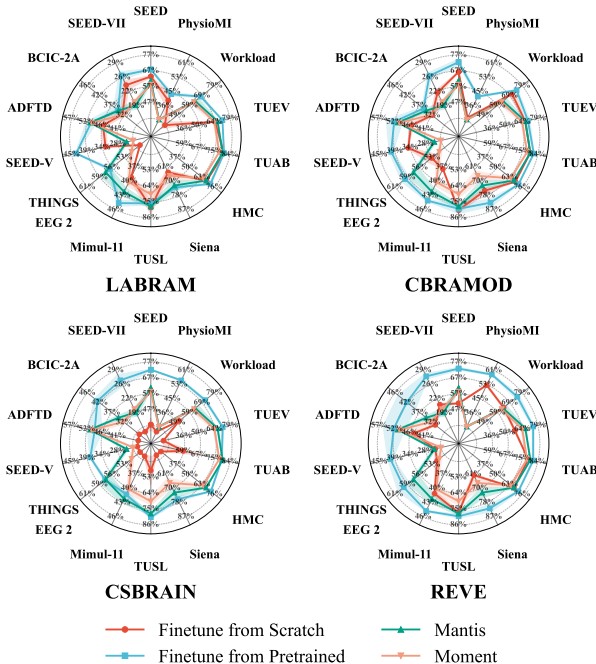

*Figure 6.* Performance comparison between fine-tuning from scratch (red) and from pre-trained model (blue). General time-series models are included for reference. Best viewed via zoom-in.

## 4. Experiment and Analysis

### 4.1. Experiment Setup

All models are evaluated within the EEG-FM-Bench framework. For each experiment, a classifier head is attached to the EEG-FM for downstream tasks. We report the mean and standard deviation across five independent runs with different random seeds. For analysis, we use 32 fixed batches to probe gradients and intermediate representations throughout the training budget. Unless otherwise mentioned, all the results are reported under multi-task, full-parameter, average pooling decoder with pretrained EEG-FMs. **See Appx. B for implementation details; Appx. C, D for detailed results and analysis; Appx. E for detailed visualization**.

### 4.2. Impact of Task Organization

**Performance Comparison.** Standard fine-tuning isolates downstream datasets, often missing cross-task knowledge sharing. Therefore, we compare two strategies: (1) **Single-Task**, where models are fine-tuned on each dataset individually; and (2) **Multi-Task**, where models are fine-tuned jointly on a mixture of all tasks. Fig. 5 presents the performance comparison across four representative models using radar charts. First, multi-task learning (blue) improves many model-dataset pairs and is especially helpful for small or unstable datasets (*e.g.*, ADFTD), where isolated single-task optimization is more prone to variance and overfitting. However, the improvement is not uniform. Negative transfer appears in specific paradigms, particularly motor imagery

(*e.g.*, PhysioMI and BCIC-2A) and visual target detection (THINGS-EEG2), where single-task baselines can occasionally outperform joint training. These results suggest that multi-task supervision is a useful regularizer, but its benefit depends on task compatibility and may require task balancing or conflict-aware optimization.

**Mechanism Analysis: Gradient and Subspace Dynamics.** We examine optimization dynamics via gradient alignment and feature space geometry to explain the multi-task generalizability. Using LaBraM (others in Appx. D) as a case, we compute inter-task gradient **Correlation** and **Subspace Affinity** across model components. Figures 3 and 4 show that **Normalization** and **Temporal Embedding** layers exhibit consistently high gradient correlations. This suggests these components learn task-agnostic temporal dynamics shared across datasets. Conversely, **MLP** and **Attention** layers show lower affinity, implying they capture task-specific patterns. High correlation in shared components aligns optimization of fundamental extractors, preventing the model from collapsing into task-specific local minima. Thus, we recommend multi-task joint training as a strong default when tasks are compatible and labeled datasets can be pooled, while treating task grouping and conflict mitigation as important under large discrepancies.

### 4.3. Impact of Pre-training

**Performance Benchmarking: EEG-FMs vs. Time-FMs.** We evaluate pre-training utility by comparing three

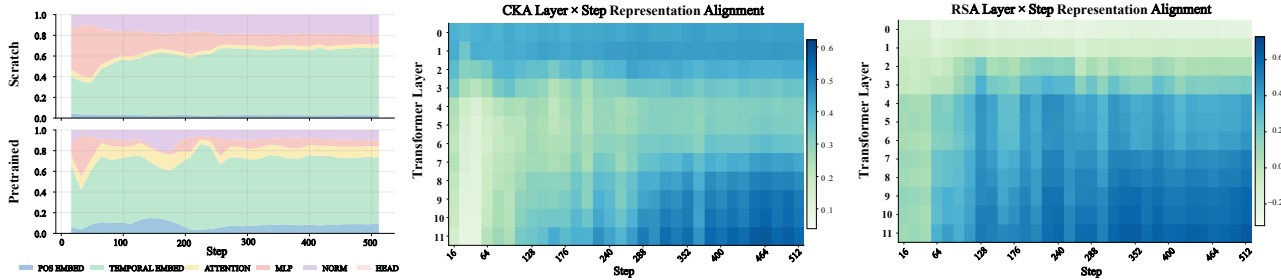

*Figure 7.* **Evolution of optimization dynamics during fine-tuning.** Left: Relative gradient norm intensity across modules for Scratch vs. Pre-trained settings. Mid and Right: Layer-wise CKA and RSA between Scratch and Pre-trained models over training steps, showing representational convergence. Best viewed via zoom-in.

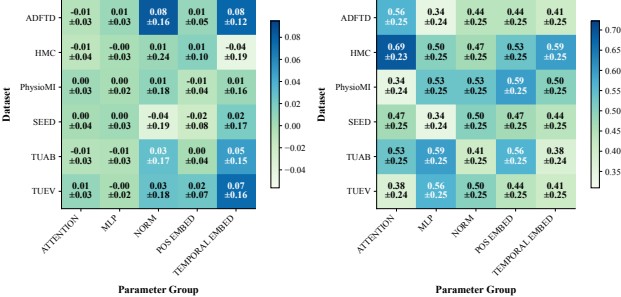

*Figure 8.* **Gradient alignment between pre-training and downstream tasks.** Left: Cosine similarity of gradients derived from Reconstruction and Classification losses. Right: Probability of gradient conflicts (sign disagreement). Best viewed via zoom-in.

paradigms: scratch, pre-trained checkpoints, and general time-series models (Mantis, Moment). As shown in Fig. 6, pre-training (blue) generally outperforms training from scratch (red), especially on complex tasks such as SEED-V and BCIC-2A, indicating that pre-trained EEG-FMs usually provide a stronger initialization. However, the performance gap varies across datasets and can be narrow in selected cases, suggesting that the learned representations are not uniformly task-ready. Moreover, general time-series models achieve competitive performance on some tasks despite being trained on non-EEG data, indicating that part of the transferable structure may come from generic temporal modeling rather than EEG-specific pre-training alone.

**Optimization Dynamics.** We visualize LaBraM's training dynamics to explain pre-training advantages. We monitor gradient norms and representational similarity every 16 steps on a controlled fine-tuning subset. Fig. 7 (left) shows the gradient norm intensity. In *Pre-trained* settings, optimization focuses on the Temporal Embedding (green area), while the backbone (Attention, MLP, Norm) remains stable. In contrast, *Scratch* requires intensive optimization across all components. This indicates pre-training stabilizes the Transformer backbone for sequence modeling. Therefore, fine-tuning primarily adapts the Temporal Embedding to bridge the gap between raw signals and latent space. Figs. 7 (mid) and (right) track representational evolution via CKA and RSA, showing progressively increasing align-

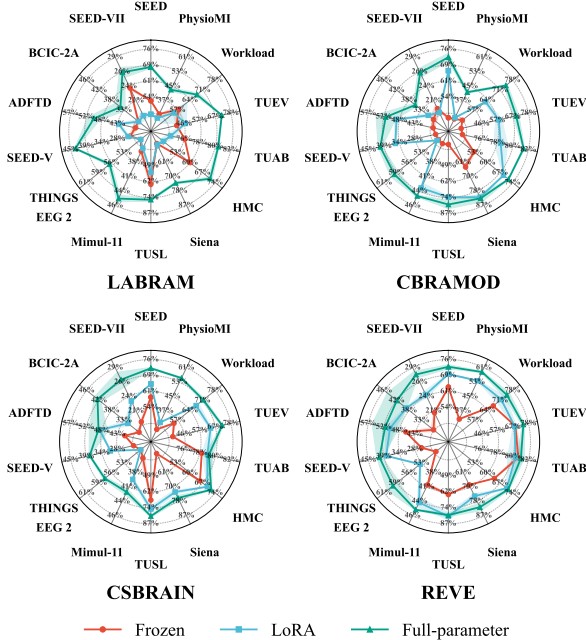

*Figure 9.* Performance comparison among fine-tuning strategies.

ment scores among fine-tuning. CKA shows strengthening structural correspondence, while RSA indicates converging sample geometry. Therefore, the multi-task fine-tuning acts as a attractor, guiding the scratch model to align representations with the pre-trained model, validating the robustness of the learned features. Notably, CSBrain tends to collapse under the scratch setting (see Appx. D for detailed analysis).

**Inefficiency in Pre-training: Gradient Analysis.** Performance gains of EEG-FMs are often not proportional to the pre-training data scale. We investigate this inefficiency by analyzing gradient alignment between pre-training (Masked Reconstruction, MSE) and downstream (Classification, Cross-Entropy) objectives on identical samples. To enable scalable comparison, we project gradients to a CountSketch space with dimension ($d = 1024$) before computing correlations; this projection dimension is distinct from the rank-6 PCA basis used later for subspace-affinity analysis. Fig. 8 (left) shows near-zero or negative

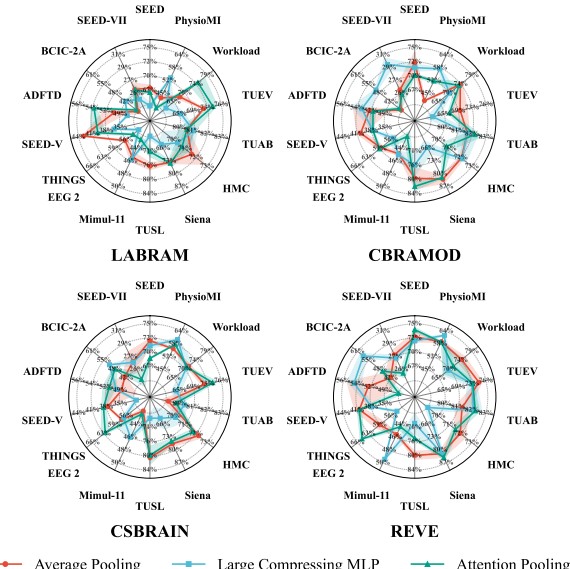

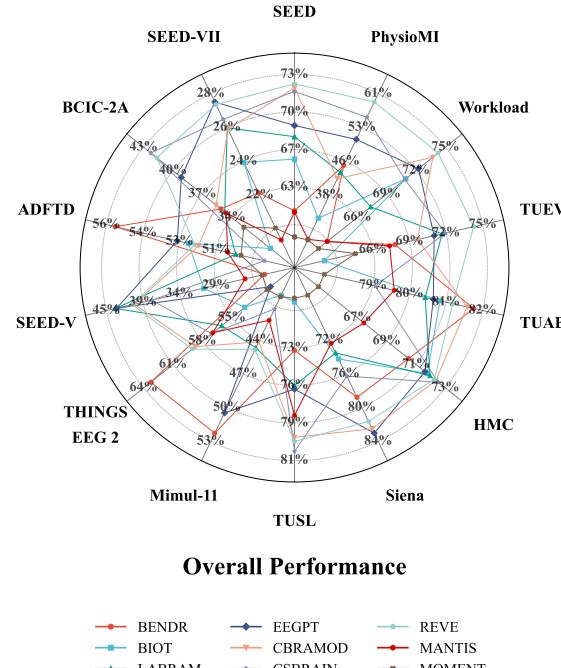

*Figure 10.* Performance comparison among classifier heads.

gradient correlations across most datasets. Furthermore, Fig. 8 (right) reveals a high probability of gradient conflict (blue blocks). The high variance and misalignment imply that reconstruction optimization is disjoint or conflicting with semantic classification. Current pre-training functions primarily as a "time-series aware initialization" rather than effective knowledge transfer. Given EEG's low signal-to-noise ratio, scaling reconstruction objectives alone may yield diminishing downstream returns when objective alignment remains weak; thus, developing objectives that capture discriminative task-relevant features is critical.

### 4.4. Impact of Fine-tuning Strategies

We compare three adaptation strategies, Frozen, LoRA, and Full-parameter, across four EEG-FMs (Fig. 9). First, Full-parameter fine-tuning (green) consistently yields the highest performance in most of datasets, establishing the upper bound for adaptation. Second, the Frozen backbone (red) suffers from a severe **generalization gap**; the sharp drop indicates fixed representations lack sufficient discriminability for direct transfer. Third, LoRA (blue) offers an efficient alternative but performance varies by base model. We observe that LoRA performs competitively in models where pre-training demonstrates a significant advantage over scratch (*e.g.*, CBraMod and REVE), whereas it struggles in weaker baselines. This implies parameter-efficient tuning relies heavily on the backbone's quality.

### 4.5. Impact of Classifier Head

We examine the decoder head by comparing three configurations: Average Pooling, Large Compressing MLP, and Attention Pooling (Fig. 10). The comparison reveals a

*Figure 11.* Holistic performance comparison of seven EEG-FMs and two Time-FMs across 14 datasets. Best viewed via zoom-in.

capacity-stability trade-off. **Average Pooling** (red) delivers the most balanced performance; its simplicity acts as regularization, mitigating overfitting to noisy EEG data. Conversely, high-capacity decoders like **Attention Pooling** and **MLP** show higher variance and a tendency to overfit on simpler tasks. However, the **Large Compressing MLP** (blue) excels in Motor Imagery tasks (*e.g.*, PhysioMI, BCIC-2A). This suggests preserving high-dimensional embeddings captures fine-grained signal fluctuations and temporal dynamics smoothed by global pooling. Therefore, richer heads should be treated as task-specific tools rather than universal improvements. Future research should prioritize refining the foundation model and adaptation interface so performance depends less on high-capacity task-specific decoders.

### 4.6. Overall Benchmarking and Analysis

We synthesize the performance of seven EEG-FMs and two general time-series models in Fig. 11 and Fig. 1. The comparative analysis yields a more nuanced picture of the relationship between scale, inductive bias, and transfer performance. REVE achieves the strongest average balanced accuracy under the matched full-parameter multi-task protocol, which is consistent with a benefit from large-scale pretraining, although this comparison also reflects differences in objectives, architectures, and checkpoint compatibility. However, this advantage is not uniform across datasets: REVE does not dominate every downstream task (*e.g.*, ADFTD, THINGS-EEG2, Mimul-11), and compact EEG-specific models such as EEGPT, CSBrain, and CBraMod

still achieve competitive results with substantially fewer parameters and much smaller pre-training corpora.

Diagnose analysis indicate that nominal model size and pre-training data scale alone are insufficient to explain downstream transfer, and that **architectural design** (*e.g.*, EEGPT's cascade structure or CBraMod's dual-branch design), **objective alignment**, and **task compatibility** are plausible contributors to the observed transfer differences. Moreover, REVE already leverages a large collection of Open-Neuro (Markiewicz et al., 2021) datasets for pre-training, which highlights a practical bottleneck for further scaling that the existing stock of publicly available EEG data may not be sufficient to support another order-of-magnitude increase in foundation-model pre-training demand. Thus, while scaling remains useful, future progress in EEG-FMs should not rely solely on larger corpora or parameter counts, but should also prioritize more data-efficient objectives, better adaptation interfaces, and EEG-specific model design. Furthermore, under our adaptation protocol, specialized EEG-FMs often outperform the two general time-series baselines, suggesting that domain-specific preprocessing and model design remain beneficial for EEG transfer.

The detailed results in Appx. C further show that the overall comparison should be interpreted as protocol-conditioned rather than as a protocol-independent model ranking. Model-dataset gaps change when switching from single-task to multi-task fine-tuning (Tabs. 2 and 3), and the apparent advantage of a checkpoint also depends on the allowed adaptation capacity, as shown by the frozen-backbone, LoRA, full-parameter, and from-scratch settings (Tabs. 4, 5, 3, and 6). Decoder capacity is another non-negligible factor: the average-pooling, attention-pooling, and large-MLP variants lead to different task-level trade-offs (Tabs. 3, 7, and 8). Finally, the general time-series baselines provide a reference point for separating EEG-specific gains from generic temporal modeling effects (Tab. 9).

## 5. Conclusion and Future Perspectives

In this work, we introduce EEG-FM-Bench, a standardized benchmark for EEG-FMs, establishing baselines for seven EEG-FMs and two general time-series models across 14 datasets and 10 paradigms. We identify a significant generalization gap in frozen-backbone settings, indicating that current reconstruction objectives at pretraining stage do not yield sufficiently discriminative representations for downstream tasks. Our results show that nominal scale alone does not reliably explain downstream transfer under released checkpoints and a matched protocol. REVE achieves the strongest average performance, while compact EEG-specific architectures remain competitive. Together with diagnostic analyses, these results highlight objective alignment, transfer compatibility, and EEG-specific induc-

tive bias as key factors for EEG-FM performance beyond scaling. Multi-task learning often mitigates overfitting and improves robustness, but can induce negative transfer under certain paradigms, motivating future work on task grouping and conflict-aware optimization.

Based on these findings, we propose three directions for future research. First, **rethinking pre-training objectives** is necessary to resolve the gradient conflict between reconstruction and downstream classification, moving towards methods that capture discriminative semantic features. Second, developing **neuro-informed architectures** that incorporate neurophysiological constraints and brain-connectivity priors is a promising direction for learning more generalizable representations. Finally, **multi-task learning** remain promising for improving data utilization and integrating complementary signals, but future work should consider task compatibility, adaptive task weighting, and conflict-aware optimization to avoid negative transfer.

## 6. Limitations and Maintenance

The current benchmark focuses on several public datasets, released checkpoints, and classification-oriented downstream tasks, and therefore does not fully cover more EEG datasets and settings, closed-source clinical systems, or deployment-specific constraints. Moreover, our evaluation protocol is designed to improve comparability, but it also introduces unavoidable model-specific adaptation choices, such as checkpoint-compatible preprocessing and channel handling. In addition, the current implementation uses a shared optimization strategy across tasks. This may be suboptimal when datasets differ substantially in signal quality, montage configuration or optimization objectives. Finally, EEG foundation models are evolving rapidly in architecture, pre-training objectives, data scale, and adaptation interfaces, making any static benchmark a moving target. We view EEG-FM-Bench as a maintainable infrastructure and future updates should continuously integrate new datasets, models, protocols, and diagnostic tools.

## Acknowledgments

This work is supported by the Discipline Breakthrough Pilot Project of the Ministry of Education of the People's Republic of China (No. JYB2025XDXM122) and National Natural Science Foundation of China (No. 62472319).

## Impact Statement

This paper presents work whose goal is to advance the field of machine learning. There are many potential societal consequences of our work, none of which we feel must be specifically highlighted here.

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

This appendix provides additional implementation details, analyses, and experiments to support our main findings. The contents are organized as follows:

- Appx. A and B provides detailed descriptions of the datasets used for fine-tuning, along with our complete experimental settings.

- Appx. C presents the detailed experimental data of all 9 models on the 14 benchmark datasets that are not included in the main text due to space limitations.

- Appx. D and E contains further analysis on the training dynamics and visualizations of the EEG classification behavior.

- Appx. F concludes with a discussion of the limitations of the current work and outlines promising directions for future research.

## A. Datasets Description

Detailed information for the evaluation datasets is in Tab. 1:

1. Seizure Detection: Siena (Detti et al., 2020) (binary classification with seizure or healthy);

2. Emotion Recognition: SEED (Zheng & Lu, 2015) (3-class classification with sad, neutral or happy), SEED-V (Liu et al., 2022) (5-class classification with disgust, fear, sad, neutral or happy), SEED-VII (Jiang et al., 2024a) (7-class classification with disgust, fear, sad, neutral, happy, anger or surprise);

3. Motor Imagery: PhysioMI (Schalk et al., 2004) (4-class classification with left fist, right fist, both fists or feet), Mimul-11 (Jeong et al., 2020) (3-class classification with reaching, grasping or twisting), BCIC-2a (Rashid et al., 2020) (4-class classification with left hand, right hand, feet or tongue;

4. Mental Stress: Workload (Zyma et al., 2019) (binary-class classification with arithmetic calculation or resting);

5. Sleep Staging: HMC (Alvarez-Estevez & Rijsman, 2021) (5-class classification with wake, REM, N1, N2 or N3);

6. Anomalous Event Detection: TUEV (Harati et al., 2015) (6-class classification with spike and slow wave, generalized periodic epileptiform discharge, periodic lateralized epileptiform dischage, eye movement artifact or background);

7. Abnormal Classification: TUAB (Lopez et al., 2015) (binary-class classification with abnormal or normal);

8. Visual Target Detection: Things-EEG-2 (Gifford et al., 2022) (binary-class classification with target or non-target);

9. Alzheimer's Disease Identification: ADFTD (Miltiadous et al., 2023) (3-class classification with Alzheimer's Disease, Frontotemporal Dementia or healthy);

10. Slowing Event Classification: TUSL (von Weltin et al., 2017) (3-class classification with seizure, slow wave or background).

## B. Experimental Setup

### B.1. Data pre-processing

Our dataset pre-processing pipeline follows a systematic procedure to process EEG data from multiple sources, implemented using MNE-Python (Gramfort et al., 2013). The process begins by resampling the data to a uniform sampling rate based on the pretraining setting of each model, which allows for consistent patch division. To eliminate low-frequency noise, we apply a high-pass Finite Impulse Response (FIR) filter that uses an overlap-add method, a technique chosen for its effectiveness on signals of variable or short durations. Moreover, power-line interference is suppressed using a notch filter at either 50 Hz or 60 Hz; the specific frequency is chosen by manually inspecting each dataset's power spectrum or its geographical origin. Prior to model input, channels not supported by the model are discarded. The remaining channels from each dataset are then mapped to the standard 10-10 electrode montage based on their names. The data units are then converted from micro-volts ($\mu$V) to Volts to align with the MNE-Python standard. For optimized data management, the

| Dataset | Category | #Channel | Duration | #Train | #Valid | #Test | Task |
|---------|----------|----------|----------|--------|--------|-------|------|
| TUAB | Abnormal Classification | 23 | 30 | 247728 | 12315 | 12277 | Binary Classification |
| TUEV | Anomalous Event Detection | 21 | 5 | 87834 | 12473 | 13046 | 6-class Classification |
| TUSL | Slowing Event Classification | 21,22 | 10 | 210 | 43 | 37 | 3-class Classification |
| SEED | Emotion Recognition | 60 | 10 | 22455 | 7875 | 7560 | 3-class Classification |
| SEED-V | Emotion Recognition | 60 | 10 | 3552 | 4638 | 4128 | 5-class Classification |
| SEED-VII | Emotion Recognition | 60 | 15 | 15536 | 1942 | 1942 | 7-class Classification |
| HMC | Sleep Staging | 4 | 30 | 91681 | 22804 | 22440 | 5-class Classification |
| Workload | Mental Stress | 19 | 10 | 1537 | 300 | 297 | Binary Classification |
| Siena | Seizure Detection | 29 | 10 | 41631 | 5592 | 3607 | Binary Classification |
| Mimul-11 | Motor Imagery | 60 | 5 | 31398 | 5000 | 4949 | 3-class Classification |
| PhysioMI | Motor Imagery | 64 | 4 | 6210 | 1734 | 1803 | 4-class Classification |
| BCIC-2a | Motor Imagery | 22 | 4 | 2784 | 1152 | 1152 | 4-class Classification |
| Things-EEG-2 | Visual Target Detection | 63 | 5 | 24915 | 8324 | 8331 | 2-class Classification |
| ADFTD | Alzheimer's Disease Identification | 19 | 10 | 4743 | 1115 | 1155 | 3-class Classification |

*Table 1.* Detailed information about evaluation datasets.

processed EEG signals are serialized into either the `Parquet` format with Zstandard compression for storage efficiency or the `Arrow` format to accelerate data loading and computation. The pipeline also supports large-scale distributed training by accessing datasets directly from remote storage via the S3 protocol. The entire implementation is built on parallel processing to efficiently handle the terabytes of data in the pre-training corpus.

## B.2. Comparing Foundation Models

Our analysis contrasts seven state-of-the-art EEG Foundation Models: BENDR (Kostas et al., 2021), BIOT (Yang et al., 2023), LaBraM (Jiang et al., 2024b), EEGPT (Wang et al., 2024), CBraMod (Wang et al., 2025a), CSBrain (Zhou et al., 2026) and REVE (El Ouahidi et al., 2026). For comparison, we also incorporated two general temporal models Mantis (Feofanov et al., 2025) and Moment (Goswami et al., 2024). BENDR (Kostas et al., 2021) employs a BERT-style objective on a large clinical EEG corpus. To manage data heterogeneity, BIOT (Yang et al., 2023) tokenizes different biosignals into a single, sentence-like format. LaBraM undergoes pre-training on 2,500 hours of data with VQ-VAE (Van Den Oord et al., 2017) modules for dual-domain (frequency/phase) mask learning. EEGPT (Wang et al., 2024) merges dual self-supervised learning with stabilization mechanisms and supports multi-task evaluation by applying different linear probes to its frozen pre-trained backbone. CBraMod (Wang et al., 2025a) uses criss-cross attention to capture spatial and temporal features separately in the same transformer layer, which refines its representational ability. CSBrain (Zhou et al., 2026) employs cross-scale spatiotemporal tokenization and structured sparse attention to obtain robust EEG representations. REVE (El Ouahidi et al., 2026) is characterized by the use of the largest pre-training dataset available and 4D Fourier positional encoding, which is paired with an improved MAE reconstruction objective. Mantis (Feofanov et al., 2025) and Moment (Goswami et al., 2024) are recently proposed, well-performing general-purpose pre-trained models for time series. We include these models in our benchmark as their parameter counts are comparable to current EEG foundation models. Additionally, their high-quality open-source implementations facilitate fair reproduction. In our experiments, we reproduce these models by the benchmark framework on 14 downstream tasks.

## B.3. Evaluation Partition

For most datasets, we partition the data using a subject-level split. In line with common practice, a subject-dependent strategy is adopted for the SEED, SEED-V, SEED-VII datasets to ensure that our metrics are comparable with other baselines. Moreover, we use a greedy, multi-label stratified splitting algorithm to maintain balanced label distributions across the training, validation, and test partitions according to predefined ratios:

- **Siena**: Subject 0-7, 9-13, 16-17 are assigned to the training, validation, test set correspondingly;

- **SEED**: Following prior research, the 15 trials are divided into three sets in a 9:3:3 ratio, and all sessions are merged together thereafter;

- **SEED-V**: Following prior research, the 15 trials are divided into three sets in a 1:1:1 ratio;

- **SEED-VII**: Subjects are randomly split into training, validation, and test sets at a ratio of 8:1:1;

- **PhysioMI**: Subject 1-69,70-88, 89-110 are assigned to the training set, valid set, test set correspondingly;

- **Mimul-11**: Stratified splitting is employed to achieve approximate ratios of 0.76, 0.12, and 0.12 for training, validating, and testing;

- **BCIC-2a**: Subject 1-5,6-7, 8-9 are assigned to the training set, valid set, test set correspondingly;

- **Workload**: Stratified splitting is employed to achieve approximate ratios of 0.72, 0.14, and 0.14 for training, validating, and testing;

- **HMC**: Subjects are randomly split into training, validation, and test sets at a ratio of 103:24:24;

- **TUEV** and **TUSL**: Owing to the highly imbalanced label distribution in these datasets, the stratified splitting function is employed to create three splits from all the data, approximately aligning with predefined ratios of 0.8, 0.1, and 0.1;

- **TUAB**: The validation and test sets are obtained by equally splitting the original evaluation set by subject, while the training set remains unchanged;

- **Things-EEG-2**: Split by subject with predefined ratios of 0.6, 0.2, and 0.2;

- **ADFTD**: Stratified splitting is employed to achieve approximate ratios of 0.70, 0.15, and 0.15 for training, validating, and testing.

### B.4. Evaluation Metrics

To address the class imbalance in downstream datasets, the following evaluation metrics are adopted for comparison:

- **Balanced Accuracy**: the arithmetic mean of recall (sensitivity) across all classes, mitigating the impact of imbalanced class distributions. It is effective for evaluating classification models on datasets with significant disparities in class proportions, which is formulated as:

$$\text{B-Acc} = \frac{1}{C}\sum_{i=1}^{C}\frac{TP_i}{TP_i + FN_i}, \tag{6}$$

where $C$ is the number of classes, $TP_i$ and $FN_i$ denote true positives and false negatives for class $i$.

- **Weighted F1**: a harmonic mean of precision and recall, weighted by the number of true instances in each class. This metric accounts for class imbalance by assigning higher importance to classes with larger sample sizes, ensuring a more representative evaluation of model effectiveness, which can be formulated as

$$\text{Pre}_i = \frac{TP_i}{TP_i + FP_i}, \quad \text{Rec}_i = \frac{TP_i}{TP_i + FN_i}, \tag{7}$$

$$\text{W-F1} = \sum_{i=1}^{C} w_i \cdot \frac{2 \cdot \text{Pre}_i \cdot \text{Rec}_i}{\text{Pre}_i + \text{Rec}_i}, \tag{8}$$

where $FP_i$ denotes false positives for class $i$, and $w_i$ is the weight of class $i$ based on its support.

- **AUROC**: area under the ROC curve. It reflects the model's ability to discriminate between classes across all possible decision boundaries, which is formulated as

$$\text{TPR} = \frac{TP}{TP + FN}, \quad \text{FPR} = \frac{FP}{FP + TN}, \tag{9}$$

$$\text{AUROC} = \int_0^1 \text{TPR}(f)\,df, \quad f = \text{FPR}. \tag{10}$$

- **AUC-PR**: area under the precision-recall curve. It provides a holistic evaluation of model performance under class imbalance, which can be formulated as

$$\text{AUC-PR} = \int_0^1 \text{Pre}(r)\,dr, \quad r = \text{Rec.} \tag{11}$$

- **Cohen's Kappa**: the agreement level between predicted and true labels by comparing observed and expected frequencies along the diagonal of a confusion matrix. It is particularly suited for multi-class classification scenarios, which can be formulated as

$$\kappa = \frac{p_o - p_e}{1 - p_e}, \tag{12}$$

where $p_o$ is the observed agreement and $p_e$ is the expected agreement.

Among these metrics, AUROC and AUC-PR are used to evaluate binary classification tasks, while Cohen's Kappa and Weighted F1 are applied to multi-category classification. Together, these metrics provide a robust evaluation framework under class imbalance.

## B.5. Training Settings

To facilitate data loading, all samples in the datasets are transformed into an `Arrow` dataset after pre-processing, thus speeding up distributed computing and leveraging the GPU's direct data access functionality. All experiments are conducted using Python 3.11.13, PyTorch 2.7.1, and CUDA 12.8 on one A800 GPU. We enable autonomous mixed precision in the bfloat16 data type to improve GPU memory utilization and introduce `GradScaler` to prevent gradient explosion. We employ the AdamW and a two-phase learning rate scheduler, which combines linear warm-up with cosine annealing. The warm-up learning rate factor is set to 0.5. The total training duration is 30 epochs, including 3 warm-up epochs. The gradient clipping value is 1.0. For EEGPT, we adopt the OneCycle scheduler according to the original paper. To ensure consistency, we adopt the procedure from the REVE paper and apply a freezing warmup epoch for optimal classification head initialization. Weight decay is set to 0.01 except CBraMod for 0.05 to enhance the regularization. Batch size is 128. During downstream fine-tuning, we first set the maximum learning rate following the corresponding open-source implementations. On this basis, we performed a grid search over the learning rate and the hidden dimension of the average-pooling classification head. All experiments were repeated across 5 different random seeds for reliability. We also adopted a differential-learning-rate strategy, configuring the backbone's learning rate to a fraction (e.g., one-tenth, one-fifth, or one-half) of that assigned to the classifier. The final model for each task is selected based on its validation set performance, with results reported on the held-out test set.

## B.6. Model Configurations

Because several pre-trained architectures were optimized for training with additional information or fixed input montage, they cannot directly accommodate the default setting used in our pipeline. We therefore applied model-specific adaptation strategies:

- **BENDR** and **BIOT**: A dynamic-routing convolutional block was inserted ahead of the backbone. This block employs several $Conv1dwithConstraint$ layers that project the input recordings onto the channel configuration expected by the pre-trained weights, thereby harmonizing mismatched channel counts.

- **EEGPT**: As EEGPT already covers almost all electrodes in the 'EasyCap-M1' montage provided by MNE, performance loss due to the slightly sparser 10-10 layout is negligible. We thus implemented an Adapter that simply removes the electrodes unsupported by the model.

- **LaBraM** and **CBraMod**: For these models, the original spatial channel layout was flattened into a one-dimensional ordering compatible with their input format.

- **CSBrain**: We aligned the input of all benchmark datasets with the CSBrain's specification by reordering channels according to their brain region labels, as per its open-source implementation.

• **REVE** relies on a fixed, predefined 4D positional encoding scheme. We load the official released positional encoding checkpoint in the adapter.

EEG-FM-Bench preserves checkpoint compatibility through model-specific preprocessing and adapters. Concretely, BENDR follows the original per-sequence min-max scaling convention, where each EEG sequence is linearly scaled to $[-1, 1]$, BIOT keeps its 95th-percentile-based amplitude normalization; EEGPT follows its mV-scale convention; LaBraM, CBraMod, and CSBrain use the 100 $\mu$V-style scaling in their released implementations; and REVE is evaluated with the closest checkpoint-compatible preprocessing variant exposed by the released implementation. For REVE, we use a window-level z-score approximation rather than exact session-level statistics to avoid leakage from test-session statistics. These adaptations allow each model to ingest the full breadth of our dataset while preserving compatibility with their pre-trained parameters.

## C. Detailed Results

This section presents the complete quantitative results for the 14 datasets not included in the main text due to space limitations. We analyze performance variations across different fine-tuning strategies (Full-Parameter, Frozen, LoRA), task setups (Single-Task vs. Multi-Task), and classifier head architectures. Tab. 2- 9 compares all 9 models across the benchmark tasks under different fine-tuning strategies. Specifically:

• Tab. 2 reports **full-parameter single-task** fine-tuning results;

• Tab. 3 reports **full-parameter multi-task** fine-tuning results;

• Tab. 4 reports **frozen-backbone multi-task** results;

• Tab. 5 reports **LoRA multi-task** fine-tuning results;

• Tab. 6 **multi-task training from scratch** (randomly initialized backbone) to isolate the benefit of pre-training.

• Tab. 7 and Tab. 8 study the role of **classifier head design** under multi-task fine-tuning;

• Tab. 9 reports the performance of two **general time-series foundation models** (adapted to EEG inputs) to contextualize the advantage of EEG-specific inductive biases.

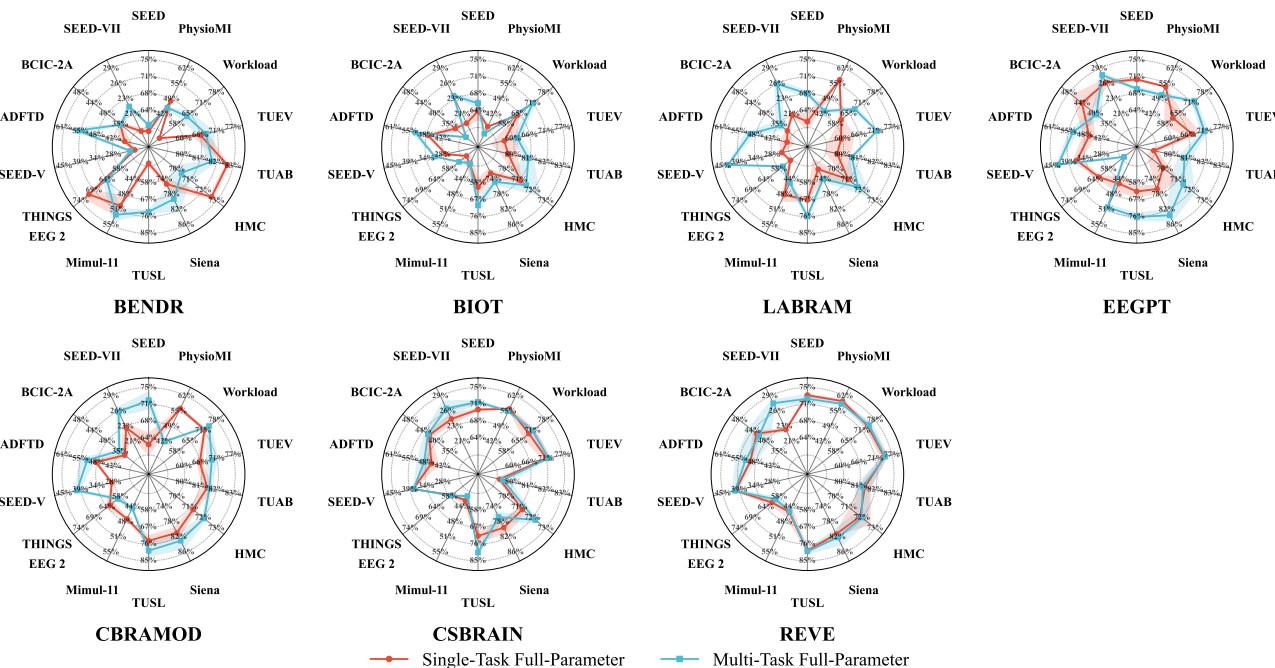

*Figure 12.* Performance comparison between single-task and multi-task fine-tuning for all 7 EEG-FMs.

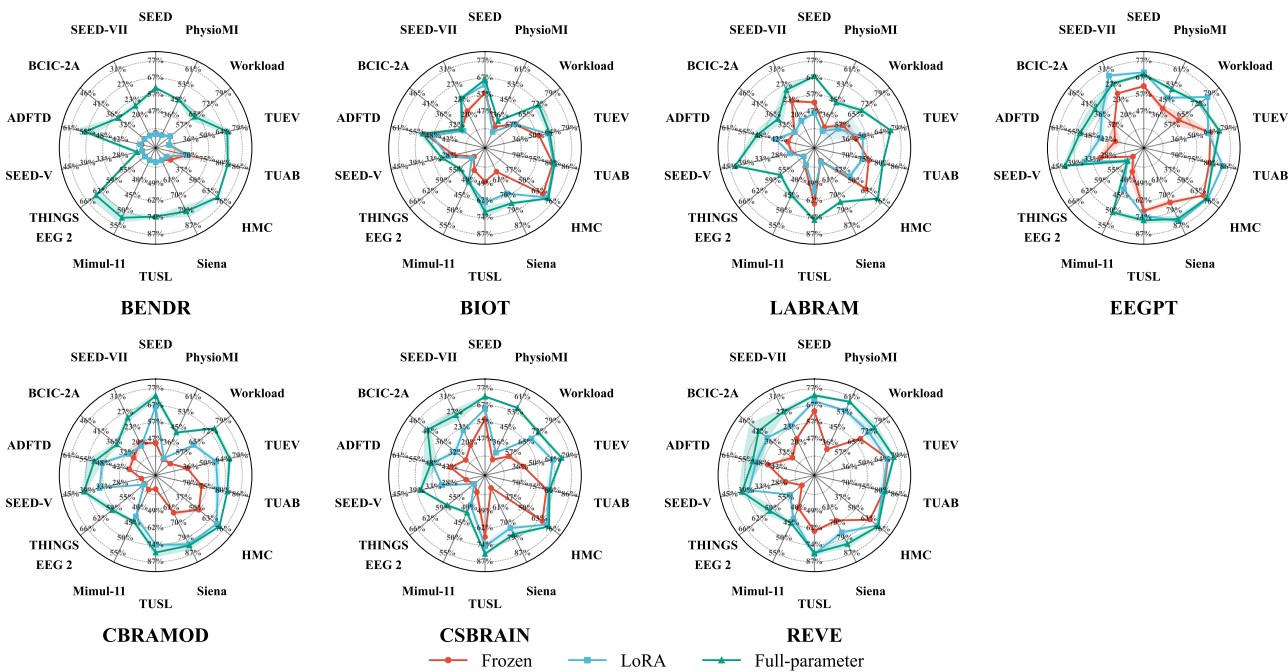

*Figure 13.* Performance comparison among fine-tuning strategies for all 7 EEG-FMs.

## C.1. Overlap-sensitive cases

Some downstream datasets, such as SEED and PhysioMI, are widely used public EEG benchmarks and may overlap with the pre-training corpora of some released EEG-FMs. We retain these datasets for comparability with the existing literature, but we explicitly treat them as overlap-sensitive. Related results are marked in detailed tables and excluded from calculating overall performance for LaBraM and EEGPT.

## C.2. Key patterns across fine-tuning strategies

**Single-task vs. multi-task.** Comparing Tab. 2 (single-task) and Tab. 3 (multi-task), multi-task learning often acts as a regularizer that mitigates overfitting and improves robustness on several datasets, but can also introduce negative transfer for certain paradigms. This motivates the diagnostic analyses in Appx. D and highlights the need for better multi-task optimization (e.g., task balancing and conflict-aware updates).

**Frozen backbone vs. LoRA adaptation.** Tab. 4 shows that freezing the backbone can lead to a generalization gap, suggesting that pre-trained representations are not always directly usable without adaptation. Tab. 5 demonstrates that LoRA provides a parameter-efficient alternative that can close part of this gap on many tasks, while still being limited by task conflicts and representation mismatch in hard-transfer settings.

**Fine-tune from checkpoint vs. training from scratch.** Tab. 6 shows that while from-scratch training generally lags behind pre-trained models, the gap is often narrow, and in some cases it even achieves comparable or slightly better results. This suggests that the current pre-training paradigm may not be fully efficient, as the architectural capacity itself can approach similar performance without extensive pre-training.

## C.3. Classifier head ablations

Tab. 7, 8 show that stronger heads do not uniformly improve performance across paradigms. While high-capacity heads can help in tasks where discriminative evidence is sparse or distributed, they may also exacerbate overfitting when the backbone representation is misaligned with downstream supervision.

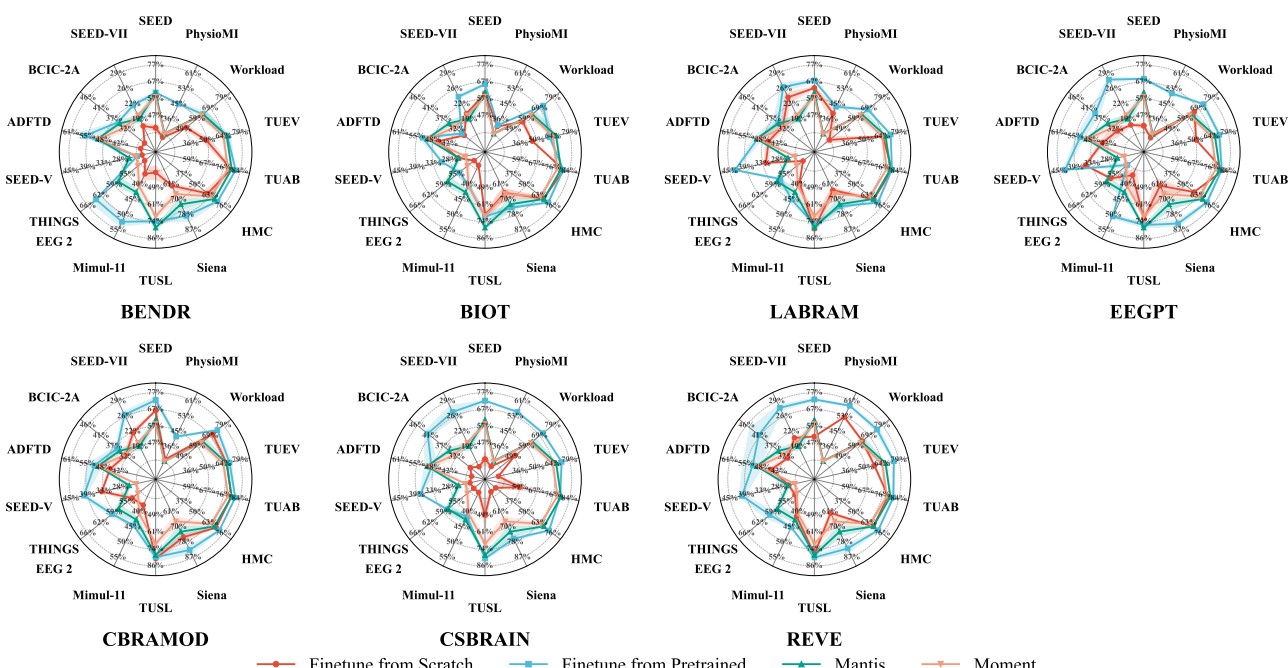

*Figure 14.* Performance comparison between fine-tuning from scratch and from pre-trained model for all 7 EEG-FMs.

### C.4. General time-series foundation models

Tab. 9 compares EEG-FMs against general time-series foundation models. The performance gap in most EEG paradigms suggests that EEG-specific design choices (e.g., channel handling, spatial–temporal tokenization, and objective alignment) remain important for reliable transfer.

## D. Extended Training Dynamics Analysis

### D.1. CBraMod

The results in Fig. 15 and 16 suggest that multi-task training leads to measurable improvements in CBraMod's representation quality, reflected in higher cosine similarity and subspace affinity. Fig. 18 illustrates that CbraMod's pre-training objective and fine-tuning objective exhibit a significant conflict in their gradient dynamics, and they primarily affect different model layers. Fig. 19 reveals an interesting divergence: while the intermediate representations from scratch training can rapidly align with the fine-tuned pre-trained model, the underlying gradient energy flow follows a fundamentally different trajectory.

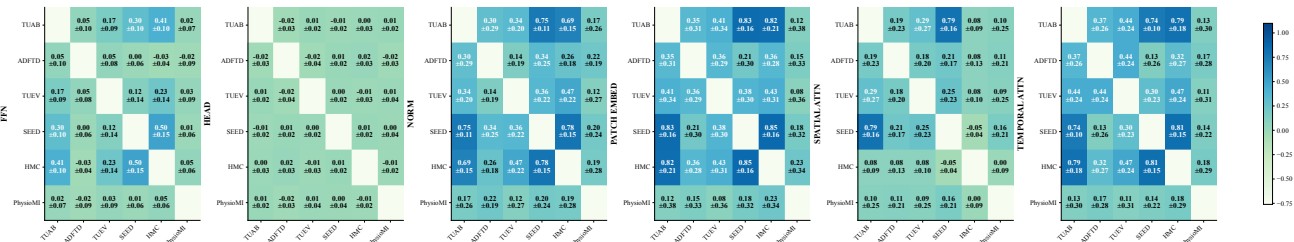

*Figure 15.* Cross-task gradient cosine similarity correlation analysis for CBraMod.

### D.2. CSBrain

For CSBrain, Fig. 20, 21 provide the cross-task gradient correlation and subspace affinity analyses, respectively. The results in these two figures indicate that the core architecture of CSBrain struggles to achieve performance gains through

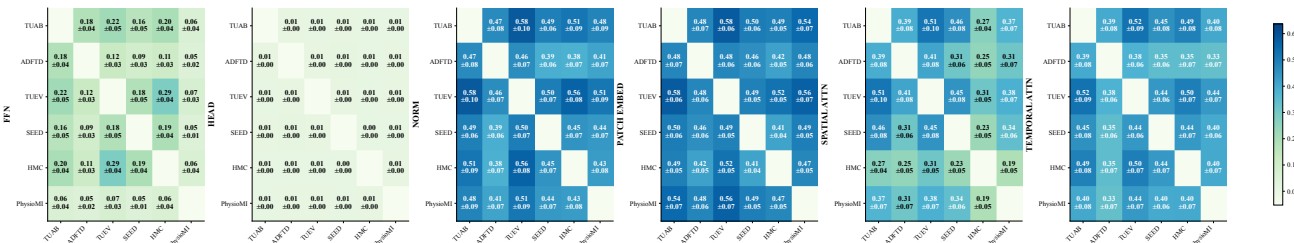

*Figure 16.* Cross-task subspace affinity analysis for CBraMod.

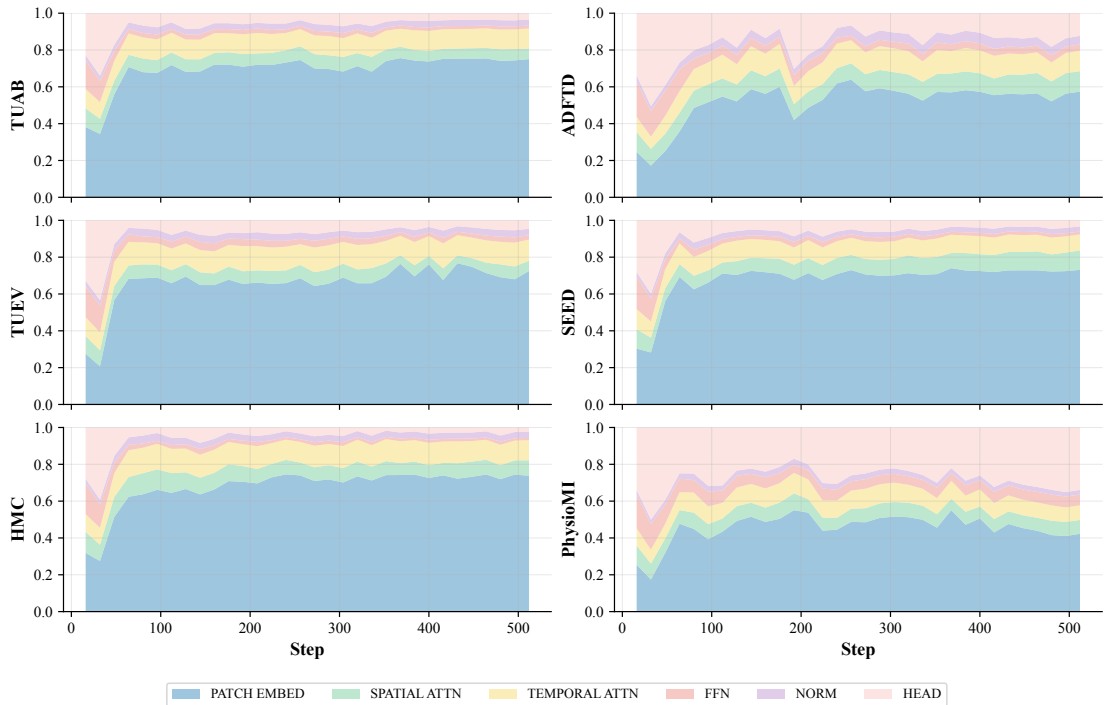

*Figure 17.* Cross-task evolution dynamics analysis for CBraMod.

multi-task training. We hypothesize that the multi-scale modules and brain region attention in CSBrain's backbone design have difficulty adapting to the rapidly shifting data characteristics during fine-tuning.

Fig. 23 analyzes the gradient alignment between pre-training and downstream tasks. It reveals that CSBrain struggles to effectively optimize its embedding module parameters in both scenarios, and does so in divergent directions, as evidenced by low cosine similarity and high conflict frequency. Fig. 24 indicates that the shallow layers in the CSBrain backbone exhibit markedly different behaviors between the randomly initialized and pre-trained models. Furthermore, modules such as the region attention and feed-forward networks appear particularly resistant to effective optimization if training starts from scratch. Fig. 25 shows that the randomly initialized CSBrain model exhibits abnormally large gradient norms and loss values. Although both metrics drop rapidly during training, we observed in subsequent phases that the model remains prone to collapse when fine-tuned from scratch.

## E. Visualization

To complement the quantitative results, this section provides a qualitative analysis of the feature representations learned by each foundation model. We use t-SNE (Maaten & Hinton, 2008) to visualize the feature embeddings and Integrated Gradients (Sundararajan et al., 2017) to infer the neurophysiological basis for model decisions, focusing on the results from the full-parameter multi-task fine-tuning strategy. Additional visualizations are presented for TUEV, SEED-VII, Mimul-11, ADFTD, and HMC datasets.

- **TUEV**: This 6-class classification task involves identifying various epileptiform discharges and artifacts. The t-SNE plots (Fig. 26) show that CBraMod, EEGPT and CSBrain produce more coherent clusters, aligning with their superior B-Acc scores in the main text. In contrast, the embeddings from BENDR and BIOT are heavily overlapped or same category fail to cluster together, reflecting their lower quantitative performance. The Integrated Gradients maps for the top models indicate a focus on temporal and central channels, which is consistent with the typical scalp distribution of spike-wave discharges.

- **SEED-V**: This 5-class emotion task is notoriously difficult. The visualizations reveal that all models struggle to form well-separated clusters, which corresponds to the low B-Acc scores across the board (Fig. 27). Nonetheless, EEGPT and LaBraM show emergent structures in their embeddings that are absent in other models. The saliency maps suggest that these models attend to channels over the prefrontal and temporal cortices, regions known to be involved in emotion processing, reaffirming the findings from the main text.

- **Mimul-11**: For this 3-class motor imagery task, the t-SNE (Fig. 28) plots for EEGPT and BENDR show the clearest class separation, which is reflected in their higher B-Acc scores relative to other models in the multi-task setting. The Integrated Gradients visualizations highlight activity in the central and parietal areas of the scalp, corresponding to the location of the motor and somatosensory cortices that govern upper limb movements.

- **ADFTD**: This task requires discriminating between Alzheimer's Disease (AD), Frontotemporal Dementia (FTD), and healthy controls (CN). The t-SNE (Fig. 29) plots show that BENDR and EEGPT, which performed surprisingly well on this task in the multi-task setting, produces reasonably distinct clusters for the AD and CN classes, though the FTD class remains mixed for BENDR. The saliency maps for the better-performing models indicate a focus on frontal, temporal and parietal channels, which aligns with the known progression of cortical atrophy in Alzheimer's Disease.

- **HMC**: For this 5-class sleep staging task, the t-SNE visualizations (Fig. 30) show that the clusters of EEGPT appear highly fragmented, while CBraMod's visualization exhibits a radial pattern where the central N2 category overlaps excessively with other categories, which indicates the suboptimal metrics in the table of main text.

## F. Limitations and Summary

### F.1. Limitations

**Scope of Benchmark**    While EEG-FM-Bench covers a diverse set of datasets and paradigms, it cannot represent the full spectrum of EEG applications (e.g., multi-modal EEG settings, closed-source proprietary models, and certain clinical workflows). Future iterations can expand the suite of tasks and continuously integrate new state-of-the-art models to ensure the benchmark remains relevant.

**Negative transfer in multi-task learning.**    Multi-task fine-tuning can improve generalization but also risks negative transfer when datasets have conflicting objectives or incompatible inductive biases. Our current implementation uses a unified optimizer and learning strategy for all tasks, which may be suboptimal for managing gradient conflicts. Conflict-aware updates, adaptive task weighting, or modular/shared-private designs are promising directions.

**Rapid evolution of EEG foundation models.**    EEG-FMs are evolving quickly in architecture, pre-training objectives, and data scale. This creates a moving target for fair comparison and emphasizes the need for open, reproducible, and continuously maintainable benchmarking infrastructure.

### F.2. Summary

This appendix provides supplementary details that support the main paper, including:

- Dataset-level descriptions and experimental specifics;
- Detailed training settings;
- Complete quantitative performance tables under multiple fine-tuning strategies and classifier heads;
- Qualitative visualizations and diagnostic analyses that help interpret generalization and optimization behavior

Together with the main paper, these results provide standardized baselines and mechanistic insights to guide future EEG-FM development, with an emphasis on benchmarking unity, pre-training efficiency, and robust multi-task transfer.

*Table 2.* Performance comparison of 7 EEG-FMs on 14 BCI tasks under **full-parameter single-task** fine-tuning with **average pooling** classification head. † indicates overlap between the model's pre-training datasets and our benchmark datasets.

| Dataset | Metrics | BENDR | BIOT | LaBraM | EEGPT | CBraMod | CsBrain | REVE |
|---|---|---|---|---|---|---|---|---|
| **SEED** | B-Acc | 59.50±00.42 | 63.87±01.77 | 61.59±01.71$^{\dagger}$ | 70.83±00.28$^{\dagger}$ | 62.59±02.40 | 70.23±00.25 | 73.37±00.30 |
| | F1/AUROC | 58.88±00.98 | 63.12±01.94 | 60.30±01.60$^{\dagger}$ | 70.48±00.16$^{\dagger}$ | 61.67±02.66 | 70.70±00.22 | 73.32±00.41 |
| | Kappa/AUCPR | 39.42±00.64 | 45.99±02.67 | 43.28±02.48$^{\dagger}$ | 56.41±00.42$^{\dagger}$ | 44.36±03.58 | 55.70±00.28 | 62.77±00.44 |
| **PhysioMI** | B-Acc | 47.78±00.28 | 36.22±00.10 | 57.27±00.26 | 54.16±00.18$^{\dagger}$ | 56.74±00.36 | 56.57±00.45 | 60.03±00.58 |
| | F1/AUROC | 42.72±08.93 | 35.65±00.18 | 57.29±00.25 | 53.27±00.92$^{\dagger}$ | 56.74±00.39 | 56.57±00.53 | 60.08±00.62 |
| | Kappa/AUCPR | 36.48±08.47 | 14.95±00.15 | 43.00±00.35 | 38.04±01.08$^{\dagger}$ | 42.31±00.48 | 42.07±00.59 | 46.65±00.80 |
| **Workload** | B-Acc | 50.00±00.00 | 63.98±00.66 | 61.17±01.01 | 62.31±03.98 | 71.94±00.60 | 69.53±02.65 | 74.12±01.01 |
| | F1/AUROC | 52.36±01.25 | 77.68±01.62 | 67.78±01.48 | 70.46±04.00 | 81.63±01.63 | 77.07±04.26 | 83.47±01.74 |
| | Kappa/AUCPR | 30.70±00.55 | 69.30±02.27 | 45.83±02.21 | 48.13±06.90 | 59.73±01.79 | 64.67±03.20 | 64.43±03.71 |
| **TUEV** | B-Acc | 65.53±02.40 | 57.07±04.29 | 59.58±04.31 | 66.86±02.54 | 65.42±00.90 | 71.20±01.34 | 73.98±01.21 |
| | F1/AUROC | 80.95±02.18 | 74.34±04.00 | 77.72±03.25 | 83.95±00.14 | 79.03±01.36 | 85.07±00.85 | 90.25±01.42 |
| | Kappa/AUCPR | 66.62±04.07 | 39.55±04.15 | 64.06±05.03 | 74.45±00.09 | 64.13±02.48 | 74.67±01.44 | 85.07±01.80 |
| **TUAB** | B-Acc | 82.72±00.12 | 79.52±00.59 | 79.50±00.81 | 78.66±00.24 | 81.40±00.33 | 78.97±00.17 | 81.32±00.35 |
| | F1/AUROC | 89.52±00.14 | 88.27±00.16 | 86.91±00.51 | 87.52±01.01 | 89.02±00.16 | 85.33±01.51 | 87.93±01.76 |
| | Kappa/AUCPR | 90.68±00.13 | 88.45±00.07 | 83.86±01.96 | 86.94±02.15 | 89.93±00.31 | 82.73±04.69 | 87.38±01.56 |
| **HMC** | B-Acc | 72.63±00.13 | 71.01±00.07 | 70.85±00.44 | 69.67±01.24 | 71.14±00.14 | 71.13±00.26 | 71.82±00.82 |
| | F1/AUROC | 73.67±00.53 | 72.27±00.11 | 71.52±00.55 | 73.03±00.32 | 72.76±00.55 | 73.10±00.22 | 75.08±00.74 |
| | Kappa/AUCPR | 65.86±00.70 | 64.40±00.35 | 64.47±00.70 | 64.64±00.92 | 64.86±00.41 | 65.20±00.29 | 66.49±00.83 |
| **Siena** | B-Acc | 74.93±01.76 | 72.05±01.15 | 70.97±02.68 | 76.25±01.02 | 80.64±01.40 | 79.25±01.68 | 80.95±00.32 |
| | F1/AUROC | 90.90±03.60 | 84.23±03.42 | 91.33±02.00 | 92.08±01.15 | 93.86±00.25 | 91.52±02.96 | 91.77±03.26 |
| | Kappa/AUCPR | 99.81±00.08 | 99.72±00.09 | 99.87±00.05 | 99.81±00.02 | 99.87±00.01 | 99.83±00.13 | 99.85±00.07 |
| **TUSL** | B-Acc | 47.22±01.48 | 60.05±04.47 | 66.98±00.44 | 62.41±03.10 | 74.18±02.69 | 71.60±02.63 | 79.58±01.32 |
| | F1/AUROC | 34.17±02.80 | 56.83±04.49 | 62.43±01.73 | 70.63±02.34 | 73.15±02.74 | 72.60±02.69 | 74.58±01.25 |
| | Kappa/AUCPR | 17.88±01.95 | 37.52±06.77 | 45.08±01.57 | 58.92±02.63 | 63.67±03.23 | 61.91±03.85 | 65.07±01.86 |
| **Mimul-11** | B-Acc | 50.44±02.30 | 40.55±01.14 | 47.80±01.68 | 45.19±01.69 | 47.07±00.09 | 42.67±00.77 | 45.41±01.15 |
| | F1/AUROC | 59.01±01.23 | 49.68±01.29 | 57.43±01.27 | 54.66±01.34 | 55.10±00.36 | 51.93±00.87 | 50.63±00.71 |
| | Kappa/AUCPR | 36.10±03.49 | 14.45±02.10 | 30.71±03.14 | 23.68±02.30 | 25.13±00.69 | 18.13±02.05 | 18.93±01.86 |
| **Things EEG 2** | B-Acc | 71.09±02.02 | 51.10±00.30 | 53.38±00.39 | 62.08±00.58 | 62.40±01.56 | 57.20±00.29 | 61.03±00.90 |
| | F1/AUROC | 83.09±00.69 | 62.05±01.53 | 56.38±01.08 | 75.69±01.02 | 77.61±00.32 | 62.67±00.53 | 66.77±00.52 |
| | Kappa/AUCPR | 52.68±00.45 | 18.05±01.06 | 13.65±00.45 | 33.32±01.46 | 37.48±00.78 | 21.53±00.42 | 22.87±01.72 |
| **SEED-V** | B-Acc | 20.20±00.13 | 26.66±00.13 | 24.35±00.55 | 36.43±01.33 | 28.34±00.17 | 38.23±00.24 | 40.65±00.51 |
| | F1/AUROC | 10.72±01.57 | 25.85±00.13 | 23.64±01.21 | 33.00±01.94 | 27.48±00.89 | 39.07±00.34 | 40.35±01.63 |
| | Kappa/AUCPR | 00.21±00.13 | 09.53±00.08 | 05.61±01.13 | 18.52±00.67 | 10.55±00.20 | 22.70±00.29 | 25.82±00.72 |
| **ADFTD** | B-Acc | 37.16±02.62 | 47.43±01.03 | 35.75±02.76 | 46.50±02.28 | 48.47±01.12 | 46.08±02.61 | 50.22±00.41 |
| | F1/AUROC | 40.38±01.69 | 48.17±01.38 | 34.47±03.44 | 48.77±02.56 | 51.68±01.11 | 46.63±03.41 | 50.72±01.85 |
| | Kappa/AUCPR | 07.27±04.67 | 23.77±01.52 | 11.40±03.67 | 21.81±03.72 | 31.17±01.74 | 24.27±03.85 | 29.21±02.08 |
| **BCIC-2a** | B-Acc | 35.21±00.54 | 33.33±00.79 | 32.37±00.21 | 44.07±03.27 | 33.71±00.86 | 42.10±00.87 | 42.73±01.10 |
| | F1/AUROC | 33.25±00.57 | 29.85±00.27 | 30.73±00.39 | 38.61±04.23 | 28.99±01.35 | 40.64±00.62 | 42.60±01.77 |
| | Kappa/AUCPR | 13.62±00.72 | 11.10±01.04 | 09.83±00.30 | 25.42±04.36 | 11.61±01.14 | 22.83±01.13 | 27.08±01.46 |
| **SEED-VII** | B-Acc | 17.43±00.60 | 18.89±00.43 | 20.39±00.41 | 26.42±00.55 | 23.10±00.93 | 24.73±00.79 | 24.73±00.79 |
| | F1/AUROC | 14.20±00.79 | 16.84±01.12 | 19.55±03.82 | 23.89±01.29 | 21.82±02.00 | 22.82±01.51 | 22.82±01.51 |
| | Kappa/AUCPR | 03.98±00.69 | 05.80±00.48 | 09.42±02.64 | 14.05±00.99 | 09.70±02.62 | 12.20±01.14 | 12.20±01.14 |

*Table 3.* Performance comparison of 7 EEG-FMs on 14 BCI tasks under **full-parameter multi-task** fine-tuning with **average pooling** classification head. † indicates overlap between the model's pre-training datasets and our benchmark datasets.

| Dataset | Metrics | BENDR | BIOT | LaBraM | EEGPT | CBraMod | CSBrain | REVE |
|---------|---------|-------|------|--------|-------|---------|---------|------|
| **SEED** | B-Acc | 60.82±00.19 | 65.62±00.92 | 67.71±00.55† | 68.75±00.40† | 72.25±01.90 | 71.92±00.39 | 72.62±00.92 |
| | F1/AUROC | 60.84±00.20 | 65.57±00.81 | 68.11±00.66† | 67.42±00.78† | 72.90±01.63 | 71.84±00.20 | 72.75±00.76 |
| | Kappa/AUCPR | 41.59±00.16 | 50.11±00.99 | 52.22±00.83† | 52.37±00.28† | 58.68±02.91 | 58.02±00.59 | 62.57±00.94 |
| **PhysioMI** | B-Acc | 44.77±00.31 | 33.02±00.29 | 43.19±01.03 | 50.52±00.80† | 41.80±00.69 | 55.43±00.37 | 58.82±00.45 |
| | F1/AUROC | 44.63±00.03 | 32.75±00.36 | 42.38±01.15 | 50.26±00.76† | 40.35±00.87 | 55.10±00.63 | 58.77±00.56 |
| | Kappa/AUCPR | 26.33±00.40 | 10.67±00.40 | 24.25±01.37 | 34.00±01.06† | 22.42±00.92 | 40.48±00.48 | 45.10±00.60 |
| **Workload** | B-Acc | 63.32±02.29 | 71.37±02.19 | 67.77±00.95 | 72.66±00.90 | 74.12±00.79 | 71.27±00.75 | 74.67±01.46 |
| | F1/AUROC | 74.09±01.04 | 77.32±02.88 | 77.32±02.88 | 81.16±03.42 | 83.89±00.93 | 79.63±01.20 | 82.32±01.47 |
| | Kappa/AUCPR | 54.78±01.84 | 69.52±00.23 | 61.13±03.03 | 64.30±06.48 | 72.58±01.55 | 55.10±02.21 | 68.22±03.92 |
| **TUEV** | B-Acc | 67.46±02.59 | 61.35±01.35 | 71.53±00.19 | 70.85±01.16 | 69.41±01.86 | 71.82±00.69 | 74.27±00.87 |
| | F1/AUROC | 84.56±01.82 | 80.55±00.46 | 86.22±00.73 | 86.35±00.47 | 83.44±00.31 | 87.00±00.64 | 89.37±00.33 |
| | Kappa/AUCPR | 72.85±02.87 | 68.37±00.61 | 77.42±01.20 | 77.69±01.06 | 72.02±00.65 | 77.84±01.55 | 82.12±00.76 |
| **TUAB** | B-Acc | 81.72±00.26 | 80.85±00.36 | 80.47±00.12 | 80.68±00.37 | 81.55±00.14 | 79.18±00.48 | 81.18±00.46 |
| | F1/AUROC | 90.38±00.27 | 88.67±00.62 | 88.33±00.59 | 90.10±00.52 | 89.57±00.26 | 86.32±00.28 | 88.32±00.37 |
| | Kappa/AUCPR | 90.22±00.15 | 89.01±00.47 | 88.65±00.28 | 90.35±00.54 | 89.67±00.02 | 85.78±00.25 | 89.33±00.61 |
| **HMC** | B-Acc | 70.21±00.72 | 71.45±01.05 | 71.63±01.04 | 71.30±00.91 | 72.03±00.14 | 72.18±00.24 | 71.93±00.45 |
| | F1/AUROC | 72.44±00.65 | 72.28±01.08 | 72.46±00.96 | 72.46±00.96 | 75.20±00.29 | 74.16±00.27 | 72.07±00.73 |
| | Kappa/AUCPR | 64.57±00.90 | 64.89±00.71 | 65.22±01.23 | 64.86±00.77 | 67.43±00.12 | 66.43±00.16 | 64.40±00.54 |
| **Siena** | B-Acc | 78.96±03.08 | 74.35±00.71 | 73.62±00.62 | 83.28±01.56 | 82.75±02.10 | 76.42±01.43 | 81.87±02.03 |
| | F1/AUROC | 91.31±04.74 | 81.20±00.92 | 89.43±03.59 | 93.44±00.12 | 92.03±01.90 | 92.70±00.67 | 90.70±01.25 |
| | Kappa/AUCPR | 99.81±00.11 | 99.26±00.05 | 99.82±00.07 | 99.86±00.00 | 99.83±00.04 | 99.87±00.05 | 99.88±00.03 |
| **TUSL** | B-Acc | 73.35±00.98 | 69.90±04.10 | 75.86±01.15 | 76.12±02.05 | 79.56±03.24 | 80.56±00.91 | 79.85±01.27 |
| | F1/AUROC | 70.90±01.73 | 71.03±05.19 | 72.74±01.33 | 84.19±01.04 | 76.24±04.45 | 76.72±00.38 | 77.81±01.89 |
| | Kappa/AUCPR | 56.65±02.55 | 56.00±07.50 | 59.60±01.81 | 84.30±01.27 | 64.72±05.96 | 65.57±00.41 | 66.62±02.35 |
| **Mimul-11** | B-Acc | 52.55±01.52 | 40.43±00.63 | 45.05±00.66 | 50.77±00.61 | 44.53±00.19 | 41.64±00.54 | 45.22±00.13 |
| | F1/AUROC | 59.61±02.80 | 49.02±00.73 | 53.42±01.98 | 57.44±01.11 | 52.80±00.57 | 50.70±00.61 | 53.73±00.65 |
| | Kappa/AUCPR | 38.35±04.93 | 14.63±01.88 | 22.60±02.20 | 31.16±02.43 | 19.23±00.77 | 15.94±01.33 | 21.53±01.16 |
| **Things EEG 2** | B-Acc | 63.62±01.38 | 54.15±00.25 | 56.47±00.32 | 51.54±00.76 | 59.13±00.72 | 57.49±00.27 | 59.43±01.20 |
| | F1/AUROC | 77.23±00.88 | 63.01±00.52 | 62.82±00.84 | 66.64±00.04 | 69.58±01.41 | 62.47±01.27 | 69.00±00.53 |
| | Kappa/AUCPR | 42.29±01.75 | 20.12±00.33 | 22.23±00.64 | 20.30±00.65 | 31.85±01.21 | 24.66±01.93 | 32.07±01.85 |
| **SEED-V** | B-Acc | 21.75±00.66 | 30.54±00.18 | 43.26±00.29 | 43.07±00.60 | 40.56±00.77 | 38.02±00.23 | 40.84±00.98 |
| | F1/AUROC | 19.94±02.58 | 30.97±00.26 | 43.20±00.81 | 42.99±00.54 | 41.72±00.72 | 38.07±01.03 | 40.58±02.00 |
| | Kappa/AUCPR | 02.90±00.68 | 14.14±00.21 | 28.93±00.50 | 28.62±00.86 | 26.22±00.97 | 22.48±00.42 | 25.59±01.48 |
| **ADFTD** | B-Acc | 55.71±03.76 | 52.40±01.98 | 50.27±01.74 | 52.91±01.57 | 51.95±02.84 | 48.96±02.65 | 52.12±03.78 |
| | F1/AUROC | 56.97±02.75 | 53.48±02.71 | 52.36±02.23 | 54.82±01.30 | 54.41±03.13 | 50.17±03.76 | 51.10±03.90 |
| | Kappa/AUCPR | 33.98±04.63 | 36.27±03.65 | 26.17±03.15 | 31.14±01.85 | 30.06±04.35 | 24.96±04.13 | 30.70±06.41 |
| **BCIC-2a** | B-Acc | 34.98±00.25 | 29.66±00.39 | 34.58±01.64 | 39.23±01.48 | 35.50±00.58 | 42.50±01.08 | 41.87±03.34 |
| | F1/AUROC | 30.01±02.27 | 23.61±04.64 | 31.24±02.71 | 33.08±01.86 | 23.97±00.25 | 39.97±02.34 | 38.68±02.25 |
| | Kappa/AUCPR | 13.31±00.33 | 05.99±00.45 | 12.60±02.00 | 16.95±01.97 | 14.00±00.77 | 23.32±01.48 | 22.48±03.47 |
| **SEED-VII** | B-Acc | 21.98±00.66 | 23.93±00.39 | 26.13±00.92 | 27.78±00.47 | 26.05±00.72 | 26.67±01.37 | 27.62±00.61 |
| | F1/AUROC | 19.31±00.87 | 22.13±00.52 | 24.74±00.57 | 25.40±00.12 | 25.74±01.22 | 22.68±02.69 | 23.93±01.08 |
| | Kappa/AUCPR | 09.49±01.03 | 11.38±00.38 | 13.07±00.93 | 15.80±01.12 | 13.09±00.86 | 11.40±02.04 | 14.78±00.44 |

*Table 4.* Performance comparison of 7 EEG-FMs on 14 BCI tasks under **freezing-parameter multi-task** fine-tuning with **average pooling** classification head. † indicates overlap between the model's pre-training datasets and our benchmark datasets.

| Dataset | Metrics | BENDR | BIOT | LaBraM | EEGPT | CBraMod | CSBrain | REVE |
|---|---|---|---|---|---|---|---|---|
| **SEED** | B-Acc | 34.12±00.24 | 57.85±00.19 | 52.13±00.05$^\dagger$ | 61.80±00.65$^\dagger$ | 44.32±00.28 | 58.57±00.05 | 63.22±00.13 |
| | F1/AUROC | 24.48±02.82 | 58.17±00.21 | 51.23±00.34$^\dagger$ | 60.41±01.07$^\dagger$ | 44.50±00.30 | 58.53±00.05 | 63.35±00.08 |
| | Kappa/AUCPR | 01.10±00.32 | 37.17±00.29 | 28.33±00.12$^\dagger$ | 42.89±00.97$^\dagger$ | 16.76±00.45 | 38.00±00.14 | 45.20±00.32 |
| **PhysioMI** | B-Acc | 25.22±00.24 | 29.80±00.33 | 29.63±00.34 | 39.90±01.39$^\dagger$ | 26.90±00.24 | 26.80±00.29 | 32.55±00.18 |
| | F1/AUROC | 12.55±02.53 | 29.90±00.43 | 28.97±00.77 | 39.51±01.92$^\dagger$ | 22.76±00.17 | 24.77±01.25 | 32.12±00.40 |
| | Kappa/AUCPR | 00.27±00.31 | 06.40±00.41 | 06.23±00.42 | 19.87±01.84$^\dagger$ | 02.54±00.32 | 02.40±00.37 | 10.08±00.23 |
| **Workload** | B-Acc | 50.00±00.00 | 58.25±00.50 | 57.45±02.83 | 61.05±03.33 | 50.00±00.00 | 55.07±00.94 | 67.10±00.00 |
| | F1/AUROC | 48.83±01.68 | 76.48±00.48 | 73.43±00.56 | 75.68±00.59 | 66.87±00.17 | 76.83±00.12 | 79.83±00.17 |
| | Kappa/AUCPR | 24.83±00.87 | 54.17±00.30 | 46.72±00.74 | 58.23±01.57 | 42.70±00.07 | 54.73±00.12 | 61.20±00.85 |
| **TUEV** | B-Acc | 16.70±00.00 | 52.12±00.55 | 41.27±00.12 | 63.08±00.66 | 32.50±00.04 | 38.63±00.17 | 69.03±00.17 |
| | F1/AUROC | 44.10±00.00 | 80.93±00.20 | 73.37±00.39 | 86.22±00.31 | 65.97±00.20 | 72.87±00.19 | 89.53±00.30 |
| | Kappa/AUCPR | 00.00±00.00 | 68.85±00.28 | 54.97±00.53 | 76.83±00.71 | 41.58±00.49 | 53.40±00.28 | 83.22±00.52 |
| **TUAB** | B-Acc | 68.30±05.85 | 80.30±00.22 | 75.87±00.05 | 79.53±00.05 | 73.15±00.19 | 78.20±00.00 | 80.80±00.32 |
| | F1/AUROC | 73.17±00.82 | 87.50±00.14 | 84.07±00.05 | 88.63±00.12 | 80.41±00.02 | 87.00±00.08 | 88.77±00.41 |
| | Kappa/AUCPR | 65.93±00.45 | 87.27±00.57 | 85.11±00.08 | 88.47±00.05 | 79.79±00.15 | 87.23±00.12 | 88.33±00.55 |
| **HMC** | B-Acc | 24.18±00.02 | 66.13±00.21 | 59.80±00.00 | 67.83±00.12 | 51.81±00.55 | 65.60±00.00 | 63.80±00.08 |
| | F1/AUROC | 27.02±00.02 | 70.27±00.29 | 64.10±00.37 | 71.63±00.05 | 58.11±00.82 | 69.73±00.21 | 65.33±00.09 |
| | Kappa/AUCPR | 07.50±00.05 | 61.03±00.17 | 53.40±00.16 | 62.90±00.08 | 44.95±00.68 | 62.00±00.28 | 56.27±00.05 |
| **Siena** | B-Acc | 50.00±00.00 | 56.42±00.02 | 50.17±00.24 | 73.90±01.28 | 64.17±01.18 | 50.00±00.00 | 68.60±00.08 |
| | F1/AUROC | 61.20±04.16 | 81.20±00.13 | 84.48±01.45 | 97.28±00.43 | 90.16±00.41 | 56.53±12.17 | 77.87±00.37 |
| | Kappa/AUCPR | 99.27±00.12 | 99.70±00.00 | 99.70±00.05 | 99.93±00.02 | 99.68±00.02 | 98.10±00.62 | 99.27±00.05 |
| **TUSL** | B-Acc | 33.30±00.00 | 47.62±00.05 | 63.90±02.52 | 69.13±01.65 | 33.00±00.00 | 68.17±01.04 | 63.88±00.54 |
| | F1/AUROC | 17.50±00.00 | 44.17±00.56 | 54.57±02.18 | 69.27±02.15 | 09.35±01.20 | 62.93±01.79 | 55.37±00.87 |
| | Kappa/AUCPR | 00.00±00.00 | 18.75±00.19 | 37.63±04.29 | 59.70±03.25 | 00.00±00.00 | 47.63±01.93 | 38.93±00.97 |
| **Mimul-11** | B-Acc | 33.30±00.00 | 36.65±00.13 | 35.50±00.43 | 37.11±01.97 | 33.95±00.11 | 34.80±00.59 | 40.20±00.32 |
| | F1/AUROC | 37.90±00.00 | 43.92±00.38 | 42.97±00.88 | 44.95±03.07 | 40.09±00.18 | 42.73±01.05 | 48.02±00.50 |
| | Kappa/AUCPR | 00.00±00.00 | 06.90±00.27 | 04.60±00.93 | 10.06±03.66 | 01.51±00.25 | 03.10±01.39 | 12.38±00.64 |
| **Things EEG 2** | B-Acc | 50.00±00.00 | 50.05±00.00 | 50.00±00.00 | 50.00±00.00 | 50.00±00.00 | 50.00±00.00 | 50.47±00.13 |
| | F1/AUROC | 51.17±01.46 | 53.85±00.56 | 53.77±00.45 | 59.57±00.42 | 53.85±00.07 | 51.63±02.31 | 58.18±00.06 |
| | Kappa/AUCPR | 11.23±00.39 | 12.43±00.33 | 12.57±00.17 | 14.18±00.15 | 12.29±00.13 | 12.10±00.94 | 14.92±00.06 |
| **SEED-V** | B-Acc | 20.00±00.00 | 26.27±00.47 | 23.23±00.48 | 30.48±00.73 | 20.28±00.02 | 21.97±00.12 | 25.43±00.33 |
| | F1/AUROC | 06.00±01.98 | 24.77±00.90 | 19.63±00.91 | 28.71±01.45 | 10.93±00.03 | 14.43±00.62 | 24.07±00.41 |
| | Kappa/AUCPR | 00.00±00.00 | 09.17±00.54 | 04.43±00.68 | 12.90±00.57 | 00.45±00.03 | 02.50±00.16 | 07.07±00.37 |
| **ADFTD** | B-Acc | 33.30±00.00 | 51.37±00.09 | 38.20±00.77 | 39.03±02.48 | 38.17±00.13 | 41.30±00.28 | 46.10±00.24 |
| | F1/AUROC | 22.17±00.75 | 50.47±00.05 | 38.40±01.05 | 32.02±00.53 | 36.17±00.35 | 42.23±00.25 | 49.70±00.14 |
| | Kappa/AUCPR | 00.00±00.00 | 29.47±00.21 | 09.70±01.27 | 07.72±03.37 | 09.12±00.26 | 12.40±00.45 | 23.20±00.29 |
| **BCIC-2a** | B-Acc | 25.17±00.24 | 29.73±00.54 | 28.40±00.24 | 32.02±00.06 | 29.17±01.03 | 28.13±00.17 | 29.18±00.83 |
| | F1/AUROC | 10.65±00.92 | 25.10±00.43 | 22.83±00.45 | 25.95±01.13 | 24.94±01.72 | 18.57±00.12 | 23.37±01.36 |
| | Kappa/AUCPR | 00.22±00.31 | 06.97±00.79 | 04.57±00.33 | 09.40±00.08 | 05.56±01.37 | 04.17±00.26 | 05.57±01.12 |
| **SEED-VII** | B-Acc | 14.30±00.00 | 20.60±00.45 | 23.23±00.48 | 25.20±00.08 | 19.43±00.72 | 18.90±00.29 | 20.57±00.17 |
| | F1/AUROC | 03.63±00.61 | 16.60±00.43 | 19.63±00.91 | 24.17±00.12 | 16.76±00.29 | 16.60±00.50 | 20.50±00.33 |
| | Kappa/AUCPR | 00.00±00.00 | 07.50±00.42 | 04.43±00.68 | 12.70±00.08 | 06.97±00.95 | 06.27±00.41 | 08.50±00.29 |

*Table 5.* Performance comparison of 7 EEG-FMs on 14 BCI tasks under **LoRA multi-task** fine-tuning with **average pooling** classification head. † indicates overlap between the model's pre-training datasets and our benchmark datasets.

| Dataset | Metrics | BENDR | BIOT | LaBraM | EEGPT | CBraMod | CSBrain | REVE |
|---|---|---|---|---|---|---|---|---|
| **SEED** | B-Acc | 33.30±00.00 | 62.70±00.14 | 45.90±00.22† | 69.97±00.05† | 66.07±00.21 | 64.67±00.05 | 69.23±00.21 |
| | F1/AUROC | 17.00±00.00 | 62.60±00.08 | 43.07±02.04† | 69.27±00.61† | 65.57±00.38 | 64.07±00.17 | 68.97±00.46 |
| | Kappa/AUCPR | 00.00±00.00 | 44.20±00.22 | 18.90±00.29† | 55.10±00.08† | 49.27±00.29 | 47.20±00.08 | 54.03±00.34 |
| **PhysioMI** | B-Acc | 25.10±00.08 | 26.93±00.05 | 27.40±00.16 | 45.83±00.85† | 27.60±00.22 | 30.68±00.28 | 53.95±00.33 |
| | F1/AUROC | 11.90±02.55 | 26.17±00.48 | 20.47±01.13 | 45.33±00.74† | 24.50±02.03 | 29.92±00.61 | 54.18±00.31 |
| | Kappa/AUCPR | 00.13±00.12 | 02.57±00.09 | 03.20±00.24 | 27.80±01.10† | 03.50±00.29 | 07.57±00.38 | 38.62±00.46 |
| **Workload** | B-Acc | 50.00±00.00 | 58.30±00.71 | 55.10±07.21 | 76.87±00.37 | 62.80±00.57 | 67.33±01.42 | 71.55±02.13 |
| | F1/AUROC | 49.20±00.64 | 78.83±00.58 | 72.07±02.26 | 87.63±01.09 | 81.63±00.90 | 74.53±00.31 | 81.03±01.48 |
| | Kappa/AUCPR | 25.47±01.06 | 58.60±00.57 | 43.80±03.72 | 76.43±01.52 | 62.80±01.93 | 45.48±01.25 | 61.33±03.53 |
| **TUEV** | B-Acc | 16.70±00.00 | 59.68±00.17 | 47.13±00.59 | 61.07±00.98 | 57.80±02.06 | 64.02±00.61 | 68.53±01.18 |
| | F1/AUROC | 44.10±00.00 | 81.28±00.31 | 72.33±01.32 | 86.83±00.24 | 77.60±03.19 | 79.05±01.32 | 77.83±00.86 |
| | Kappa/AUCPR | 00.00±00.00 | 68.33±00.59 | 54.33±02.28 | 79.60±01.07 | 63.07±04.76 | 65.80±02.08 | 58.30±00.99 |
| **TUAB** | B-Acc | 68.37±00.12 | 80.67±00.28 | 73.93±00.65 | 83.93±00.50 | 78.40±00.93 | 78.97±00.71 | 80.43±01.04 |
| | F1/AUROC | 76.77±00.21 | 88.10±00.29 | 82.57±00.48 | 91.93±00.39 | 86.07±00.97 | 87.07±01.01 | 88.03±00.54 |
| | Kappa/AUCPR | 71.93±00.41 | 88.78±00.32 | 81.67±00.74 | 91.93±00.21 | 87.07±00.95 | 87.23±01.09 | 88.07±01.16 |
| **HMC** | B-Acc | 20.00±00.00 | 70.43±00.19 | 44.10±01.31 | 71.97±00.12 | 69.13±00.17 | 69.93±00.09 | 71.03±00.25 |
| | F1/AUROC | 20.30±00.00 | 74.03±00.19 | 47.10±01.56 | 72.27±00.39 | 71.60±00.73 | 73.13±00.17 | 71.77±00.25 |
| | Kappa/AUCPR | 00.00±00.00 | 65.90±00.24 | 32.67±01.39 | 64.63±00.21 | 63.00±00.28 | 64.70±00.14 | 63.83±00.26 |
| **Siena** | B-Acc | 50.00±00.00 | 68.83±00.47 | 50.80±00.57 | 84.57±00.61 | 82.13±00.78 | 72.87±01.11 | 75.33±01.23 |
| | F1/AUROC | 80.70±05.21 | 88.20±01.00 | 68.57±17.66 | 94.50±00.45 | 93.20±01.35 | 87.80±03.18 | 88.73±01.05 |
| | Kappa/AUCPR | 99.67±00.19 | 99.80±00.00 | 99.20±00.65 | 99.90±00.00 | 99.83±00.05 | 99.77±00.11 | 99.83±00.05 |
| **TUSL** | B-Acc | 33.30±00.00 | 62.90±00.28 | 54.68±00.47 | 72.59±00.81 | 73.90±00.74 | 75.03±02.53 | 80.68±02.81 |
| | F1/AUROC | 25.70±00.00 | 54.53±00.47 | 37.08±01.33 | 71.10±00.03 | 67.57±01.41 | 71.53±03.20 | 80.78±02.41 |
| | Kappa/AUCPR | 00.00±00.00 | 36.93±00.33 | 25.47±00.82 | 56.74±00.23 | 53.93±01.47 | 58.12±04.65 | 71.43±03.54 |
| **Mimul-11** | B-Acc | 33.30±00.00 | 40.57±00.37 | 34.87±00.24 | 43.00±00.36 | 42.90±00.92 | 39.10±00.24 | 43.77±00.82 |
| | F1/AUROC | 37.90±00.00 | 49.53±00.31 | 41.67±00.45 | 52.00±00.50 | 50.00±01.22 | 48.23±00.26 | 50.33±01.98 |
| | Kappa/AUCPR | 00.00±00.00 | 12.40±00.57 | 03.63±00.40 | 18.90±00.29 | 15.07±01.62 | 12.13±00.37 | 17.90±02.01 |
| **Things EEG 2** | B-Acc | 50.00±00.00 | 50.67±00.05 | 50.00±00.00 | 51.70±00.37 | 50.23±00.05 | 50.00±00.00 | 53.77±00.12 |
| | F1/AUROC | 50.27±00.12 | 64.13±00.17 | 55.27±02.33 | 61.83±00.21 | 57.60±00.79 | 54.43±00.62 | 60.70±00.24 |
| | Kappa/AUCPR | 11.03±00.05 | 19.47±00.05 | 13.13±01.13 | 14.97±00.31 | 14.17±00.31 | 13.03±00.21 | 15.70±00.22 |
| **SEED-V** | B-Acc | 20.00±00.00 | 28.37±00.12 | 23.93±00.53 | 36.07±00.85 | 35.03±00.27 | 30.43±00.17 | 38.37±00.21 |
| | F1/AUROC | 08.80±00.00 | 27.87±00.33 | 22.10±00.80 | 35.77±01.77 | 33.22±00.54 | 28.20±00.65 | 38.87±00.42 |
| | Kappa/AUCPR | 00.00±00.00 | 11.33±00.19 | 05.17±00.69 | 20.10±00.75 | 18.88±00.33 | 13.17±00.29 | 22.70±00.49 |
| **ADFTD** | B-Acc | 33.87±00.80 | 48.77±01.40 | 43.07±00.66 | 44.93±01.18 | 48.83±00.50 | 48.60±00.45 | 50.73±01.28 |
| | F1/AUROC | 18.77±07.78 | 51.33±01.88 | 45.77±00.76 | 45.80±01.31 | 51.57±00.25 | 47.20±00.57 | 24.73±02.80 |
| | Kappa/AUCPR | 00.83±01.18 | 25.27±01.93 | 18.40±01.28 | 17.73±01.70 | 26.27±01.86 | 27.33±00.74 | 56.03±00.40 |
| **BCIC-2a** | B-Acc | 25.00±00.00 | 29.00±00.14 | 28.30±00.73 | 36.70±01.15 | 31.00±00.70 | 32.00±00.99 | 38.63±00.19 |
| | F1/AUROC | 10.00±00.00 | 19.40±00.14 | 17.87±01.01 | 32.67±03.16 | 22.20±02.19 | 20.77±01.11 | 38.43±00.17 |
| | Kappa/AUCPR | 00.00±00.00 | 05.33±00.19 | 04.43±00.93 | 15.60±01.53 | 08.03±00.98 | 09.30±01.35 | 18.17±00.25 |
| **SEED-VII** | B-Acc | 14.30±00.00 | 23.85±00.08 | 18.13±00.71 | 29.83±00.53 | 18.73±00.19 | 22.73±00.09 | 23.93±00.58 |
| | F1/AUROC | 04.77±00.33 | 21.95±00.12 | 13.30±01.36 | 27.30±01.55 | 14.87±01.00 | 22.27±00.24 | 21.67±01.76 |
| | Kappa/AUCPR | 00.00±00.00 | 11.37±00.11 | 04.37±01.23 | 17.73±01.03 | 05.00±00.36 | 10.80±00.08 | 11.48±00.88 |

*Table 6.* Performance comparison of 7 EEG-FMs on 14 BCI tasks under **full-parameter multi-task** fine-tuning from **randomly initialized** model with **average pooling** classification head. † indicates overlap between the model's pre-training datasets and our benchmark datasets.

| Dataset | Metrics | BENDR | BIOT | LaBraM | EEGPT | CBraMod | CSBrain | REVE |
|---|---|---|---|---|---|---|---|---|
| **SEED** | B-Acc | 40.50±01.92 | 60.40±00.29 | 63.43±00.62† | 41.70±01.88† | 66.27±00.21 | 38.20±03.67 | 51.27±00.31 |
| | F1/AUROC | 37.07±02.82 | 60.53±00.26 | 63.20±00.51† | 41.53±02.30† | 64.97±00.47 | 30.77±04.46 | 50.50±00.33 |
| | Kappa/AUCPR | 10.73±02.88 | 40.73±00.46 | 45.33±01.00† | 12.83±03.06† | 49.57±00.29 | 07.30±05.49 | 27.03±00.47 |
| **PhysioMI** | B-Acc | 26.37±01.24 | 32.27±00.28 | 39.53±03.19 | 26.10±00.36† | 29.30±00.83 | 25.37±00.12 | 52.62±00.61 |
| | F1/AUROC | 13.97±02.43 | 32.17±00.29 | 38.93±04.42 | 22.93±02.14† | 28.73±00.97 | 17.97±01.05 | 52.45±00.77 |
| | Kappa/AUCPR | 01.80±01.69 | 09.70±00.37 | 19.47±04.27 | 01.43±00.45† | 05.73±01.16 | 00.47±00.21 | 36.80±00.84 |
| **Workload** | B-Acc | 50.97±01.37 | 56.32±01.24 | 39.53±03.19 | 65.82±00.98 | 70.53±02.00 | 50.73±00.77 | 63.90±00.45 |
| | F1/AUROC | 46.67±05.48 | 71.87±01.31 | 38.93±04.42 | 76.98±03.14 | 77.17±02.44 | 58.13±03.42 | 68.07±01.37 |
| | Kappa/AUCPR | 25.73±04.25 | 50.35±00.95 | 19.47±04.27 | 59.37±04.15 | 60.50±01.10 | 32.37±00.74 | 44.20±02.87 |
| **TUEV** | B-Acc | 49.30±03.54 | 46.23±00.52 | 64.63±01.20 | 50.63±00.69 | 66.87±00.17 | 16.70±00.00 | 57.35±00.93 |
| | F1/AUROC | 70.10±05.93 | 73.37±00.48 | 76.73±00.70 | 80.87±00.92 | 81.83±00.82 | 44.10±00.00 | 73.35±01.08 |
| | Kappa/AUCPR | 52.80±09.26 | 55.60±00.85 | 60.80±01.31 | 69.43±01.55 | 69.73±01.24 | 00.07±00.09 | 57.13±01.28 |
| **TUAB** | B-Acc | 79.47±00.77 | 81.13±00.49 | 80.43±00.39 | 81.57±00.33 | 79.60±00.08 | 58.28±10.87 | 78.47±00.09 |
| | F1/AUROC | 89.43±01.05 | 90.27±00.40 | 89.17±00.91 | 89.17±00.33 | 87.43±00.73 | 60.72±14.31 | 87.73±00.09 |
| | Kappa/AUCPR | 89.77±01.01 | 90.53±00.45 | 89.70±00.36 | 89.23±00.12 | 88.00±00.36 | 57.32±15.40 | 86.33±00.40 |
| **HMC** | B-Acc | 60.47±03.28 | 66.73±00.31 | 67.70±00.41 | 58.80±00.45 | 68.77±00.21 | 20.00±00.00 | 67.22±00.38 |
| | F1/AUROC | 59.70±04.10 | 67.43±00.12 | 68.37±00.82 | 59.07±01.31 | 69.90±00.14 | 16.80±00.00 | 69.70±00.55 |
| | Kappa/AUCPR | 51.13±04.25 | 59.53±00.39 | 60.43±00.77 | 49.57±00.83 | 61.50±00.08 | 00.00±00.00 | 61.32±00.66 |
| **Siena** | B-Acc | 62.07±01.54 | 66.07±02.64 | 64.57±01.54 | 62.07±04.08 | 75.57±01.54 | 50.00±00.00 | 61.97±01.02 |
| | F1/AUROC | 92.90±03.43 | 79.37±04.79 | 89.27±03.11 | 93.10±00.73 | 92.20±01.48 | 52.05±02.90 | 74.07±01.19 |
| | Kappa/AUCPR | 99.87±00.05 | 99.30±00.14 | 99.80±00.08 | 99.90±00.00 | 99.87±00.05 | 99.03±00.05 | 99.27±00.05 |
| **TUSL** | B-Acc | 37.37±04.99 | 68.80±02.30 | 78.70±04.33 | 71.37±00.60 | 80.53±01.94 | 48.57±09.61 | 77.48±01.82 |
| | F1/AUROC | 21.53±09.24 | 64.50±03.88 | 71.30±05.30 | 68.90±02.98 | 78.05±01.44 | 43.10±14.54 | 72.75±02.21 |
| | Kappa/AUCPR | 03.50±04.74 | 50.50±03.64 | 60.67±06.76 | 53.10±02.98 | 67.40±02.66 | 23.30±15.21 | 54.35±03.20 |
| **Mimul-11** | B-Acc | 36.43±02.24 | 33.60±00.24 | 39.50±00.59 | 36.87±02.49 | 37.73±00.05 | 33.47±00.24 | 41.47±00.40 |
| | F1/AUROC | 43.37±03.78 | 41.70±00.29 | 46.57±00.97 | 42.60±03.52 | 46.70±00.16 | 39.80±01.84 | 48.18±01.22 |
| | Kappa/AUCPR | 07.00±04.96 | 01.07±00.90 | 11.90±00.83 | 07.33±05.15 | 09.37±00.42 | 00.33±00.47 | 14.50±01.44 |
| **Things EEG 2** | B-Acc | 50.00±00.00 | 50.00±00.00 | 50.13±00.05 | 56.22±00.31 | 53.53±01.01 | 50.10±00.14 | 52.37±00.31 |
| | F1/AUROC | 51.33±01.64 | 55.80±00.62 | 59.23±00.26 | 61.93±00.82 | 61.00±00.88 | 48.03±01.60 | 54.80±00.36 |
| | Kappa/AUCPR | 11.53±00.76 | 13.77±00.50 | 14.13±00.09 | 22.75±00.92 | 15.50±00.28 | 10.50±00.54 | 13.07±00.25 |
| **SEED-V** | B-Acc | 20.03±00.05 | 24.67±00.09 | 32.37±00.62 | 35.87±00.17 | 34.27±00.21 | 20.60±00.33 | 24.27±00.45 |
| | F1/AUROC | 09.43±00.53 | 22.70±01.00 | 30.83±00.86 | 36.87±00.54 | 34.20±01.22 | 09.47±02.38 | 22.67±01.76 |
| | Kappa/AUCPR | 00.07±00.09 | 06.70±00.37 | 16.30±00.65 | 20.20±00.36 | 18.23±00.34 | 00.70±00.37 | 05.50±00.37 |
| **ADFTD** | B-Acc | 33.90±00.43 | 46.23±00.45 | 45.97±01.10 | 44.17±01.25 | 45.83±00.69 | 33.30±00.00 | 44.88±01.33 |
| | F1/AUROC | 28.30±05.20 | 47.37±01.09 | 48.67±01.35 | 45.57±01.44 | 47.27±02.89 | 17.77±06.98 | 45.62±00.68 |
| | Kappa/AUCPR | 01.03±00.77 | 20.47±01.30 | 24.70±02.26 | 20.50±02.21 | 19.90±01.94 | 00.00±00.00 | 20.33±02.07 |
| **BCIC-2a** | B-Acc | 25.13±00.12 | 28.80±00.54 | 32.87±00.88 | 30.83±01.70 | 32.23±00.66 | 26.50±00.57 | 31.00±00.33 |
| | F1/AUROC | 13.83±04.34 | 20.50±01.73 | 28.13±02.88 | 24.23±03.35 | 27.97±01.48 | 18.77±02.02 | 30.30±00.33 |
| | Kappa/AUCPR | 00.13±00.12 | 05.10±00.70 | 10.50±01.21 | 07.80±02.30 | 09.63±00.88 | 01.97±00.78 | 08.00±00.49 |
| **SEED-VII** | B-Acc | 17.33±00.62 | 20.13±00.05 | 23.80±01.06 | 17.57±00.09 | 22.00±00.64 | 14.33±00.05 | 20.83±00.33 |
| | F1/AUROC | 09.47±00.25 | 17.80±00.49 | 21.13±01.96 | 10.73±01.09 | 18.53±00.73 | 04.43±00.54 | 20.50±01.27 |
| | Kappa/AUCPR | 03.80±00.37 | 07.50±00.08 | 11.37±01.27 | 04.37±00.17 | 08.57±00.78 | 00.03±00.05 | 08.03±00.34 |

*Table 7.* Performance comparison of 4 EEG-FMs on 14 BCI tasks under **full-parameter multi-task** fine-tuning with **attention pooling** classification head. † indicates overlap between the model's pre-training datasets and our benchmark datasets.

| Dataset | Metrics | LaBraM | CBraMod | CSBrain | REVE |
|---|---|---|---|---|---|
| **SEED** | B-Acc | 66.93±00.09$^\dagger$ | 70.00±00.29 | 68.77±00.17 | 73.90±00.42 |
| | F1/AUROC | 66.43±00.09$^\dagger$ | 69.40±00.50 | 67.93±00.48 | 73.87±00.44 |
| | Kappa/AUCPR | 50.53±00.17$^\dagger$ | 55.13±00.49 | 53.30±00.22 | 61.10±00.57 |
| **PhysioMI** | B-Acc | 38.30±01.16 | 51.33±00.76 | 57.52±00.48 | 58.62±00.27 |
| | F1/AUROC | 35.50±01.63 | 51.27±00.73 | 57.57±00.41 | 58.72±00.41 |
| | Kappa/AUCPR | 17.77±01.53 | 35.10±01.00 | 43.37±00.63 | 44.82±00.39 |
| **Workload** | B-Acc | 75.10±02.63 | 73.63±01.84 | 70.67±00.19 | 71.27±03.87 |
| | F1/AUROC | 85.60±02.76 | 79.60±01.67 | 73.63±01.36 | 80.00±04.97 |
| | Kappa/AUCPR | 71.20±03.97 | 49.67±00.41 | 45.63±02.89 | 55.87±08.97 |
| **TUEV** | B-Acc | 73.93±01.15 | 66.67±00.46 | 73.90±01.31 | 72.70±00.29 |
| | F1/AUROC | 85.82±01.00 | 82.40±00.94 | 86.46±02.11 | 89.53±00.65 |
| | Kappa/AUCPR | 75.80±01.77 | 70.23±01.84 | 75.43±02.37 | 82.90±01.43 |
| **TUAB** | B-Acc | 80.47±00.52 | 82.07±00.53 | 79.72±00.24 | 82.10±00.29 |
| | F1/AUROC | 87.10±02.56 | 89.70±01.39 | 87.17±01.01 | 87.73±00.05 |
| | Kappa/AUCPR | 86.67±02.98 | 90.03±01.27 | 86.63±01.54 | 87.77±01.19 |
| **HMC** | B-Acc | 70.83±00.48 | 70.80±00.36 | 71.77±00.21 | 71.53±00.23 |
| | F1/AUROC | 71.00±00.67 | 71.30±00.86 | 74.33±00.24 | 74.50±00.44 |
| | Kappa/AUCPR | 63.20±00.78 | 63.93±00.90 | 66.40±00.08 | 66.53±00.64 |
| **Siena** | B-Acc | 74.03±01.89 | 81.87±00.47 | 75.17±01.25 | 83.87±01.25 |
| | F1/AUROC | 85.63±01.82 | 91.87±01.86 | 93.93±02.82 | 89.33±01.03 |
| | Kappa/AUCPR | 99.77±00.05 | 99.83±00.05 | 99.90±00.08 | 99.77±00.05 |
| **TUSL** | B-Acc | 71.43±01.57 | 82.07±01.25 | 79.85±02.75 | 74.67±00.70 |
| | F1/AUROC | 65.95±03.63 | 80.28±02.08 | 79.43±03.28 | 75.63±01.00 |
| | Kappa/AUCPR | 57.65±03.16 | 70.78±02.28 | 69.10±05.59 | 63.00±01.44 |
| **Mimul-11** | B-Acc | 42.37±00.40 | 41.87±00.78 | 42.10±00.51 | 44.03±00.50 |
| | F1/AUROC | 50.55±01.09 | 51.30±01.04 | 49.83±00.59 | 51.67±01.15 |
| | Kappa/AUCPR | 16.62±01.76 | 18.00±01.85 | 15.07±00.83 | 21.87±00.31 |
| **Things EEG 2** | B-Acc | 54.17±00.31 | 59.52±00.36 | 62.32±01.15 | 64.62±00.55 |
| | F1/AUROC | 56.47±00.12 | 67.98±01.01 | 70.58±02.11 | 72.43±00.74 |
| | Kappa/AUCPR | 15.30±00.00 | 30.98±01.18 | 35.97±01.69 | 39.03±00.59 |
| **SEED-V** | B-Acc | 40.15±01.55 | 37.90±00.22 | 38.90±00.45 | 40.85±00.43 |
| | F1/AUROC | 39.28±01.96 | 37.40±00.91 | 38.80±00.29 | 39.87±00.80 |
| | Kappa/AUCPR | 25.75±01.84 | 22.30±00.50 | 23.30±00.51 | 25.65±00.58 |
| **ADFTD** | B-Acc | 53.80±00.54 | 51.30±00.45 | 51.15±01.07 | 46.80±00.83 |
| | F1/AUROC | 54.08±00.78 | 52.13±01.44 | 51.65±01.38 | 47.50±01.51 |
| | Kappa/AUCPR | 34.10±01.35 | 27.37±01.30 | 27.90±01.67 | 22.53±01.59 |
| **BCIC-2a** | B-Acc | 35.60±01.00 | 36.67±01.11 | 48.27±01.21 | 46.70±01.25 |
| | F1/AUROC | 31.18±01.69 | 26.83±02.01 | 46.75±01.92 | 43.52±01.65 |
| | Kappa/AUCPR | 14.13±01.37 | 15.53±01.52 | 31.07±01.60 | 28.95±01.67 |
| **SEED-VII** | B-Acc | 26.07±00.64 | 25.43±00.83 | 24.20±00.99 | 25.07±00.63 |
| | F1/AUROC | 22.42±02.49 | 24.15±00.40 | 23.33±01.50 | 24.38±01.01 |
| | Kappa/AUCPR | 13.67±01.17 | 13.12±00.70 | 12.07±01.72 | 12.72±00.92 |

*Table 8.* Performance comparison of 4 EEG-FMs on 14 BCI tasks under **full-parameter multi-task** fine-tuning with **large compressing MLP** classification head. † indicates overlap between the model's pre-training datasets and our benchmark datasets.

| Dataset | Metrics | LaBraM | CBraMod | CSBrain | REVE |
|---|---|---|---|---|---|
| **SEED** | B-Acc | 64.53±00.62$^{\dagger}$ | 71.38±00.65 | 71.03±00.60 | 71.92±00.31 |
| | F1/AUROC | 64.52±00.74$^{\dagger}$ | 71.28±00.65 | 70.93±00.55 | 71.93±00.48 |
| | Kappa/AUCPR | 47.03±00.94$^{\dagger}$ | 57.42±00.94 | 56.88±00.95 | 58.08±00.46 |
| **PhysioMI** | B-Acc | 52.90±00.37 | 57.23±00.63 | 59.83±00.77 | 61.90±00.51 |
| | F1/AUROC | 52.60±00.49 | 57.02±00.60 | 59.47±00.84 | 61.33±00.61 |
| | Kappa/AUCPR | 37.17±00.50 | 42.95±00.85 | 46.40±01.00 | 48.50±00.73 |
| **Workload** | B-Acc | 62.05±02.83 | 69.73±02.04 | 70.37±02.58 | 69.18±02.43 |
| | F1/AUROC | 74.82±01.89 | 77.13±02.25 | 81.73±00.95 | 79.80±01.55 |
| | Kappa/AUCPR | 52.48±04.74 | 55.03±03.58 | 63.00±03.82 | 54.73±03.07 |
| **TUEV** | B-Acc | 65.65±00.85 | 62.30±01.71 | 64.45±01.31 | 72.48±01.16 |
| | F1/AUROC | 86.62±00.67 | 79.63±02.07 | 81.88±01.52 | 88.05±02.31 |
| | Kappa/AUCPR | 78.22±01.07 | 66.17±03.60 | 69.93±02.30 | 80.53±04.06 |
| **TUAB** | B-Acc | 80.93±00.24 | 81.54±00.73 | 80.30±00.22 | 81.97±00.60 |
| | F1/AUROC | 87.90±00.96 | 87.93±01.27 | 85.87±00.90 | 86.63±00.64 |
| | Kappa/AUCPR | 87.35±00.88 | 86.61±01.45 | 84.00±00.50 | 83.72±01.04 |
| **HMC** | B-Acc | 70.45±01.19 | 72.10±00.33 | 70.27±00.57 | 69.25±00.34 |
| | F1/AUROC | 73.13±01.05 | 75.13±00.21 | 73.67±00.68 | 71.30±00.34 |
| | Kappa/AUCPR | 66.18±01.54 | 67.33±00.19 | 65.23±01.03 | 62.65±00.81 |
| **Siena** | B-Acc | 63.71±02.06 | 65.40±03.60 | 62.50±05.88 | 81.78±02.63 |
| | F1/AUROC | 84.13±02.78 | 84.70±05.26 | 81.78±06.98 | 91.07±02.80 |
| | Kappa/AUCPR | 99.67±00.06 | 99.63±00.12 | 99.65±00.12 | 99.83±00.06 |
| **TUSL** | B-Acc | 66.67±01.13 | 75.53±03.36 | 68.48±01.66 | 70.05±03.07 |
| | F1/AUROC | 59.00±01.36 | 73.37±04.73 | 65.77±02.54 | 64.42±01.90 |
| | Kappa/AUCPR | 41.60±01.87 | 60.27±06.65 | 49.03±02.54 | 52.45±04.86 |
| **Mimul-11** | B-Acc | 45.45±00.30 | 44.67±00.52 | 45.67±00.60 | 49.00±00.08 |
| | F1/AUROC | 52.59±00.38 | 51.73±01.01 | 53.17±00.94 | 53.07±00.98 |
| | Kappa/AUCPR | 21.38±00.63 | 20.40±01.42 | 22.25±01.12 | 22.77±00.45 |
| **Things EEG 2** | B-Acc | 52.46±00.60 | 55.53±00.98 | 56.92±02.09 | 54.40±00.41 |
| | F1/AUROC | 61.65±01.82 | 60.33±00.66 | 64.90±01.36 | 55.66±00.47 |
| | Kappa/AUCPR | 18.28±01.24 | 16.67±00.48 | 25.53±04.62 | 13.93±00.21 |
| **SEED-V** | B-Acc | 36.59±00.35 | 37.93±00.12 | 31.80±00.50 | 38.23±00.35 |
| | F1/AUROC | 36.58±00.31 | 37.17±00.26 | 31.82±00.99 | 38.22±00.99 |
| | Kappa/AUCPR | 20.66±00.48 | 21.93±00.25 | 14.80±00.78 | 22.75±00.61 |
| **ADFTD** | B-Acc | 47.97±00.20 | 53.27±01.72 | 49.90±00.42 | 54.62±01.18 |
| | F1/AUROC | 51.18±00.72 | 55.77±01.84 | 49.80±00.75 | 53.65±02.97 |
| | Kappa/AUCPR | 28.90±00.43 | 33.90±03.48 | 28.83±00.41 | 33.73±02.83 |
| **BCIC-2a** | B-Acc | 42.16±00.11 | 50.52±00.54 | 51.53±00.17 | 58.08±01.73 |
| | F1/AUROC | 41.25±00.17 | 49.15±00.72 | 51.05±00.31 | 57.33±02.25 |
| | Kappa/AUCPR | 22.88±00.14 | 34.03±00.71 | 35.07±00.18 | 44.10±02.29 |
| **SEED-VII** | B-Acc | 24.77±01.30 | 29.60±00.88 | 26.57±00.61 | 27.17±00.44 |
| | F1/AUROC | 22.63±02.00 | 27.27±01.69 | 24.13±00.65 | 25.87±00.76 |
| | Kappa/AUCPR | 12.40±01.42 | 17.57±01.24 | 13.97±00.69 | 14.82±00.51 |

*Table 9.* Performance comparison of 2 **foundation model for general time series** on 14 BCI tasks under **full-parameter multi-task** fine-tuning with **average pooling** classification head.

| Dataset | Metrics | Mantis | | | Moment | | |
|---|---|---|---|---|---|---|---|
| | | Full Param | Freeze | LoRA | Full Param | Freeze | LoRA |
| **SEED** | B-Acc | 60.70±01.22 | 47.50±00.36 | 60.10±00.99 | 58.40±00.22 | 54.03±00.71 | 57.33±00.50 |
| | F1/AUROC | 60.97±00.86 | 43.97±01.17 | 59.27±01.58 | 56.50±00.80 | 52.60±01.27 | 54.70±01.45 |
| | Kappa/AUCPR | 41.17±01.92 | 21.37±00.54 | 40.33±01.44 | 37.80±00.37 | 31.20±01.04 | 36.13±00.71 |
| **PhysioMI** | B-Acc | 28.33±00.58 | 27.10±00.33 | 28.60±00.22 | 28.20±00.62 | 26.23±00.47 | 28.23±00.77 |
| | F1/AUROC | 22.20±01.23 | 19.27±02.31 | 24.37±00.56 | 23.80±02.08 | 18.03±02.29 | 22.67±02.69 |
| | Kappa/AUCPR | 04.43±00.74 | 02.80±00.41 | 04.77±00.26 | 04.23±00.82 | 01.60±00.64 | 04.30±01.02 |
| **Workload** | B-Acc | 63.33±02.68 | 62.73±01.05 | 63.93±01.60 | 62.43±02.09 | 50.00±00.00 | 53.70±05.09 |
| | F1/AUROC | 75.60±01.88 | 74.03±00.53 | 80.20±00.78 | 69.43±00.95 | 72.90±00.29 | 72.33±03.99 |
| | Kappa/AUCPR | 51.83±01.97 | 45.83±01.04 | 57.57±01.16 | 40.63±00.45 | 41.27±00.34 | 42.30±05.40 |
| **TUEV** | B-Acc | 67.00±00.70 | 52.97±00.54 | 54.93±00.09 | 64.03±01.28 | 42.43±00.70 | 59.87±00.09 |
| | F1/AUROC | 87.43±00.45 | 85.20±00.65 | 70.93±01.69 | 85.47±00.52 | 74.10±00.99 | 76.77±00.26 |
| | Kappa/AUCPR | 79.43±01.02 | 76.07±01.11 | 55.40±01.96 | 76.67±00.95 | 58.87±00.90 | 62.57±00.48 |
| **TUAB** | B-Acc | 79.67±00.26 | 75.97±00.12 | 76.93±00.19 | 77.87±00.24 | 74.30±00.28 | 77.80±00.36 |
| | F1/AUROC | 86.83±00.25 | 84.07±00.12 | 85.37±00.26 | 86.03±00.40 | 82.87±00.05 | 86.27±00.76 |
| | Kappa/AUCPR | 87.63±00.33 | 84.20±00.14 | 85.93±00.33 | 85.20±00.42 | 81.07±00.09 | 85.43±00.77 |
| **HMC** | B-Acc | 67.23±00.57 | 56.23±00.29 | 65.33±00.59 | 64.20±00.90 | 55.00±00.88 | 61.37±00.69 |
| | F1/AUROC | 71.10±00.54 | 60.17±00.65 | 69.23±00.81 | 68.00±00.92 | 58.23±00.82 | 64.87±00.38 |
| | Kappa/AUCPR | 63.03±00.41 | 49.47±00.76 | 61.00±01.19 | 59.67±01.11 | 48.13±01.08 | 55.33±00.56 |
| **Siena** | B-Acc | 72.47±02.68 | 53.70±00.00 | 61.23±01.67 | 66.67±02.91 | 52.40±00.99 | 64.93±02.67 |
| | F1/AUROC | 90.80±00.99 | 92.10±00.93 | 89.77±03.20 | 85.83±02.49 | 85.07±00.79 | 79.77±02.74 |
| | Kappa/AUCPR | 99.77±00.05 | 99.90±00.00 | 99.87±00.05 | 99.77±00.09 | 99.80±00.00 | 99.67±00.05 |
| **TUSL** | B-Acc | 78.00±01.18 | 65.57±03.14 | 71.37±02.32 | 69.63±02.59 | 40.87±00.38 | 59.50±02.65 |
| | F1/AUROC | 78.40±00.99 | 53.43±04.78 | 66.17±02.95 | 70.03±02.17 | 26.10±00.28 | 54.23±02.93 |
| | Kappa/AUCPR | 66.80±01.70 | 39.43±04.88 | 51.30±03.85 | 54.00±03.36 | 05.80±00.28 | 34.27±03.87 |
| **Mimul-11** | B-Acc | 42.63±00.87 | 40.33±00.47 | 41.00±00.08 | 40.57±00.83 | 34.37±00.50 | 40.43±00.91 |
| | F1/AUROC | 50.30±02.36 | 49.00±00.59 | 49.63±00.87 | 48.53±00.31 | 40.07±01.15 | 49.40±00.96 |
| | Kappa/AUCPR | 17.97±01.82 | 15.00±01.14 | 15.57±00.56 | 14.27±01.23 | 02.43±01.16 | 15.07±01.88 |
| **Things EEG 2** | B-Acc | 57.37±01.08 | 50.00±00.00 | 50.23±00.05 | 51.90±00.59 | 50.00±00.00 | 50.87±00.37 |
| | F1/AUROC | 67.10±00.57 | 60.63±00.68 | 59.63±01.05 | 63.00±00.37 | 56.47±00.98 | 61.20±02.20 |
| | Kappa/AUCPR | 27.07±01.23 | 15.90±00.29 | 15.40±00.50 | 19.83±00.54 | 13.37±00.65 | 18.60±01.28 |
| **SEED-V** | B-Acc | 24.57±00.80 | 21.10±00.22 | 23.70±00.33 | 22.10±00.36 | 20.00±00.00 | 21.57±00.52 |
| | F1/AUROC | 21.00±02.28 | 15.30±01.99 | 19.03±00.66 | 16.47±01.11 | 08.80±00.00 | 15.17±01.76 |
| | Kappa/AUCPR | 06.57±01.32 | 01.73±00.40 | 05.40±00.36 | 03.30±00.65 | 00.00±00.00 | 02.53±00.91 |
| **ADFTD** | B-Acc | 50.63±00.09 | 34.87±00.66 | 45.00±01.13 | 50.03±02.15 | 40.40±00.90 | 49.13±01.91 |
| | F1/AUROC | 48.70±00.36 | 30.97±03.54 | 44.77±01.32 | 48.93±01.89 | 38.53±01.92 | 46.87±02.13 |
| | Kappa/AUCPR | 30.77±00.12 | 02.83±01.18 | 21.10±01.84 | 29.60±03.72 | 13.63±01.72 | 28.23±03.24 |
| **BCIC-2a** | B-Acc | 34.43±00.74 | 31.93±00.45 | 31.63±00.77 | 32.53±00.83 | 27.97±02.50 | 29.07±00.74 |
| | F1/AUROC | 28.40±01.39 | 27.03±00.60 | 24.90±02.01 | 25.27±01.43 | 15.37±03.28 | 18.70±01.53 |
| | Kappa/AUCPR | 12.57±00.97 | 09.30±00.57 | 08.87±01.03 | 10.10±01.06 | 03.97±03.35 | 05.43±00.95 |
| **SEED-VII** | B-Acc | 18.97±00.87 | 18.00±00.22 | 19.03±00.26 | 19.73±00.37 | 17.80±00.08 | 20.67±01.43 |
| | F1/AUROC | 16.17±01.01 | 15.73±00.69 | 17.27±01.06 | 17.87±00.42 | 14.60±00.14 | 17.73±02.00 |
| | Kappa/AUCPR | 06.23±01.14 | 04.83±00.31 | 06.30±00.33 | 07.43±00.53 | 04.73±00.12 | 08.70±02.00 |

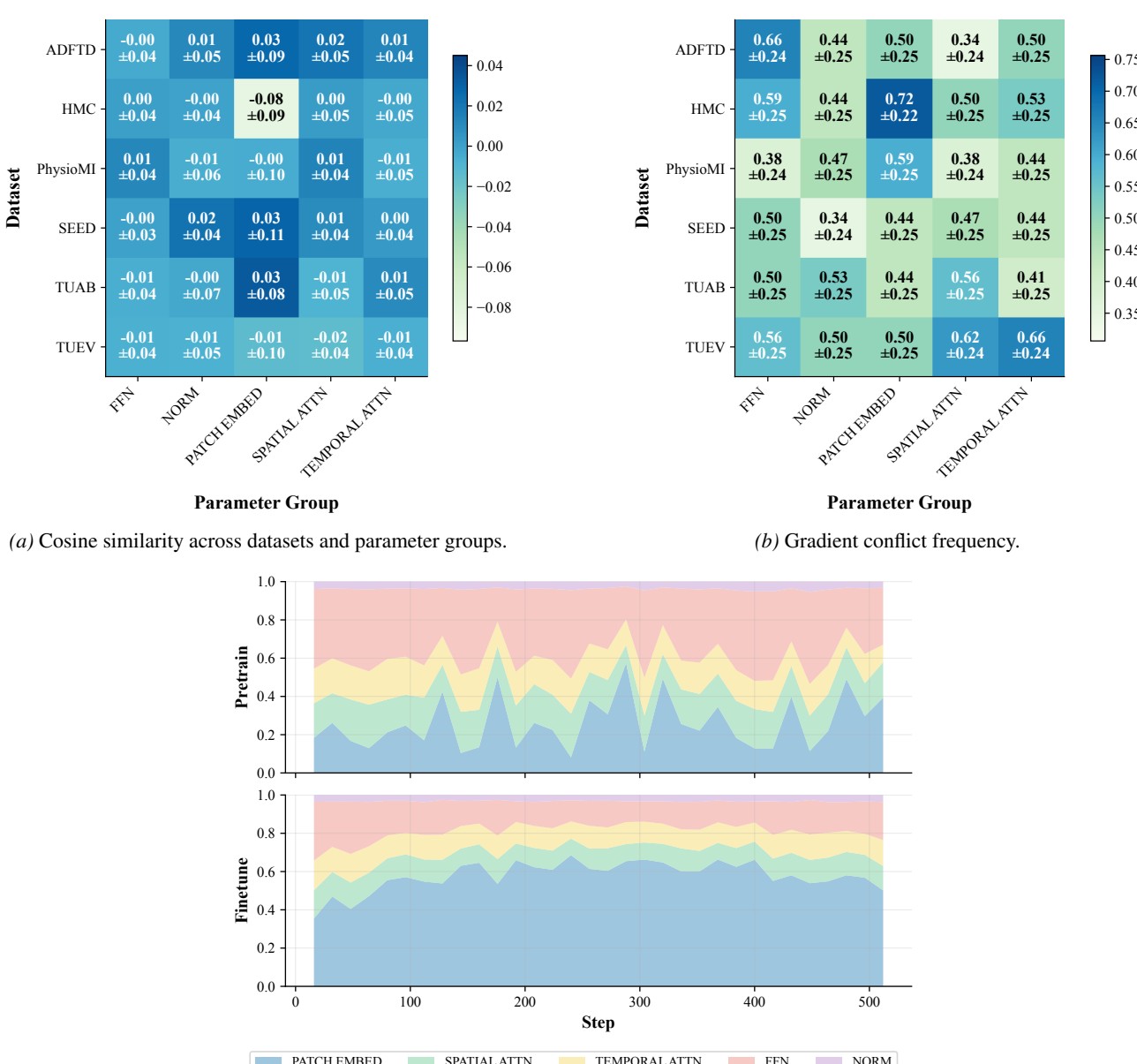

*(a)* Cosine similarity across datasets and parameter groups.

*(b)* Gradient conflict frequency.

*(c)* Relative gradient norm intensity across modules.

*Figure 18.* Gradient alignment between pre-training and downstream tasks for CBraMod.

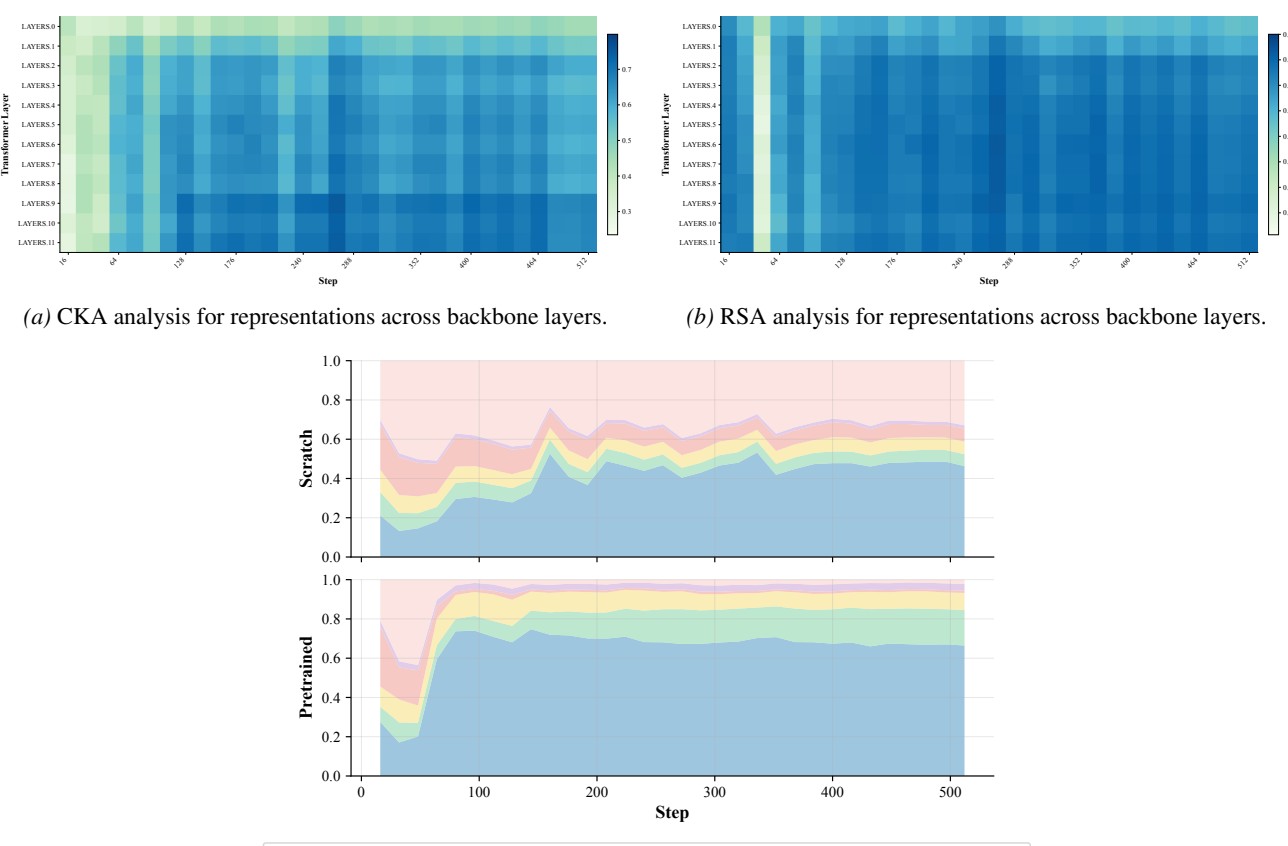

*(a)* CKA analysis for representations across backbone layers.

*(b)* RSA analysis for representations across backbone layers.

*(c)* Relative gradient norm intensity across modules.

*Figure 19.* Evolution of optimization dynamics during fine-tuning between from scratch and from pretrained checkpoint for CBraMod.

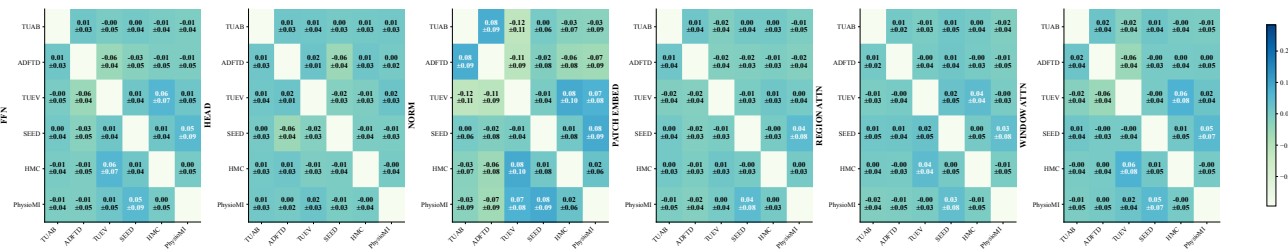

*Figure 20.* Cross-task gradient correlation analysis for CSBrain.

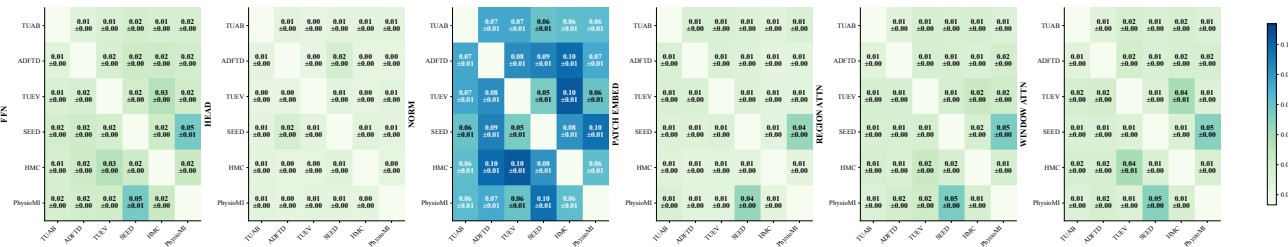

*Figure 21.* Cross-task subspace affinity analysis for CSBrain.

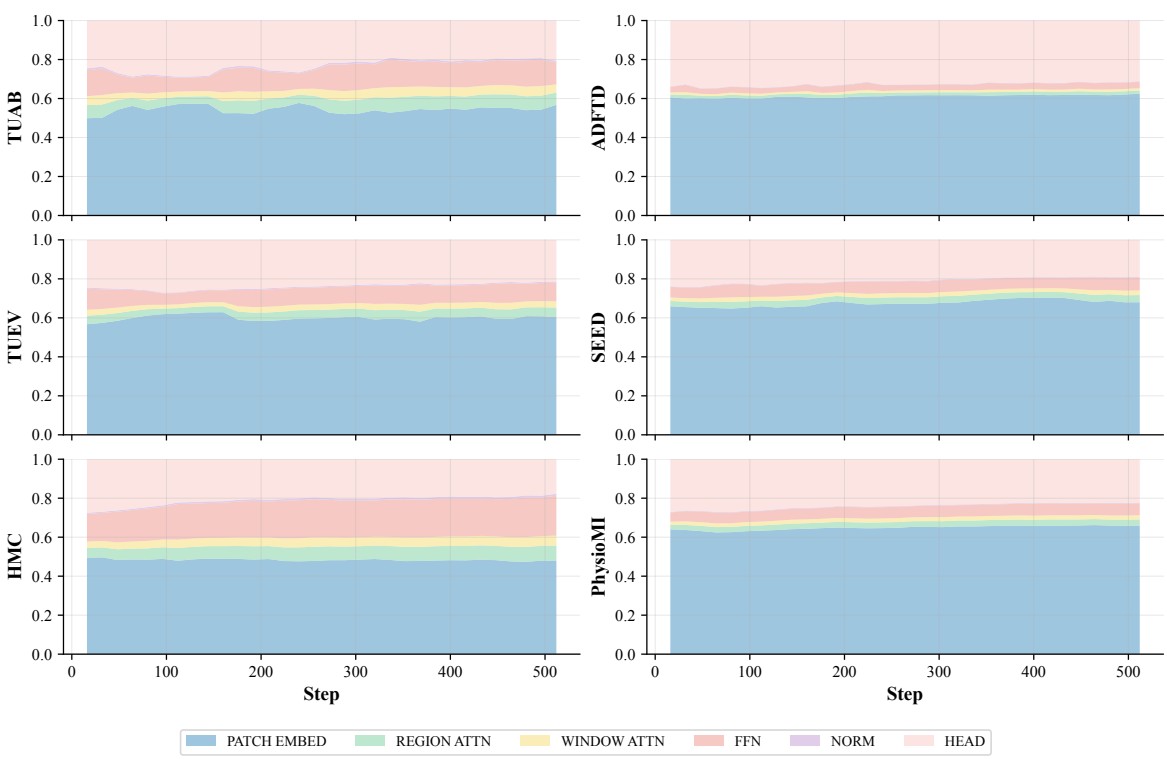

*Figure 22.* Cross-task evolution dynamics analysis for CSBrain.

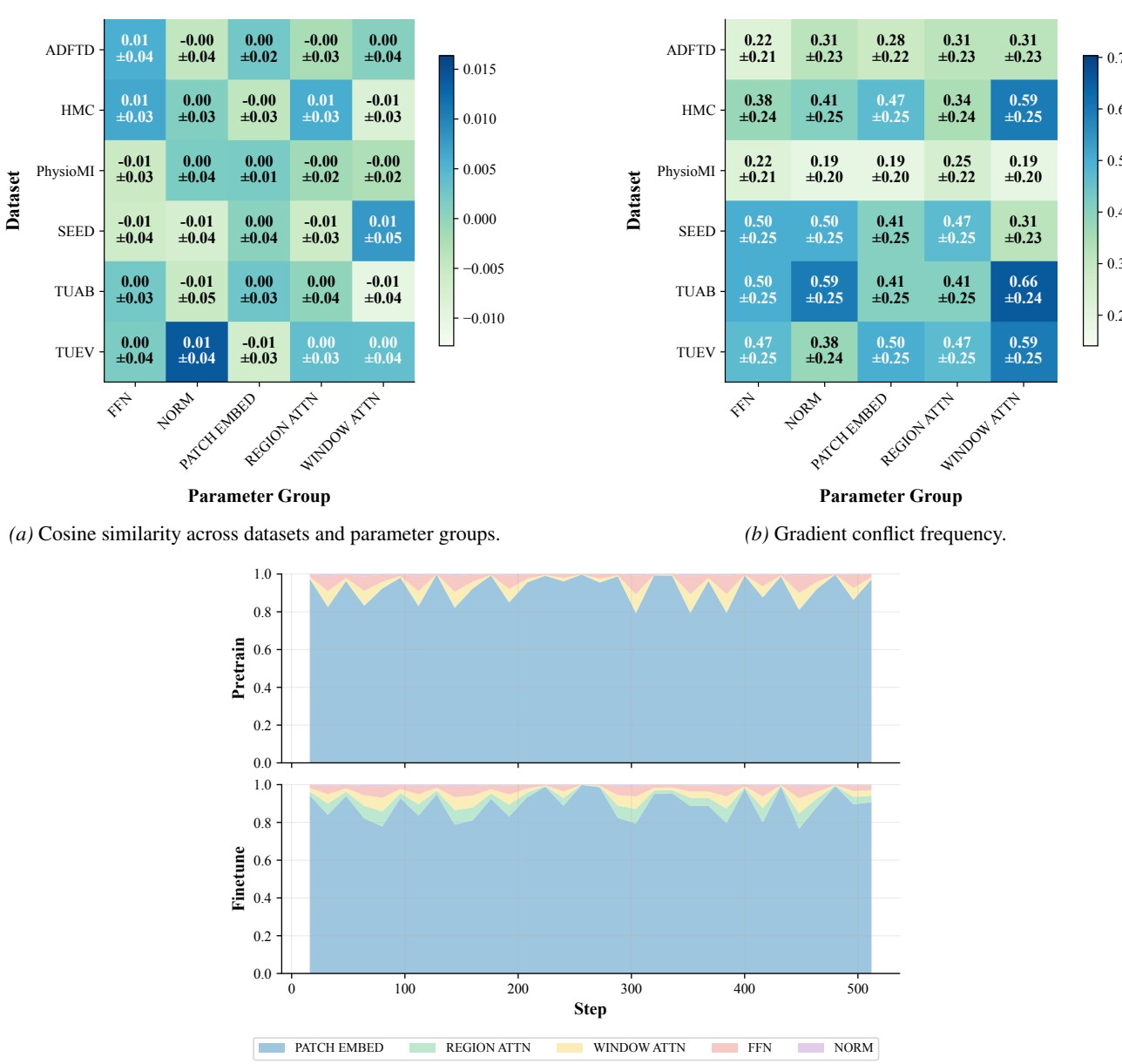

*(a)* Cosine similarity across datasets and parameter groups.

*(b)* Gradient conflict frequency.

*(c)* Relative gradient norm intensity across modules.

*Figure 23.* Gradient alignment between pre-training and down- stream tasks for CSBrain.

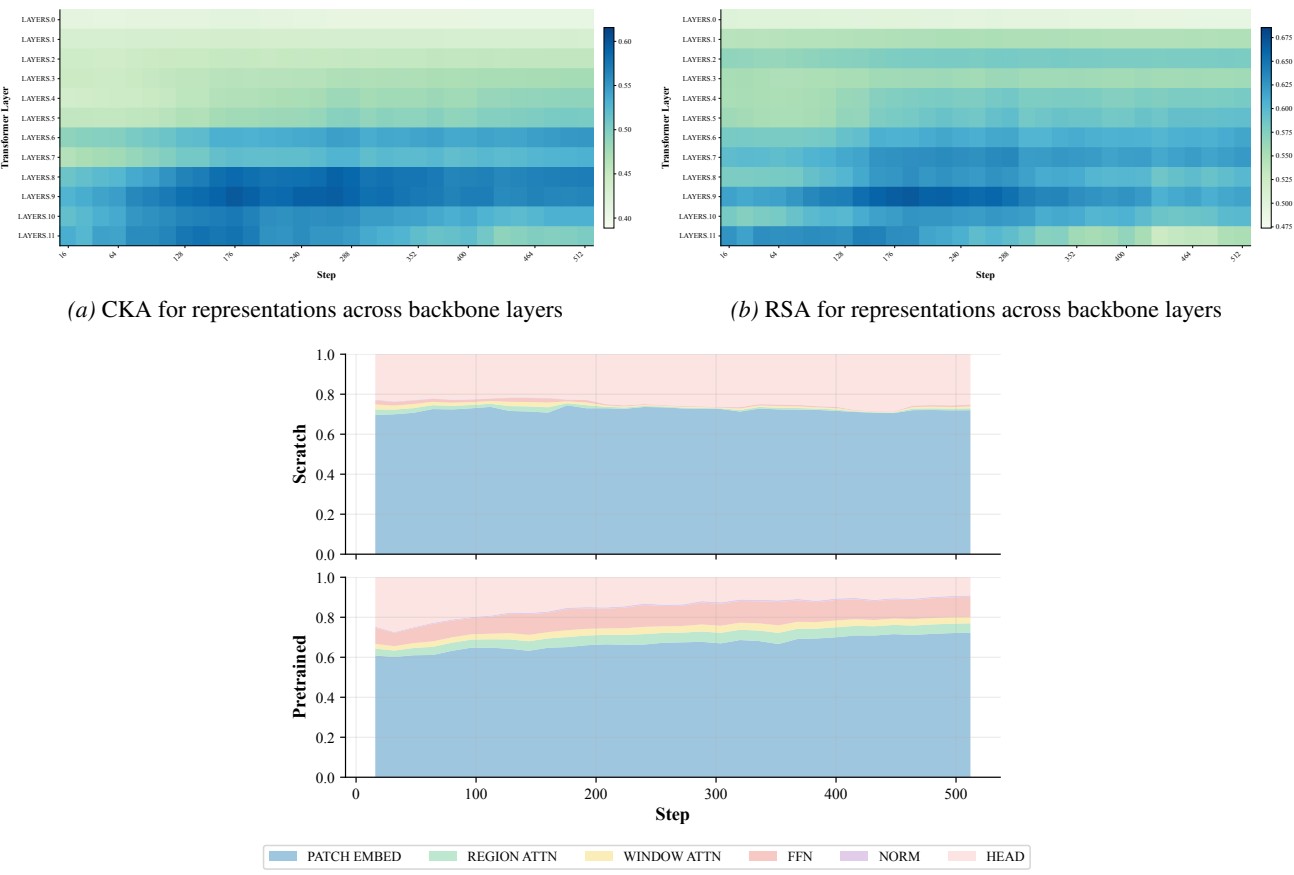

*(a)* CKA for representations across backbone layers

*(b)* RSA for representations across backbone layers

*(c)* Relative gradient norm intensity across modules.

*Figure 24.* Evolution of optimization dynamics during fine-tuning between from scratch and from pretrained checkpoint for CSBrain.

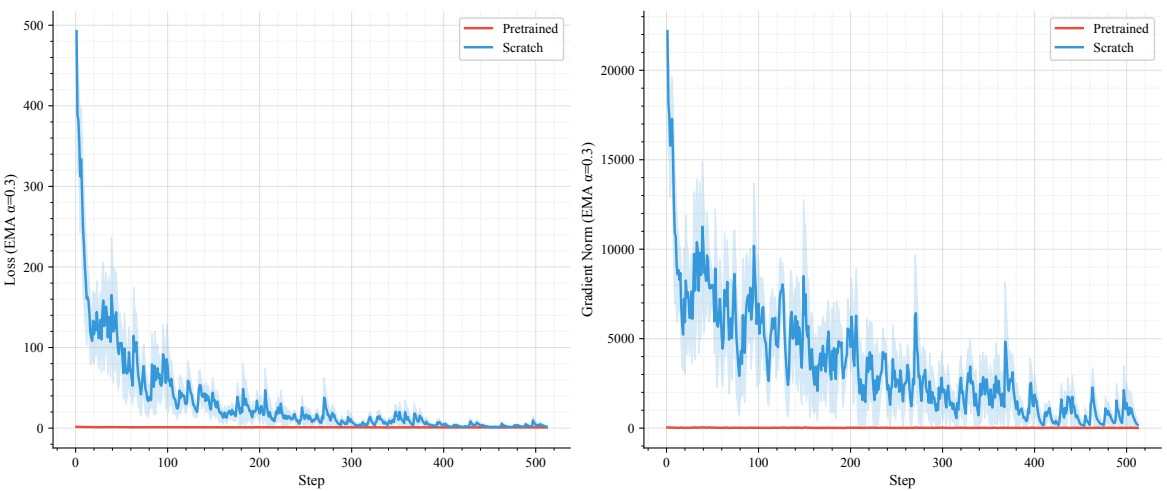

*Figure 25.* Training loss and gradient norm during fine-tuning between from scratch and from pretrained checkpoint for CSBrain.

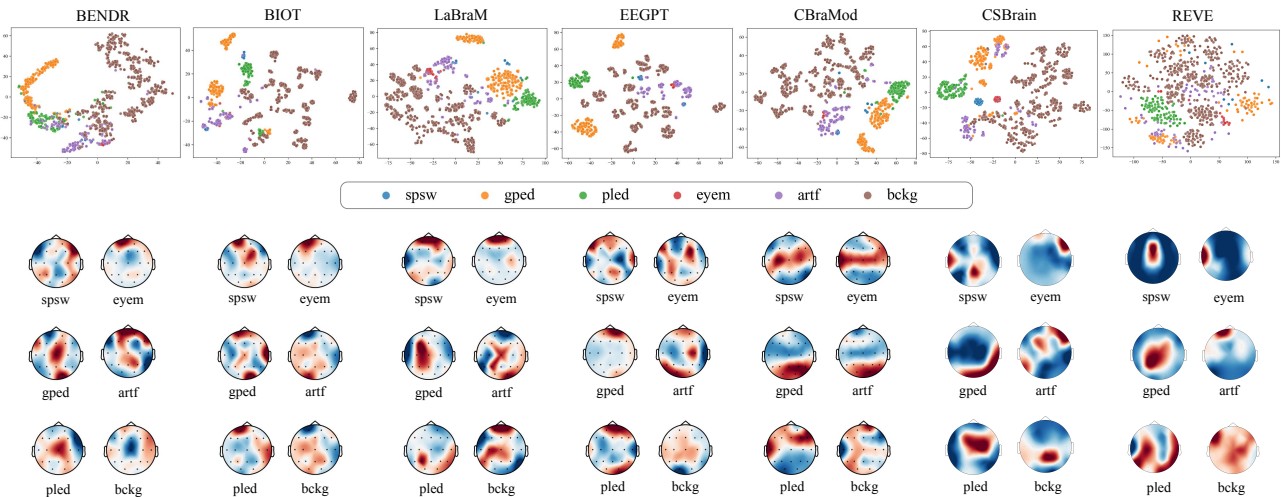

*Figure 26.* Visualization of models prediction results on TUEV.

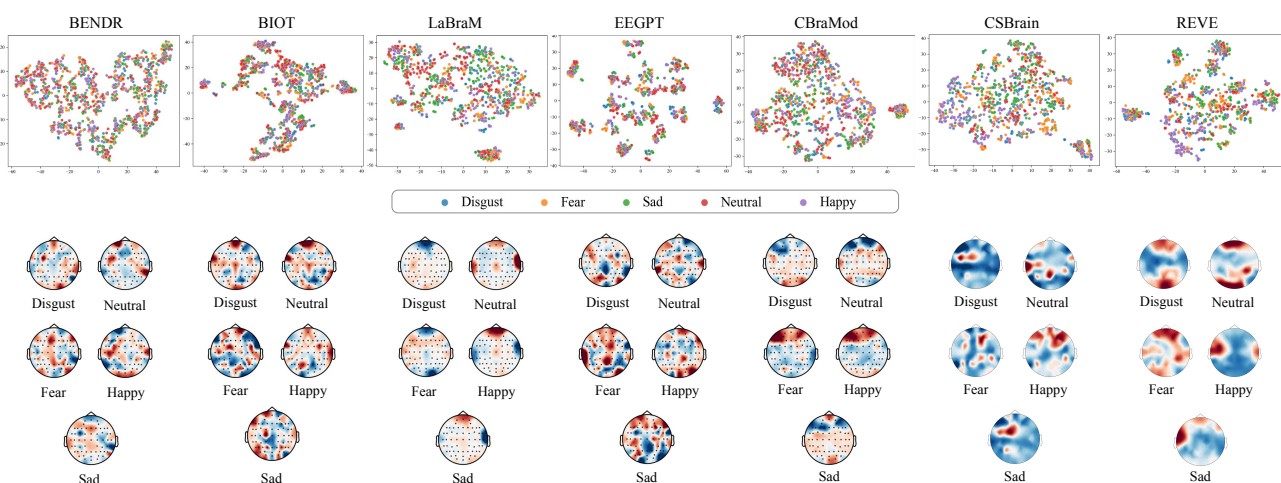

*Figure 27.* Visualization of models prediction results on SEED-V.

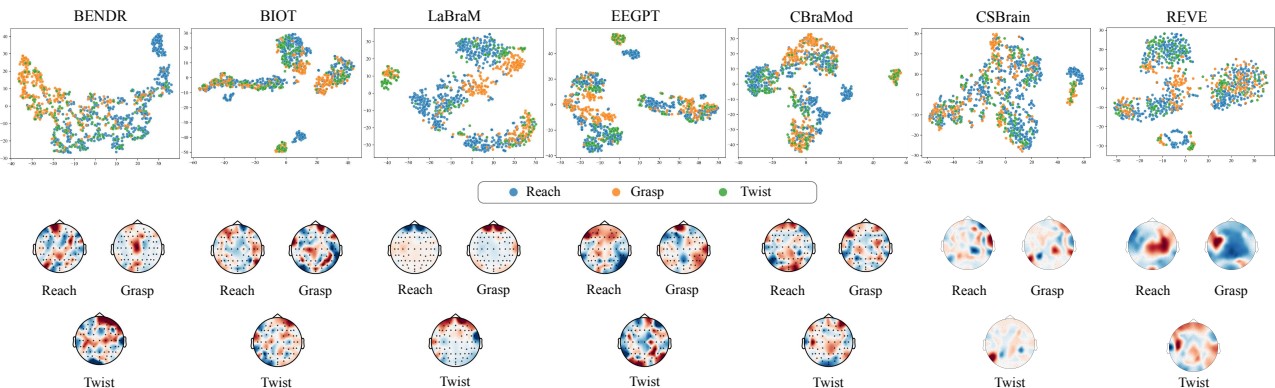

*Figure 28.* Visualization of models prediction results on Mimul-11.

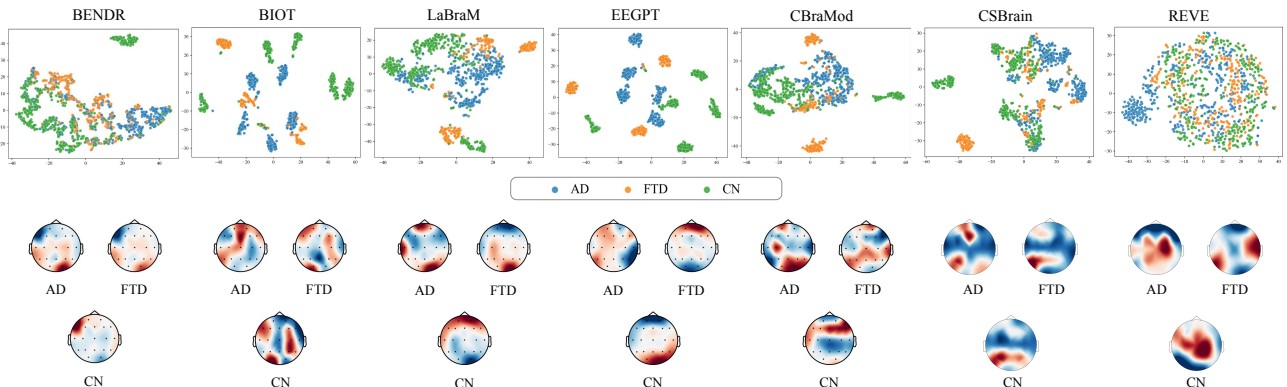

*Figure 29.* Visualization of models prediction results on ADFTD.

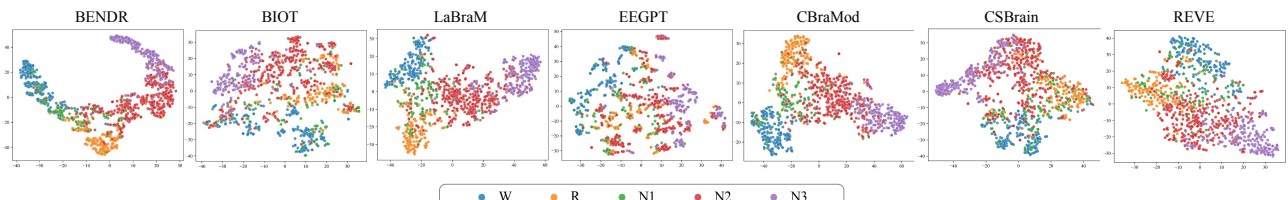

*Figure 30.* Visualization of models prediction results on HMC.

