# OpenReview forum: "EEG-FM-Bench: A Comprehensive Benchmark for the Systematic Evaluation and Diagnostic Analyses of EEG Foundation Models"
_ICML.cc/2026/Conference — ICML 2026 regular_

### Official Review · Reviewer_F4JV · 2026-03-01

**Soundness:** 3
**Presentation:** 3
**Significance:** 3
**Originality:** 2
**Overall Recommendation:** 4
**Confidence:** 5

**Summary:**

This work introduces a new benchmark for EEG foundation models. A critical issue in the field of EEG foundation models is the lack of benchmarks to compare their capabilities. This work extends previous efforts in the area like "Are Large Brainwave Foundation Models Capable Yet? Insights from Fine-tuning", Na Lee et al. and provides new insights in the field of EEG foundation models.

**Compliance With Llm Reviewing Policy:**

Affirmed.

**Final Justification:**

As mentioned in my rebuttal comment, the clarification improves the clarity of the message. I thank the authors for their responses. However, my concerns regarding the datasets remain unresolved. In conjunction with the argument of this being “another EEG benchmark,” I maintain my original assessment score.

**Key Questions For Authors:**

- I would like to see a more extensive discussion on the results of this benchmark. More particularly, what are the conclusions ? Why model X is better than model Y or overall ? Is it parameters, training corpus, architecture ? Essentially, what can we learn from this work and these insights ?
- What made the authors choose these datasets ? Aren't some of those present in the pre-training phase of some EEG foundation models ? Why choose SEED datasets where the labels are not objective or Physionet-MI which is a contaminated MI dataset ? In other words, the choice of this datasets can change completely the results we get out of this benchmark and yet it was justified why this particular selection took place.

**Limitations:**

The authors do not discuss the limitations of their work. Although they do provide some insights on future perspectives and directions for the community

**Strengths And Weaknesses:**

Strengths:
- This work provides a detailed analysis of the tackled issue (the lack of EEG foundation models benchmarks). The authors have extended  previous efforts in the field like "Are Large Brainwave Foundation Models Capable Yet? Insights from Fine-tuning", Na Lee et al. by adding several new aspects beyond fine-tuning strategies (frozen backbone, full-parameter, parameter efficient). The authors have included tasks setups (single-task, multi-task setting) and classification heads (patch average pooling, dimension compression, attention pooling).
- The evolution of optimisation dynamics during fine-tuning is also something novel in the EEG foundation models comparisons.
- The authors also provided cross-task gradient correlation and subspace affinity analyses further enhancing the insights into the pre-trained state-of-the-art EEG foundation models.

Weaknesses:
- I would like to see a more extensive discussion on the results of this benchmark. (In more detail in the questions part)
- The choice of dataset is not justified and I feel some architectures might have advantage over others if these datasets are included in the pre-training phase of their model  (In more detail in the questions part)

---

> ### Author Rebuttal · Authors · 2026-03-31
>
> ## W1-Q1. What can be learned from the benchmark?
>
> Thank you for highlighting the need for a clearer takeaway. Under a controlled downstream protocol, the benchmark indicates that current EEG-FM performance is better understood through a small set of bottlenecks.
>
> The most immediate one is the **upstream-downstream gap**: the consistently large frozen-backbone gap indicates that current pretraining still does not yield sufficiently task-ready representations for direct transfer. The second bottleneck is **adaptation stability in the small-data EEG regime**. Once full adaptation is allowed, multi-task training often improves robustness, suggesting that downstream performance is limited not only by representation quality, but also by optimization instability and overfitting. The third is the **adaptation interface inside the model**: our layer-wise analyses suggest that much of the Transformer backbone remains relatively stable during adaptation, while the input embedding bears much of the burden of bridging raw EEG and latent features; related cross-task analyses also suggest that normalization and temporal-embedding components capture more task-shared structure than attention or MLP blocks.
>
> This also helps explain why **nominal scale alone is not a sufficient account** of performance. Many of the original model papers already include their own analyses of parameter scaling and data scaling, but that is not the primary focus of our benchmark. Instead, under a matched downstream budget, our goal is to evaluate what released checkpoints actually deliver in transfer. From that perspective, compact EEG-specific architectures can remain competitive with, or stronger than, substantially larger alternatives, and richer heads are not a universal solution beyond motor imagery. Overall, the benchmark suggests that **pretrain/downstream gradient alignment, adaptation stability, and the adaptation interface** currently matter more than scale alone. Future progress is therefore more likely to come from improving pretraining objectives, adaptation interfaces, and downstream optimization stability.
>
> ## W2-Q2. Dataset choice and overlap concerns
>
> Thank you for asking us to make the dataset rationale explicit. The benchmark dataset selection follows two principles: **breadth across canonical EEG paradigms**, and **continuity with datasets that have already become standard reference points in the EEG-FM literature**. The benchmark is intended as a cross-paradigm diagnostic suite rather than a benchmark tailored to a single task family or a single model’s preferred domain. That is why it spans 14 datasets across 10 paradigms, including motor imagery, sleep staging, abnormal detection, event classification, seizure detection, emotion recognition, and others.
>
> At the same time, overlap-sensitive subsets do require careful treatment. Our position is not to remove such cases from the benchmark, but to isolate them clearly and interpret them with explicit caveats. **On the model side**, EEGPT remains a central reference in the current landscape, so its relevance is not negated by overlap with downstream datasets such as PhysioMI and SEED. Similarly, LaBraM’s pretraining data includes SEED, though not SEED-V or SEED-VII. **On the dataset side**, excluding every overlap-sensitive case would also remove some of the most established public EEG benchmarks: SEED remains a widely used benchmark for simple affective transfer, and PhysioMI remains one of the most widely used public motor-imagery datasets. We therefore retain these models and datasets, and we will make the overlap caveat explicit and avoid over-interpreting such results in isolation.
>
> To make the clarification more explicit, in **Table R-3,** we report an overlap-focused table that addresses the stronger question: does documented data overlap by itself determine the benchmark ranking?
>
> **Table R-3. Explicit overlap-sensitive cases. The benchmark values come from our Tables 2, 3, and 6 in the appendix.**
> https://anonymous.4open.science/r/EEG-FM-Rebuttal-Repo/Table-R3.md
>
> From **Table R-3**, we observe that overlap can matter, but it is not sufficient to explain the overall ranking by itself. This is consistent with our gradient-based diagnostics as shown in Figure 3,4, and 5 in the original paper: in the cross-task gradient-correlation and subspace-affinity analyses, shared structure is observed more broadly across datasets rather than being restricted to SEED or PhysioMI, and it is concentrated mainly in task-shared modules such as temporal embedding and normalization. The training-from-scratch setting is also informative here: it helps diagnose how much current EEG-FMs benefit from in-domain initialization, but it is not decisive in isolation for the broader benchmark conclusions. For this reason, we retain SEED and PhysioMI as informative and widely used evaluation subsets, while treating them explicitly as overlap-sensitive rather than as clean standalone evidence.

---

> > ### Author Rebuttal · Reviewer_F4JV · 2026-04-01
> >
> > The authors have responded to my questions. Especially question 1 makes the message clear now. I am not particularly in favour of keeping these datasets for example but I am still keeping the paper on the "weak accept" side.

---

> > > ### Author Response · Authors · 2026-04-02
> > >
> > > Thank you for acknowledging our rebuttal. We appreciate your thoughtful feedback, especially on the benchmark conclusions and dataset selection. We will make sure the overlap-related caveats are stated as clearly as possible in the revision. Thank you again for your time and consideration.

---

### Official Review · Reviewer_Jzyh · 2026-03-13

**Soundness:** 2
**Presentation:** 3
**Significance:** 3
**Originality:** 3
**Overall Recommendation:** 4
**Confidence:** 4

**Summary:**

The paper presents EEG-FM-Bench, a benchmark suite for the systematic evaluation of EEG foundation models under a unified pipeline. The benchmark covers 14 datasets across 10 EEG paradigms, compares multiple pretrained backbones under several fine-tuning strategies and classifier heads, and includes diagnostic analyses based on gradients and representations. The main claims are that multi-task training improves generalization, full fine-tuning remains the strongest adaptation strategy, reconstruction-based pretraining is only weakly aligned with downstream objectives, and compact EEG-specific architectures can outperform much larger models.

**Compliance With Llm Reviewing Policy:**

Affirmed.

**Final Justification:**

The rebuttal clarified my questions about the experimental validity and the performance gaps relative to prior work. The explanation of these differences is convincing; therefore, I am raising my rating to a 4.

**Key Questions For Authors:**

1. Since the benchmark is intended to provide fair comparison, could the authors show reproduction of the original downstream results for the main evaluated models, under settings as close as possible to the respective papers, before drawing conclusions from the unified pipeline? It is difficult to know whether discrepancies come from the models or from the benchmark implementation.
2. For the gradient analysis, why is the CountSketch projection dimension set to such a low value, and how sensitive are the conclusions to this choice? Could the authors provide ablations over projection dimension and also a normalization that accounts for parameter-group size?
3. Is code going to be released upon publication? Given that the central contribution is a benchmark, this would substantially affect the paper’s value to the community.

**Limitations:**

yes

**Strengths And Weaknesses:**

**Strengths**

The paper addresses a real need in the area, since EEG foundation models are currently compared under highly heterogeneous pipelines. Evaluation in EEG foundation models is currently fragmented, and a unified benchmark could be very useful to the community if done carefully. The benchmark is also very broad in scope, covering many datasets and paradigms.

Another strength is that the paper does not only report downstream metrics, but also attempts to analyze optimization and representation dynamics through gradient cosine similarity, subspace affinity, CKA, and RSA.  The experimental design is comprehensive: the authors vary fine-tuning strategy, task setup, and classifier head, which gives a richer picture than a single leaderboard.

**Weaknesses**

The main weakness is benchmark validity. The core promise of the paper is that it enables fair comparison through a unified and standardized pipeline. But this only holds if the benchmark reproduces the intended downstream behavior of the original models reasonably well. The paper states that it ensures consistency with prior settings, yet several reproduced results appear much worse than those reported in the original papers. If the benchmark substantially degrades some methods relative to their native pipelines, then the benchmark may be measuring compatibility with the authors’ chosen setup rather than intrinsic model quality. This is especially important because many of the paper’s conclusions, including claims about pretraining efficiency, scaling, and adaptation, depend on the trustworthiness of these reproduced baselines.

The paper repeatedly emphasizes reproducibility and even mentions an open-source implementation, but no code appears to be provided with the submission. For a benchmark paper, this is a serious limitation. Without code, it is difficult to verify preprocessing, model adaptation details, or the exact evaluation pipeline, which is precisely what the paper asks the community to trust.

---

> ### Author Rebuttal · Authors · 2026-03-31
>
> ## W1-Q1. Benchmark validity and closest-to-native reproduction
>
> Thank you for pushing on benchmark validity.
> Our benchmark focuses on building a unified benchmark to ensure fair comparison and provide diagnostic analyses with interpretable insights.
> Therefore, the goal of our benchmark is not to reproduce exactly every model’s best native number under all source-paper recipes, but rather to examine whether discrepancies follow a meaningful and interpretable pattern.
> If deviations are concentrated in settings that are especially dependent on specialized downstream engineering, while much smaller elsewhere, that pattern is more consistent with benchmark control than with a broken implementation.
>
> To make this point explicit, we report a closest-to-native sanity-check table in **Table R-2**. The table focuses on cases where the native recipes are especially distinctive and therefore most likely to diverge from a controlled benchmark. Motor imagery is the clearest example: EEGPT’s strongest BCIC-2a configuration depends on linear probing together with an adaptive spatial filter and MSConvNet, while REVE’s MI evaluations also use task-specific preprocessing such as Euclidean Alignment on BCIC-2a. By contrast, tasks such as TUAB and TUEV are much closer to a standard full-parameter downstream protocol, so benchmark-to-native gaps there are a more direct proxy for implementation fidelity.
>
> **Table R-2. Closest-to-native sanity-check cases. The unified-pipeline values are from our Table 2 (full-parameter single-task, average pooling setting) in the appendix.**
> https://anonymous.4open.science/r/EEG-FM-Rebuttal-Repo/Table-R2.md
>
> The resulting pattern is diagnostic. The largest gaps cluster in MI, exactly where the source papers rely most on task-specific downstream engineering. On non-MI datasets such as TUAB and TUEV, the benchmark is substantially closer to the native numbers, and in EEGPT-on-TUEV the unified result is even competitive with or above the paper value. The same interpretation is reflected by our classifier-head analysis: increasing head complexity mainly helps MI, which points to recipe sensitivity rather than generic implementation failure. For example, REVE rises from 30.63 ± 0.06 to 48.26 ± 1.24 on PhysioMI and from 36.89 ± 2.52 to 46.73 ± 0.79 on BCIC-2a when the head is made more MI-oriented.
>
> An exhaustive native audit over every model–dataset pair is also not the right standard for a benchmark of this kind, since several source papers depend not only on nominal preprocessing choices but also on random seeds, batch partitioning, training hardware, and task-specific hyperparameter-search budgets that are not exposed in a benchmark-ready form. The relevant question is whether a controlled benchmark preserves the robust ordering and failure pattern once those task-specific advantages are removed.
>
> ## W2-Q2. CountSketch dimension and PCA rank
>
> Thank you for raising the transparency issue in the gradient analysis. The previous presentation compressed two different stages of the pipeline into one sentence, which obscured the distinction.
>
> The **CountSketch projection dimension is 1024** for scalable gradient collection. The **6-dimensional quantity** used later is the PCA rank in the subspace-affinity analysis. They are not the same quantity. We will revise Sec. 4.3 to make the CountSketch-to-PCA pipeline explicit and add the related formulas.
>
> We also ran a CountSketch-dimension sensitivity analysis over 256, 512, 1024, 2048, 4096 before conducting the main gradient analysis in the original paper. The qualitative conclusions remain stable across this range: the sign pattern of cross-task cosine similarities, the relative ordering of task-pair affinities, and the cross-model trends do not materially change. To further motivate the PCA summary, we report a rank-sweep plot of shared gradient energy in **Figure R-1**, showing that the dominant shared structure saturates quickly at low rank, so so rank-6 PCA provides a compact summary rather than an arbitrary cutoff.
>
> **Figure R-1. Rank sweep of shared gradient energy.**
> https://anonymous.4open.science/r/EEG-FM-Rebuttal-Repo/Figure-R1.png
>
> Shared gradient energy $E_{\mathrm{shared}}(r)$ as a function of PCA rank $r$, showing that most cross-dataset shared energy is captured at low rank and motivating the compact PCA summary used in the subspace-affinity analysis.
>
> ## W3–Q3. Code release
>
> Thank you for emphasizing code availability. The complete implementation is already included in the supplementary material as a zip archive and has been publicly available to the community for about six months. Due to the ICML's double-blind policy, we cannot attach links to the codebase.

---

> > ### Author Rebuttal · Reviewer_Jzyh · 2026-04-03
> >
> > Thank you for the detailed rebuttal. I appreciate the added closest-to-native sanity check, it makes the benchmark behavior much easier to interpret. The pattern you show, with the largest gaps concentrated on motor imagery where native pipelines rely more on task-specific engineering, addresses a substantial part of my concern. The clarifications on CountSketch vs. PCA rank and on code availability are also helpful.
> >
> > I still think the paper should frame its conclusions carefully. The new analysis supports that the benchmark is not obviously broken, but it also suggests that some findings are best understood as results under a controlled common protocol.
> >
> > On the comparison standard: I agree that using each paper’s maximum best configuration would not be appropriate, since that would mix in unequal amounts of task-specific engineering and tuning and could favor whichever model was optimized most heavily in its original paper. In that context, the closest-to-native sanity check is exactly the right complementary validation.
> >
> > Overall, this rebuttal addresses a meaningful part of my concern, and I will increase my score to 4.

---

> > > ### Author Response · Authors · 2026-04-06
> > >
> > > Thank you for your reconsideration and helpful follow-up. We will incorporate all clarifications and additional analyses from our rebuttal into the revision. We will also revise the conclusion accordingly and state the claims more conditionally where appropriate.
> > >
> > > We also sincerely appreciate your note that you would increase your score to 4. If this remains your final assessment, we just wanted to gently note that the formal score update step may still be pending. We would be very grateful if you could kindly complete the corresponding process in the OpenReview system.

---

### Official Review · Reviewer_fM2N · 2026-03-13

**Soundness:** 2
**Presentation:** 3
**Significance:** 4
**Originality:** 2
**Overall Recommendation:** 3
**Confidence:** 5

**Summary:**

The paper presents benchmark results of EEG foundation models over 14 datasets including 10 paradigms: motor imagery (BCIC-2a, PhysioMI, Mimul11), emotion recognition (SEED, SEED-V, SEED-VII), sleep stage classification (HMC), seizure detection (Siena), mental stress assessment (Workload), abnormal detection (TUAB), event type classification (TUEV), visual target detection (Things-EEG-2), Alzheimer’s Disease recognition (ADFTD), and slowing event classification (TUSL).
It evaluates seven EEG foundation models: BENDR, BIOT, LaBraM, EEGPT, CBraMod, CSBrain, and REVE.
The evaluation is structured along three dimensions: (a) fine-tuning strategy (frozen-backbone, full-parameter, parameter-efficient via LoRA), (b) task setup (single-task vs multi-task with resampling to mitigate imbalance), and (c) classifier head (MLP with patch average pooling, dimension compression, or attention pooling).

**Compliance With Llm Reviewing Policy:**

Affirmed.

**Final Justification:**

I will maintain my grade

**Key Questions For Authors:**

1. You claim “all components are implemented within a unified open-source codebase,” but no link is provided—can you share the repository (and ideally a reproducibility checklist / instructions) or revise the claim?
2. How do you ensure fairness across EEG-FMs given that different models were pretrained with different signal scaling conventions (e.g., BIOT percentile scaling vs dividing $\mu V$ by 100)? Will you add model-specific input scaling (or an ablation showing robustness to scaling) so the benchmark does not penalize models for mismatched units?
3. Several datasets appear to be non-open (e.g., SEED family; Temple University datasets). What is your reproducibility plan for researchers without access?
4. Would you add a linear-probe baseline head (single linear layer) to complement the MLP heads and more directly assess whether pretrained features are linearly separable?
5. Can you temper or qualify the high-level conclusion about data size vs model size, given heterogeneity in architectures/datasets and the volume of analyses (risk of cherry-picking)? What additional controls or sensitivity analyses support that takeaway?
6. Nit: Please use the correct citation for MOABB, i.e., Zenodo DOI below.

**References**

Aristimunha, B., Carrara, I., Guetschel, P., Sedlar, S., Rodrigues, P., Sosulski, J., Narayanan, D., Bjareholt, E., Barthelemy, Q., Schirrmeister, R. T., Kobler, R., Kalunga, E., Darmet, L., Gregoire, C., Abdul Hussain, A., Gatti, R., Goncharenko, V., Thielen, J., Moreau, T., … Chevallier, S. (2025). *Mother of all BCI Benchmarks*. Zenodo. https://doi.org/10.5281/zenodo.10034223

**Strengths And Weaknesses:**

1. **Strength — Standardized evaluation design.** Using unified abstractions and a single pipeline to evaluate models under matched preprocessing and optimization conditions is a clear step in the right direction for comparability.
2. **Strength — Structured ablations.** The explicit factorization into fine-tuning strategy, task setup (single vs multi-task), and classifier head enables controlled comparisons rather than ad-hoc fine-tuning.
3. **Weakness (major) — Preprocessing/scaling mismatch across pretrained models.** Signals are converted from $\mu V$ to $V$, but the compared EEG-FMs were pretrained with different scaling conventions (e.g., BIOT per-channel per-window scaling to the 95th percentile; LaBraM/CBraMod dividing $\mu V$ by 100). Benchmarking a model on signals that are ~100× larger than what it was pretrained on is likely to make it fail for avoidable reasons; the benchmark should accommodate each model’s expected scaling (or otherwise justify the chosen standardization and how it affects fairness).
4. **Weakness — No code link despite “open-source” claim.** The paper states the benchmark is implemented in a unified open-source codebase, but provides no link. Without access, reproducibility cannot be verified. Either publish the code or remove/soften the open-source claim.
5. **Weakness — Dataset accessibility limits reproducibility.** Some included datasets (e.g., SEED family; Temple University datasets) are not open, which limits who can reproduce or extend the benchmark without institutional access.
6. **Weakness / suggestion — Missing very simple head baseline.** In addition to the MLP-based heads, it could be informative to include a single linear layer (linear probing) to test linear separability of representations.
7. **Weakness — Overstated high-level takeaway risk.** The “data size/model size” conclusion should be toned down because architectures and datasets differ and the breadth of analyses increases the risk of cherry-picking.

**Overall recommendation:** Weak Reject — solid and useful benchmarking direction, but major fairness (model-specific scaling) and reproducibility (code availability) concerns need to be addressed before publication.

---

> ### Author Rebuttal · Authors · 2026-03-31
>
> ## W1-Q1 / W3-Q3. Implementation status and reproducibility plan
>
> Thank you for emphasizing code availability. The implementation is included in the supplementary material as a zip file and has been publicly available to the community for about six months. Due to ICML's double-blind policy, we cannot attach any link to the codebase.
>
> On dataset accessibility, the benchmark uses public datasets, although some are request-based rather than direct-download. In particular, the SEED family and TUH resources require an application step, which makes them less convenient but not institution-restricted in the sense of being closed to outside researchers. This distinction matters for reproducibility: these datasets are publicly accessible through standard request channels, and our own access followed exactly that route. The final manuscript states this access model explicitly so that the reproducibility expectation is described precisely.
>
> ## W2-Q2. Fairness under model-specific scaling conventions
>
> We appreciate this concern, and we believe it mainly comes from an ambiguity between a common storage unit and a common model interface. In our benchmark, these two are not treated as the same.
>
> The raw format in MNE-Python stores signals in (volts)**V**, but this does not imply that every backbone receives the same effective scale at the model interface. In our pipeline, fairness is enforced by a **dual compatibility mechanism**. On the runtime side, model-specific data adaptation allows each backbone to ingest the same benchmark datasets without violating its pretraining assumptions. On the dataset-construction side, YAML-based preprocessing configurations make the generated samples match the expected input convention as closely as the released checkpoints allow. The benchmark therefore standardizes evaluation protocol and downstream budget, not by forcing all backbones into one incompatible amplitude convention, but by preserving checkpoint compatibility while holding the downstream comparison fixed.
>
> Concretely, BIOT keeps its 95th-percentile-based amplitude normalization; EEGPT follows its mV-scale convention; LaBraM, CBraMod and CSBrain use the 100 µV-style scaling in their released implementations; and REVE is run with the closest checkpoint-compatible preprocessing variant exposed by the released implementation. For REVE, the current interface uses a window-level z-score approximation rather than exact session-level statistics, and that approximation will be stated explicitly in the revision.
>
> ## W4-Q4. Single-Layer linear classification head setting
>
> Thank you for this suggestion. In **Table R-1**, we include a strict single-layer linear head as an additional head setting because it makes the head analysis more interpretable. In prior EEG-FM work, “linear probe” does not always correspond to a minimal-capacity readout: some settings still include multiple linear transformations or extra task-specific modules, and even a nominal linear head is often substantially larger than our average-pooling head because the linear layer is applied to all patch embeddings rather than to a single summary token. Reporting a strict linear head separately therefore lets us distinguish representation linear separability from decoder capacity much more cleanly, and makes the head comparison materially stronger.
>
> **Table R-1. Performance comparison of 7 EEG-FMs on 14 BCI tasks under the full-parameter multi-task setting with a single-layer linear classification head.**
> https://anonymous.4open.science/r/EEG-FM-Rebuttal-Repo/Table-R1.md
>
> ## W5-Q5 / W6. High-level takeaway and citation correction
>
> We thank the reviewer for highlighting this important scope issue.
>
> Existing work on EEG-FMs, including studies on LaBraM and CBraMod, has shown that scaling model size and training data can benefit EEG-FMs. Therefore, our benchmark focuses on building a unified benchmark to ensure fair comparison and provide diagnostic analyses with interpretable insights.
> During our diagnostic analyses of pretrain/downstream gradients, we observed that the gradients from pretrain reconstruction tasks and downstream classification tasks among different EEG-FMs are nearly orthogonal (cosine similarity lower than 0.08), which suggests that the pretrained objectives make inefficient use of information relevant to downstream tasks.
> Therefore, we conclude that **under released checkpoints and a matched downstream protocol, scale does not appear to be the dominant bottleneck at the present stage of EEG-FMs**.
>
> We also appreciate the MOABB citation correction and use the Zenodo entry in the final manuscript.

---

> > ### Author Rebuttal · Reviewer_fM2N · 2026-04-04
> >
> > Regarding code openness, you could have provided the repository anonymously on https://anonymous.4open.science/ (like you did for the additional tables and figures). The correctness of the code is a crucial aspect of a benchmark. Not providing it significantly limits what we can verify as reviewers.
> >
> > According to the description provided, the signal scaling seems to be done fairly. In particular, for REVE, I support the approximation of window-level statistics because extracting session-level statistics would be considered a data leak (unfair).
> > My other concerns are lifted

---

> > > ### Author Response · Authors · 2026-04-04
> > >
> > > Thank you for your follow-up response. We sincerely appreciate your support regarding our signal scaling approach (especially for REVE) and are glad to hear that all your other concerns have been successfully resolved.
> > >
> > > Regarding your concern about the code openness, **we would like to clarify a critical misunderstanding: the complete code was already provided at the time of our initial paper submission.**
> > >
> > > * **Where to find the code**: You can download our code in a .zip format directly from the OpenReview system. **Please simply click the "Supplementary Material" button located at the very top of our paper's OpenReview page**. It has been available there for all reviewers to examine since the paper submission deadline.
> > >
> > > * **Why we did not use an external repository link during rebuttal**: We strictly followed the ICML rebuttal policies. The official guidelines explicitly state that external anonymous links provided during the rebuttal phase can only be used for hosting additional tables and figures. Hosting code on external platforms like anonymous.4open.science during the rebuttal is considered a violation of the rebuttal rules. Therefore, we could only refer you to the originally submitted .zip file.
> > >
> > > Since the code has always been available in the supplementary material, we kindly request that you download the .zip file from OpenReview to verify the overall code. As this was your only remaining concern, we hope this clarification fully addresses it.
> > >
> > > Thank you again for your time and constructive feedback! Hope our clarification can resolve your misunderstanding and hope you can adjust your rating accordingly.

---

### Official Review · Reviewer_VQwz · 2026-03-16

**Soundness:** 3
**Presentation:** 2
**Significance:** 3
**Originality:** 2
**Overall Recommendation:** 4
**Confidence:** 3

**Summary:**

• The paper introduces EEG-FM-Bench, a standardized benchmark for evaluating EEG foundation models (EEG-FMs), motivated by the lack of consistent datasets, preprocessing pipelines, and evaluation protocols in prior work.

• The benchmark integrates 14 public EEG datasets across 10 paradigms and provides a unified evaluation framework covering preprocessing, training, and testing.

• Multiple EEG foundation models and time-series baselines are evaluated under different adaptation strategies, including frozen backbone transfer, LoRA fine-tuning, and full-parameter fine-tuning.

• The study also systematically analyzes single-task vs. multi-task training and different classifier head designs to understand their impact on downstream performance.

• Beyond benchmarking results, the paper includes diagnostic analyses (e.g., gradient alignment, representation similarity, and subspace affinity) to study transfer behavior and the relationship between pretraining objectives and downstream tasks.

• The experiments suggest that multi-task training often improves performance, frozen transfer remains limited, and reconstruction-based pretraining objectives may be misaligned with downstream classification tasks.

**Compliance With Llm Reviewing Policy:**

Affirmed.

**Final Justification:**

I choose to raise my score to 4

**Key Questions For Authors:**

1. For the benchmark design, I think the empirical study does produce some findings. However, a clear recipe of training a good EEG foundation model is still somewhat missing (I could see a lot pieces but a direct, clear recipe which compared against all state-of-the-art model is still missing).
2. The two main insights, "multi-task learning acts as a critical regularizer to mitigate overfitting in data-scarce EEG contexts" and "pre-training efficiency is currently limited by gradient conflicts between reconstruction objectives and downstream tasks" seems conflict with each other --> the authors only states "developing objectives or modules that capture discriminative features is critical" without giving preliminary studies on the potential redemy of the key observation makes the benchmark weaker in insight perspective.
3. The authors question on the scaling of the model/data size, according to the last point, it seems that whether is the data issue or the training objective issue is still unclear. The better way to make this conclusion should be a rigorous ablation study on data/model size instead of comparing single/multi-task learning.

**Limitations:**

yes

**Strengths And Weaknesses:**

Strengths

• Comprehensive benchmark design. The paper introduces a unified benchmark covering 14 EEG datasets across 10 paradigms, which is broader than many prior EEG-FM evaluations and helps address the lack of standardized evaluation in the field.

• Systematic experimental protocol. The study carefully separates several factors that are often confounded in prior work, including backbone models, fine-tuning strategies (frozen, LoRA, full fine-tuning), task organization (single-task vs. multi-task), and classifier heads, enabling more controlled comparisons.

• Diagnostic analysis beyond leaderboard results. In addition to reporting benchmark performance, the paper conducts analyses such as gradient alignment and representation similarity, which provide insights into the interaction between pretraining objectives and downstream tasks.


Weaknesses

• Limited actionable guidance despite extensive benchmarking. While the benchmark produces several empirical observations, the paper does not synthesize them into a clear recipe for training strong EEG foundation models or establish a configuration that consistently outperforms prior approaches.

• The two main insights are not fully reconciled. The paper argues that multi-task learning acts as a regularizer in data-scarce EEG settings, while reconstruction-based pretraining objectives conflict with downstream tasks due to gradient misalignment. However, the relationship between these observations is not clearly analyzed, and no preliminary experiments are provided to explore potential remedies.

• Scaling conclusions lack controlled ablation. The claim that compact architectures outperform larger models and that scaling behaves differently in EEG settings is interesting, but it is unclear whether this effect stems from model size, data scale, or training objectives. More controlled ablations would strengthen this conclusion.

---

> ### Author Rebuttal · Authors · 2026-03-31
>
> ## W1-Q1. Clear recipe and practical takeaway
>
> Thank you for highlighting the need for a clearer takeaway. The benchmark does support a concrete recipe, but its primary value is more diagnostic: it identifies the **key bottlenecks** of current EEG-FMs under controlled downstream adaptation.
>
> The first bottleneck is the **upstream-downstream interface**. If current pretraining already produced task-ready EEG representations, frozen transfer should remain competitive. Instead, the frozen-backbone gap is consistently large, indicating that current pretraining still does not yield sufficiently discriminative features for downstream use. The second point is **adaptation stability in the small-data EEG regime**. Once full adaptation is allowed, multi-task training often improves robustness, showing that the problem lies not only in representation quality, but also in unstable and overfit downstream adaptation. The third point is **task-specific decoder/interface design**: larger or more structured heads help mainly in motor imagery, while simpler pooling heads remain a more reliable default on most other tasks. Taken together, these results suggest that raw scale is not the dominant limiting factor: when the pretraining-downstream interface is weak and adaptation remains unstable, increases in parameter count or corpus size does not reliably translate into better transfer.
>
> This also leads to a practical takeaway. Under released checkpoints and a matched downstream budget, the strongest current default is to use an EEG-specific backbone, apply full-parameter adaptation, exploit multi-task supervision when labeled datasets can be pooled, and treat richer heads as task-specific tools rather than universal defaults. More importantly, the benchmark indicates where future progress is most likely to come from: improving **objective alignment and the adaptation interface**, rather than relying on scale alone.
>
> ## W2-Q2. Reconciling gradient conflict analysis
>
> Thank you for pressing on this point. These two findings are best understood not as competing conclusions, but as **two linked failure modes**.
>
> The gradient-misalignment result concerns the **entry point into downstream transfer**. MAE-based pretraining provides a useful initialization, but the frozen-transfer gap and the pretrain/downstream gradient mismatch show that the learned features remain only partially aligned with downstream discrimination. Thus, downstream adaptation is still necessary because the pretrained representation is not yet fully task-ready.
>
> The multi-task result concerns what happens **once adaptation is required**. Once fine-tuning starts, many EEG datasets are small and noisy, so isolated single-task optimization is easily dominated by variance and overfitting. In this regime, multi-task supervision acts as a stabilizer: it does not remove the pretraining mismatch, but it makes downstream adaptation less brittle. Although our analyses reveal optimization-gradient conflicts for some dataset pairs, the final results show that, even without explicit conflict handling, multi-task training still yields substantial gains on most small-data datasets. This suggests that, in the current EEG setting, the regularization and data-sharing benefits of joint training often outweigh the observed gradient interference.
>
> This is the key connection. Multi-task learning is not presented as a remedy for misaligned pretraining; it is a downstream compensatory mechanism once such misalignment already exists. One result explains **why** current EEG-FMs still need substantial adaptation; the other explains **how** that adaptation can be made more stable under small-data conditions.
>
> ## W3-Q3. Scaling conclusion and scope
>
> Thank you for raising the scaling issue. Existing EEG-FMs, including studies on LaBraM and CBraMod, has shown that scaling model size and training data can benefit EEG-FMs. Therefore, our benchmark focuses on building a unified benchmark to ensure fair comparison and provide diagnostic analyses with interpretable insights.
>
> During our diagnostic analyses of pretrain/downstream gradients, we observed that the gradients from pretrain reconstruction tasks and downstream classification tasks are nearly orthogonal (cosine similarity lower than 0.08), which suggests that the pretrained objectives make inefficient use of information relevant to downstream tasks. Therefore, we conclude that **under released checkpoints and a matched downstream protocol, scale does not appear to be the dominant bottleneck at the present stage of EEG-FMs**.
>
> This also helps explain why compact EEG-specific architectures remain highly competitive in the benchmark. Their advantage is not simply smaller size, but better inductive bias and transfer compatibility under controlled adaptation. So the scaling takeaway is a benchmark-level conclusion: **at the current stage of the field, better pretrain/downstream gradient alignment matters more than nominal scale alone**.

---

> > ### Author Rebuttal · Reviewer_VQwz · 2026-04-04
> >
> > I think the authors mostly addressed my comments and I hope the authors could include the new analysis / results to the final version.

---

> > > ### Author Response · Authors · 2026-04-04
> > >
> > > Thank you very much for your careful review and your constructive feedback. We sincerely appreciate your acknowledgment that our rebuttal has addressed your concerns.
> > >
> > > We also thank you for suggesting that we include the additional analyses and clarifications in the final version. We will revise the manuscript accordingly to improve the clarity of the practical takeaways, the discussion of multi-task learning versus pretraining misalignment, and the scope of our scaling-related conclusions.
> > >
> > > Thank you again for your helpful comments and support.

---

### Decision · Program_Chairs · 2026-04-30

**Decision:**

Accept (regular)

**Comment:**

The paper introduces a unified benchmark for systematic evaluation and diagnostic analysis of EEG foundation models on diverse datasets and experimental settings. The paper received 3 Weak Accepts and 1 Weak Reject. Reviewers generally agree on the importance and timeliness of the contribution. The main concerns are on benchmark validity, reproducibility (code completeness and clarity), and especially fairness in model comparison. The rebuttal was addressed most concerns about benchmarks, transparency, and code availability. However, one reviewer maintained a Weak Reject and raised technical concerns about inconsistencies between the described and actual preprocessing/scaling implementation. I recommend Weak Accept.